# MDP Playground: Controlling Orthogonal Dimensions of Hardness in Toy Environments

## Abstract

We present *MDP Playground*, an efficient benchmark for Reinforcement Learning (RL) algorithms with various dimensions of hardness that can be controlled independently to challenge algorithms in different ways and to obtain varying degrees of hardness in generated environments. We consider and allow control over a wide variety of key hardness dimensions, including delayed rewards, rewardable sequences, sparsity of rewards, stochasticity, image representations, irrelevant features, time unit, and action range. While it is very time consuming to run RL algorithms on standard benchmarks, we define a parameterised collection of fast-to-run toy benchmarks in OpenAI Gym by varying these dimensions. Despite their toy nature and low compute requirements, we show that these benchmarks present substantial challenges to current RL algorithms. Furthermore, since we can generate environments with a desired value for each of the dimensions, in addition to having fine-grained control over the environments' hardness, we also have the ground truth available for evaluating algorithms. Finally, we evaluate the kinds of transfer for these dimensions that may be expected from our benchmarks to more complex benchmarks. We believe that MDP Playground is a valuable testbed for researchers designing new, adaptive and intelligent RL algorithms and those wanting to unit test their algorithms.

## 1 Introduction

RL has succeeded at many disparate tasks, such as helicopter aerobatics, game-playing and continuous control (Abbeel et al., 2010; Mnih et al., 2015; Silver et al., 2016; Chua et al., 2018; Fujimoto et al., 2018; Haarnoja et al., 2018), and yet a lot of the low-level workings of RL algorithms are not well understood. This is exacerbated by the absence of a unifying benchmark for all of RL. There are many different types of benchmarks, as many as there are kinds of tasks in RL (e.g. Todorov et al., 2012; Bellemare et al., 2013; Cobbe et al., 2019; Osband et al., 2019). They specialise in *specific* kinds of tasks. And yet the underlying assumptions are nearly always those of a Markov Decision Process (MDP). We propose a benchmark which distills difficulties for MDPs that can be *generalised* across RL problems and allows to control these difficulties for more precise experiments.

RL algorithms face a variety of challenges in environments. For example, when the underlying environment is an MDP, however the information state, i.e., the state representation used by the agent is not Markov, we have *partial observability*, (see, e.g., Mnih et al., 2015; Jaakkola et al., 1995). There are additional aspects of environments such as having irrelevant features, having multiple representations for the same state, and the action range, that significantly affect agent performance. We aim to study what kind of failure modes can occur when an agent is faced with such environments and to allow other researchers the benefit of the same platform to be able to create their own experiments and gain high-level insights.

We identify *dimensions* of MDPs which help characterise environments, and we provide a platform with different instantiations of these dimensions in order to understand the workings of different RL algorithms better. To this end, we implemented a Python package, *MDP Playground*, that gives us complete control over these dimensions to generate flexible environments.

Furthermore, commonly used environments, that RL algorithms are tested on, tend to take a long time to run. For example, a DQN run on Atari took us 4 CPU days and 64GB of memory to run. Our

platform can be used as a low-cost testbed early in the RL agent development pipeline to develop and unit test algorithms and gain quick and coarse insights into algorithms.

The main contributions of this paper are:

- We identify dimensions of MDPs that have a significant effect on agent performance, both for discrete and continuous environments;
- We open-source a platform with fine-grained control over the dimensions; baseline RL algorithms can be run on these in as little as 30 seconds on a single core of a laptop;
- We study the impact of the dimensions on baseline agents in our environments;
- We evaluate *transfer* of some of these dimensions to more complex environments.

## 2 DIMENSIONS OF HARDNESS

We begin by defining basic *deterministic* version of MDP, followed by a POMDP and then motivate the dimensions of hardness. We define an MDP as a 7-tuple $(S, A, P, R, \rho_o, \gamma, T)$, where $S$ is the set of states, $A$ is the set of actions, $P : S \times A \times S \to S$ describes the transition dynamics, $R : S \times A \to \mathbb{R}$ describes the reward dynamics, $\rho_o : S \to \mathbb{R}^+$ is the initial state distribution, $\gamma$ is the discount factor and $T$ is the set of terminal states. We define a POMDP as an 9-tuple $(S, O, A, P, \Omega, R, \rho_o, \gamma, T)$, where $O$ represents the set of observations, $\Omega : S \times A \times O \to \mathbb{R}$ describes the probability of an observation given a state and action and the rest of the terms have the same meaning as for the MDP above.

To identify the dimensions of hardness, we went over the components of MDPs and POMDPs and tried to exhaustively list dimensions that could make an environment harder. This has resulted in many dimensions and a highly parameterisable platform. To aid in understanding we categorise the dimensions according to the component of (PO)MDPs they affect.

To clarify terminology, we will use information state to mean the state representation used by the agent and belief state to be equivalent to the full observation history. If the belief state were to be used as the information state by an agent, it would be sufficient to compute an optimal policy. However, since the full observation history isn't tractable to store for many environments, agents in practice stack the last few observations to use as their information state which renders it non-Markov.

An implicit assumption for many agents is that we receive immediate reward depending on only the current information state and action. However, this is not true even for many simple environments. In many situations, agents receive *delayed rewards* (see e.g. Arjona-Medina et al., 2019). For example, shooting at an enemy ship in Space Invaders leads to rewards much later than the action of shooting. Any action taken after that is inconsequential to obtaining the reward for destroying that enemy ship.

In many environments, we obtain a reward for a *sequence* of actions taken and not just the information state and action. A simple example is executing a tennis serve, where we need a sequence of actions which would result in a point if we served an ace. In contrast to *delayed rewards*, rewarding a sequence of actions addresses the actions taken which are consequential to obtaining a reward. Sutton et al. (1999) present a framework for temporal abstraction in RL to deal with such sequences.

Environments can also be characterised by their *reward sparsity* (Gaina et al., 2019), i.e., the supervisory reward signal is 0 throughout the trajectory and then a single non-zero reward is received at its end. This also holds true for our example of the tennis serve above.

Another characteristic of environments that can significantly impact performance of algorithms is *stochasticity*. The environment, i.e., dynamics $P$ and $R$, may be stochastic or may seem stochastic to the agent due to partial observability or sensor noise. A robot equipped with a rangefinder, for example, has to deal with various sources of noise in its sensors (Thrun et al., 2005).

Environments also tend to have a lot of *irrelevant features* (Rajendran et al., 2018) that one need not focus on. For example, in certain racing car games, though one can see the whole screen, we only need to concentrate on the road and would be more memory efficient if we did. This holds for table-based learners and approximators like Neural Networks (NNs). NNs additionally can even fit random noise (Zhang et al., 2017) and having irrelevant features is likely to degrade performance.

Another aspect we want to motivate is that of *representations*. The same underlying state may have many different external representations/observations, for example, *feature* space vs *pixel* space.

Mujoco tasks may be learnt in feature space vs directly from pixels, and Atari games can use the underlying RAM state or images. For images, various image transformations (*shift*, *scale*, *rotate*, *flip* and others; Hendrycks & Dietterich, 2019) may manifest as observations of the same underlying state and can pose a challenge to learning.

Further, we identify several additional dimensions for continuous control problems. For instance, for the task of reaching a target, we have *target radius* (see, e.g., Klink et al., 2019), a measure of the distance from the target within which we consider the target was successfully reached, *action range*, the range of actions that may be taken, *action loss weight*, a weight penalising actions, *time unit*, the discretisation of time, and *transition dynamics order*, the order of $P$ that is considered.

Dimensions of hardness of environments that we identify from the above discussion are (with the (PO)MDP component they impact in brackets):

- Reward Delay ($R$)
- Sequence Length ($R$)
- Reward Sparsity ($R$)
- Stochasticity ($P$ and $R$)
- Irrelevant Features ($O$)
- Representations ($O$)
- Action Range ($A$)
- Time Unit ($P$)
- Target Radius ($T$)
- Dynamics Order ($P$)

While we include only these for the main paper, Appendix A lists all dimensions that are controllable in MDP Playground. While many dimensions may seem a curse at first, it is also the nature of RL that different dimensions tend to be important in different specific applications. For instance, $terminal state costs$ are very important for many environments, however for the continuous environments we considered here, they were absent and hence unimportant. So, we allow setting terminal state costs (in addition to goal state rewards to distinguish good and bad terminal states) to be able to define toy environments where this dimension will be important. Another example is of $reward scale$. The agents we tested here rescale or clip rewards already and the effects of this dimension are not as important as they would be otherwise. The experiments here are only a glimpse into the power and flexibility of MDP Playground. Users can even upload custom $P$s and $R$s and custom images for representations $O$ and our platform will take care of injecting the other difficulties for them (wherever possible). This allows users to control different dimensions in the same base environment and even gain insights we haven't had so far in our extensive experiments.

We now mathematically highlight some of our dimensions of hardness to aid understanding. The information state of an agent to compute an optimal policy would need to stack the previous $n + d$ observation and action pairs from the environment where $n$ denotes a *sequence length* and $d$ denotes a *delay*, i.e., a sequence of actions needs to be followed to obtain a reward which may be delayed by a certain number of steps. Sparsity controls the number of elements in $S$ that are rewardable.

Additionally, the continuous control dimensions can mathematically be described as follows. The *target radius* sets $T = \{s \mid \|s - s_t\|_2 < target\ radius\}$, where $s_t$ is the target point. The *action range* sets $A$ to be equal to the Cartesian product of the ranges of the action dimensions where $A \subset \mathbb{R}^n$. The *action loss weight*, $\lambda$, shapes rewards to be $R(s, a)_{shaped} = R(s, a) - \lambda * \|a\|_2$. The *time unit*, $t$, sets $P(s, a) = s + \int_t P_{cont}(s, a)\, dt$ where $P_{cont}$ is the underlying continuous dynamics function. The *transition dynamics order*, $n$, sets $P$ to be in $C^n$, the set of functions differentiable $n$ times.

## 3 MDP PLAYGROUND

*MDP Playground* generates parameterised OpenAI Gym (Brockman et al., 2016) environments to facilitate researchers to benchmark algorithms across dimensions of hardness. It is also easily configurable, so that the environment *difficulty* can be controlled in a fine-grained manner. We now briefly describe the discrete and continuous environments followed by implementation details of selected dimensions.

**Discrete Environments** In the discrete case, $S$ and $A$ contain *categorical* elements, and we generate random instantiations of $P$ and $R$ after the remaining dimensions have been set. The generated $P$ and $R$ are deterministic and held fixed for the environment. We keep $\rho_o$ to be uniform over the non-terminal states, and $T$ is fixed to be a subset of $S$ based on a user-chosen *terminal state density*.

**Continuous Environments** In the continuous case, environments correspond to the simplest real world task we could think of that would be worth solving, moving a rigid body to a target point,

similar to (Haarnoja et al., 2017; Klink et al., 2019). $P$ is parameterised so that each action dimension affects the corresponding space dimension. $P$ is designed such that when the *transition dynamics order* is $n$, the $n^{th}$ derivative of $a$ is set to be equal to the action applied for *time unit* seconds on a body with configurable *inertia*. This is integrated over time to yield the next state. $R$ is designed such that the reward for the current time step is the distance travelled towards the target since the last step.

**Reward Delay** We delay the reward for a state-action pair by a non-negative integer number of timesteps, which we call the *delay*, $d$.

**Rewardable Sequence Length** For discrete environments, we reward only specific sequences of states of positive integer length $n$.[1] We consider states to be distinct along the sequences allowing for $\frac{(|S|-|T|)!}{(|S|-|T|-n)!}$ sequences. For the continuous environment of moving to a target, $n$ is variable since it already corresponds to a real world task.

**Reward Sparsity** For discrete environments, we define the *reward density $rd$* of sequences in terms of the fraction of possible sequences of length $n$ that are actually rewarded by the environment, given that $n$ is constant. If $num_r$ sequences are rewarded, we define the reward density to be $rd = num_r / \frac{(|S|-|T|)!}{(|S|-|T|-n)!}$ and the sparsity as $1 - rd$. For continuous environments, we do not define sparsity, but rather allow having a sparse or dense environment using a *make_denser* configuration option.

**Stochasticity** For discrete environments, we implement a *transition noise $t\_n \in [0, 1]$*. With probability $t\_n$, an environment transitions uniformly at random to a state that is not the *true* next state given by the generated $P$. We further implement a *reward_noise $\sigma_{r\_n} \in \mathbb{R}$* and add a normal random variable distributed according to $\mathcal{N}(0, \sigma^2_{r\_n})$ to the *true* reward. For continuous environments both these noises are normally distributed and directly added to the states and rewards.

**Irrelevant Features** For discrete environments, we introduce new discrete dimensions. Each dimension $i$ has its own transition function $P_i$ which is independent of all other transition functions. However, only one of these is *relevant* to calculate the reward function. Similarly, in continuous environments we label existing dimensions of $S$ and $A$ as irrelevant and do not consider them in the reward calculation.

**Representations** For discrete environments, when this aspect is enabled, each categorical state is associated with an image of a regular polygon which becomes the externally visible state $s$ to the agent. This image can further be transformed by *shifting*, *scaling*, *rotating* or *flipping*, which are applied at random to the polygon whenever an observation is generated. The transforms are only applied within a range so that the polygon is always present in the image. Examples of some generated states can be seen in Figure 8 in Appendix F. This dimension is currently not implemented for continuous environments because it would be too expensive.

**Very low cost of execution** Experiments with *MDP Playground* are cheap, allowing academics without special hardware to perform insightful experiments. Wall-clock times depend a lot on the algorithm, network size and dimensions of hardness. Nevertheless, to give the reader an idea of the runtimes involved, DQN experiments (with a network with 2 hidden layers of 256 units each) took on average 35s for a *complete* run of DQN for 20 000 environment steps. In this setting we restricted Ray RLLib (Liang et al., 2018) and the underlying Tensorflow (Abadi et al., 2015) to run on *one core of a laptop*.[2] This equates to roughly 30 minutes for the *entire* delay experiment shown in Figure 1a which was plotted using 50 runs (10 seeds $\times$ 5 settings for *delay*). Even when using the more expensive continuous or representation learning environments, training was only about 3-5 times slower. When parallelised on a cluster, complete experiments can finish in a few minutes.

## 4  EXPERIMENTS AND RESULTS

To demonstrate the usefulness of our benchmark, we evaluated *Rllib* implementations (Liang et al., 2018) of DQN (Mnih et al., 2015), Rainbow DQN (Hessel et al., 2018), A3C (Mnih et al., 2016) on discrete environments and DDPG (Lillicrap et al., 2016), TD3 (Fujimoto et al., 2018) and SAC

---

[1]In general, $d$ and $n$ would not be constants in real world environments, but for our toy environments, we use fixed $n$ and $d$.

[2]core-i7-8850H CPU – The full CPU specifications for a single core can be found in Appendix P.

(Haarnoja et al., 2018) on continuous environments over grids of values for the dimensions of hardness discussed above. We used fully connected networks except for pixel-based representations where we used Convolutional Neural Networks (CNNs) (LeCun et al., 1989). We first describe the discrete environment experiments, followed by those on continuous environments. All hyperparameters and the tuning procedure we used are available in Appendix M.

## 4.1 Discrete environments

We set $|S|$ and $|A|$ to 8 and the density of terminal states to 0.25 for the experiments. The *reward scale* is set to 1.0 whenever a reward is given by the environment. We generated random $P$s that were *completely connected*. This is done by selecting for each state $s$, for each action $a$, a random successor state $s'$, so that from each state there is a transition possible to every state in state space through one of the available actions. We do this to keep a regular structure for the $P$. This leaves the environments in our experiments with a high *bias* nature versus the high *variance* nature seen in RL benchmarks in practice. Having a very complex $P$ or $R$ itself can introduce "noise" into the evaluation of algorithms and require many iterations of training before we can see if the agent is learning. We leave this for complex benchmarks to capture as they are closer to real world use cases. For unit testing, especially, we need quick insights and given that most agents are agnostic to the choice of $P$ and $R$, it is beneficial to have what we term high *bias* environments to test whether agents are learning. We discuss some more aspects of such a design in Appendix C.

**Varying *reward delay*** Figures 1a-c, depict the mean and standard deviation over 10 runs for various *delays*. One run consists of 10 random seeds for the algorithm but uses a fixed seed for the environment. We plot the Area Under the Curve (AUC) which takes the mean over previous training rewards. As can be seen from the figure, all algorithms perform very well in the *vanilla environment* where the MDP is fully observable as the information state of the agent is equal to the MDP's state. For all algorithms, performance degrades in environments where the information state is non-Markov. Performance clearly degrades more as the information state needed to compute the optimal policy requires more observations to be stacked. It is interesting (and expected) that Rainbow DQN is somewhat more robust than DQN. The plots also show that DQN variants are more robust to *delay* as compared to A3C variants.

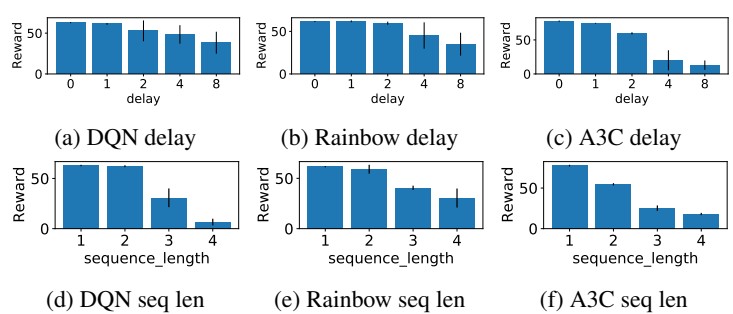

(a) DQN delay    (b) Rainbow delay    (c) A3C delay

(d) DQN seq len    (e) Rainbow seq len    (f) A3C seq len

Figure 1: AUC of episodic reward at the end of training for different agents for varying **delays** (top) and **sequence lengths** (bottom). Error bars represent 1 standard deviation. Note the reward scales.

**Varying *rewardable sequence length*** Results here are qualitatively similar to the ones for delay. However, we observe in Figures 1d-f that sequence length has a more drastic effect in terms of degradation of performance. The improvements of Rainbow DQN over DQN are also more pronounced for these harder problems.

The standard deviation in the preceding plots is high in many of the environments with non-Markov information state. Sometimes, however, the algorithms still managed to perform well, which emphasises that algorithms *can* perform well even when their assumption of having a Markov information state is violated; this is one of the possible explanatory factors for the fact that *tuning* seeds can lead to good results (refer Figure 5, Henderson et al., 2018). This also leads to noisy learning curves typically associated with RL as can be seen in Appendix J.

**Varying *representations*** For representations, we used image representations and applied various transforms (*shift*, *scale*, *rotate* and *flip*) one at a time and also all at once. We observed that the more transforms that are applied to the images, the harder it is for algorithms to learn, as can be seen in

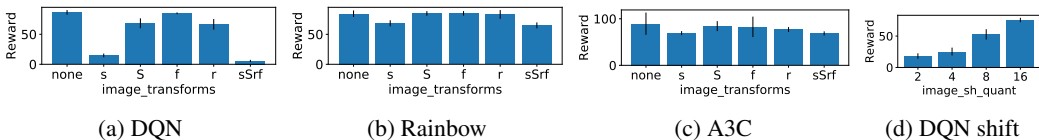

|  (a) DQN | (b) Rainbow | (c) A3C | (d) DQN shift |

Figure 2: AUC of episodic reward at the end of training for the different algorithms **when varying representation**. 's' denotes *shift* (quantisation of 1), 'S' *scale*, 'f' *flip* and 'r' *rotate* in the labels in the first three subfigures and *image_sh_quant* represents quantisation of the *shift*s in the DQN experiment for this. Error bars represent 1 standard deviation. Note the different reward scales.

Figures 2a-c. This was to be expected since there are many more combinations to generalise over for the agent. What was unexpected was that the most problematic transform for the algorithms to deal with was *shift*. Despite the spatial invariance learned in CNNs (LeCun, 2012), our results imply that that seems to be the hardest one to adapt to. As these trends were strongest in DQN, we evaluated further ranges for the individual transforms for DQN. *shift*ing had the most possible different combinations that could be applied to the images. Therefore, we quantised the *shift*s to have fewer possible values. Figure 2d shows that DQN's performance improved with increasing quantisation of *shift* ($\Rightarrow$ fewer possible values). We noticed similar trends for the other transforms as well, although not as strong as they do not have as many different values as *shift* (see Appendix G). This indicates that *shift* and other types of invariance do not come for free and that one needs to have sufficient amount of samples for the algorithm to become invariant to the transforms we desire. This is also a good sanity check that our toy environments allow such insights and other potential insights we may not yet be aware of.

Results for further dimensions and baselines can be found in the appendix. See Appendices G and I that show additional interesting results for varying *transition noise*, *reward noise*, *reward sparsity*. Further, even though our own focus with the benchmark is designing and unit testing deep RL, we believe it is also a valuable tool for theoreticians and we evaluated tabular baselines Q-learning (Sutton & Barto, 2018), Double Q-learning (van Hasselt, 2010) and SARSA (Sutton & Barto, 2018) on the discrete non-image based environments with similar qualitative results to those for deep agents. These can be found in Appendix H.

## 4.2 CONTINUOUS ENVIRONMENTS

We set the state and action space dimensionalities to 2. The state space range for each dimension was $[-10, 10]$ while the default action space range was $[-1, 1]$. The task would terminate when an algorithm would reach the *target point*, or after at most 100 timesteps. We focus on results for DDPG as results for TD3 and SAC are qualitatively similar (see Appendix G).

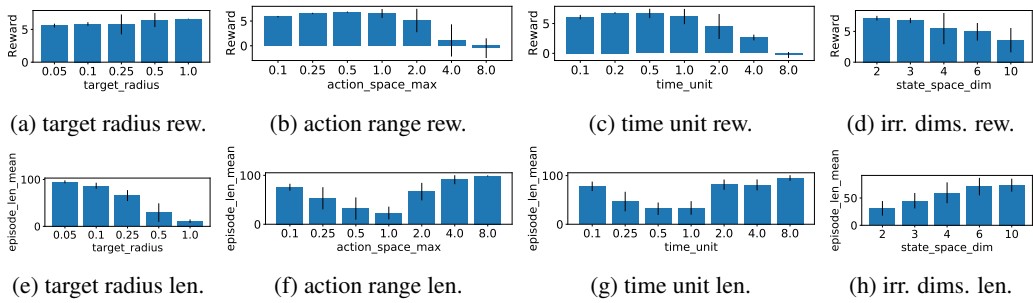

| (a) target radius rew. | (b) action range rew. | (c) time unit rew. | (d) irr. dims. rew. |
| (e) target radius len. | (f) action range len. | (g) time unit len. | (h) irr. dims. len. |

Figure 3: AUC of episodic reward (top) and lengths (bottom) for DDPG at the end of training. Error bars represent 1 standard deviation. Note the different y-axis scales.

**Varying *action range*** We observed that the total reward gets worse for action *max* > 1. Up until the value of 1, the episode lengths decreased as we would desire (see Figures 3b & 3f). This can be attributed to the fact that the exploration schedules for the studied agents take the max range available and explore based on that. But, as can be seen from these results, tuning these ranges or adapting exploration mechanisms can produce substantially better results.

**Varying *time unit*** We observed that increasing the *time unit* helps up to a point, until which the episode lengths also go down (see Figures 3c & 3g). Thus, tuning this value to learn how long we must play an action will also impact performance of algorithms significantly.

**Varying *target radius*** The *target radius* is a value which is generally set to a small enough value to be able to say that the algorithm has reached the target. However, we noticed that, for small values, all the continuous control agents oscillated around the target to reach it exactly. This can be observed in Figure 3a and 3e, where we note that even though the task *was* learnt for different *target radii*, the episode lengths were shorter for larger radii as the agents kept oscillating outside the radius. Even for such a simple task all evaluated algorithms failed to adapt to performing fine-grained control near the target. We hypothesise that the agents did not learn to slow down close to the goal. Given more experience close to the goal, we expect the agents to be able to learn this behaviour.

**Varying *irrelevant features*** We observed that introducing *irrelevant dimensions* to the control problem, while keeping the number of relevant dimensions fixed to 2, decreased an agent's performance (see Figures 3d & 3h). We believe this is because irrelevant features interfere with the learning process.

Additional interesting results, including results for *action loss weight* can be found in Appendix G.

## 5 USING MDP PLAYGROUND

We believe MDP Playground offers so much power and flexibility that we can only list a few sample use cases here and let the user choose to explore further what even we might not have thought of.

### 5.1 INSIGHTS INTO EXISTING AGENTS

**Design and Analyse Experiments** We allow the user the power to inject dimensions into a base environment in a fine-grained manner and analyse results using the 1-D plots from before or radar plots with user defined weights for the plots. Since, different users might be interested in different dimensions, these are loaded dynamically from the data. For instance, radar plots for the dimensions we varied in our experiments can be seen as in Figures 4a and 4b.

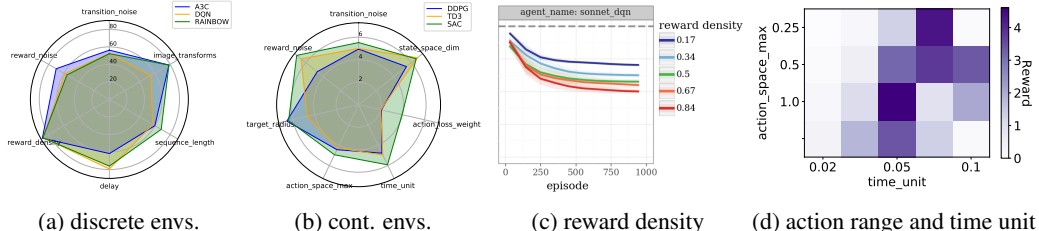

(a) discrete envs.      (b) cont. envs.      (c) reward density      (d) action range and time unit

Figure 4: Analysing and Debugging

**Varying Multiple Dimensions** It is possible to vary multiple dimensions at the same time in the same base environment. For instance, Figure 4d shows the diagonal relationship between varying the *action range* and *time unit* together. The platform allows this for all of dimensions that can be controlled together. Even more such experiments can be found in Appendix I, including varying both $P$ and $R$ *noise*s together in discrete environments and interactions between $P$ noise and the *target radius*, and interaction of the order of the transition dynamics with both *time unit* and *action range*.

**Transfer to complex environments** We have designed wrappers for Atari and Mujoco which can be used to inject some of the dimensions we have for toy environments as well. For instance, for *time unit*, there is an optimal value of the *time unit* with decreasing performances on either side of this optimum. For all three agents (Figure 5a-5c), we see peak performance for a *time unit* $0.4$ smaller than that of the *tuned* vanilla environment. The same holds for SAC on *Pusher* whereas DDPG and TD3 failed to solve *Pusher* (see Appendix K). The three agents also peaked on the toy environment at similar values (DDPG and TD3 at $0.2$ and SAC at $0.5$ closely followed by $0.2$). All three agents peaked on *Reacher* at *time unit* $1.0$. We attribute this to the *frame skip* of *Reacher* being $2$ while that of *HalfCheetah* and *Pusher* is $5$. What does not transfer from the toy environments is that the underlying optimal time unit is much smaller for the complex environments ($0.02$s vs $0.2$s). This is

likely because we require much more fine-grained control for high-dimensional environments. At the same time, this also indicates *time unit* should not be infinitesimally small to achieve too fine-grained control since there is an optimal *time unit* for which we should repeat the same action (Biedenkapp et al., 2020). This also has important consequences in environments where acquiring observations is expensive (Huang et al., 2019).

Here we have described just one of the dimensions and environments. However, we performed extensive experiments for different environments and dimensions in Appendix D. They all supported our argument that the high-level trends for the dimensions are similar to the

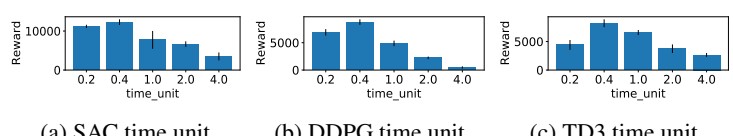

(a) SAC time unit     (b) DDPG time unit     (c) TD3 time unit

Figure 5: AUC of episodic reward at the end of training on HalfCheetah **varying time unit** (bottom). Error bars represent 1 standard deviation. Note the different y-axis scales.

trends on the toy environments. This might seem obvious to some readers after the fact, but the fact that these insights could have been gained directly on the toy environments without trying them on complex environments shows the potential insights that may be gained on toy environments.

## 5.2 DESIGNING NEW AGENTS

We hope our benchmark will help identify the inductive biases needed for designing new RL algorithms without getting bogged down by other sources of "noise" in the evaluation just as MNIST helped to identify the inductive bias of convolutions. MNIST by no means represents the true distribution of image data in the real world but it retains the key properties needed to identify inductive biases needed to perform well on image data. In the same way we believe we retain key dimensions of the problems which need inductive biases to be identified for them. The fact that CNNs can classify and learn even random noise (see, e.g., Zhang et al., 2017; Arpit et al., 2017) shows that even random data should be sufficient to identify inductive biases as long as the key dimension is present in the data.

## 5.3 DEBUGGING EXISTING AGENTS

Analysing how an agent performs under the effect of various dimensions can reveal unexpected aspects of an agent. For instance, when using bsuite agents, we noticed that when we varied our environment's *reward density*, the performance of the bsuite Sonnet DQN agent would go up in proportion to the density (see Figure 4c). This did not occur for other bsuite agents. This seemed to suggest something different for the DQN agent and when we looked at DQN's hyperparameters we realised that it had a fixed $\epsilon$ schedule while the other agents had decaying schedules. Such insights can easily go unnoticed if the environments used are too complex. We know of many researchers who have had frustrating experiences debugging in complex environments, where it's unclear whether a hyperparameter was the cause or the difficulty of the environment itself and such toy environments aid debugging by removing the "noise" of complex, real environments.

## 6 DISCUSSION AND RELATED WORK

To the best of our knowledge, we are the first to perform a principled study of how significant aspects such as non-Markov information states, irrelevant features, representations and low-level dimensions, like time discretisation, affect agent performance. MDP Playground facilitates easy study of the presented dimensions, giving us an abstraction bridging MDPs and the real world.

Our framework is very well suited to designing, developing and unit-testing new algorithms. No other benchmark that we know of offers fine-grained control over orthogonal dimensions of difficulty for agents all of which can be applied in the same base environment.

In the case of Rainbow and vanilla DQN we demonstrated how our environments can be used to compare extensions of algorithms and how one mitigates issues of the other. We show how learning

curve graphs with hardness inserted look reminiscent of typical RL curves. For example, for sequence lengths (and other dimensions in Appendix J) when the information state is non-Markov, we see the typical noise associated with RL algorithms (see, e.g., Henderson et al., 2018).

The concurrently developed *behaviour Suite for RL* (bsuite; Osband et al., 2019) is the closest related work to MDP Playground. Osband et al. (2019) collect known (toy) environments from the literature and use these to try and characterise agents based on their performance on these environments. Instead of providing individually controllable dimensions of hardness, bsuite provides environments that exhibit some of our dimensions of hardness that can *not* always be controlled by a user. Further, most environments in bsuite can be seen as an intermediate step between our MDPs and even more complex environments. One notable distinction between bsuite and MDP Playground is that bsuite provides no evidence of relevance of the contained environments to more complex problems, whereas we demonstrate that the identified trends on dimensions of hardness are not only relevant on toy environments but also transfer to more complex environments. Finally, bsuite offers no continuous benchmarks, whereas MDP Playground provides both discrete and continuous environments.

Maillard et al. (2014) defines a novel theoretical metric for defining hardness of MDPs. It captures difficulties within MDPs when the true state of the MDP is known. However, a large part of the hardness in our MDPs comes from the agent not knowing the optimal information state to use. It'd be interesting to design a metric which captures this aspect of hardness as well.

Further benchmark environments include *Procgen* (Cobbe et al., 2019), *Obstacle Tower* (Juliani et al., 2019) and *Atari* (Bellemare et al., 2013). Procgen adds various heterogeneous environments and tries to quantify generalisation in RL. In a similar vein, Obstacle Tower provides a generalization challenge for problems in vision, control, and planning. These benchmarks do not capture orthogonal dimensions of difficulty and as a result, they do not have the same type of fine-grained control over their environments' difficulty and neither can each dimension be controlled independently. We view this as a crucial aspect when benchmarking new algorithms. Dulac-Arnold et al. (2020) has some overlapping dimensions with our platform but it consists of much more specific environments, and only those that are continuous. This means, they are further down in the RL development pipeline than MDP Playground or bsuite which target the toy environment domain and thus complementary to our approach. More details on related work can be found in Appendix E.

We believe we need a *curriculum* of benchmarks for RL. In this hierarchy, our benchmark is a low-level, high bias benchmark that should be the first test for newly implemented agents and bsuite comes at the secondary level and other more specialised benchmarks that capture more specific environments like (Cobbe et al., 2019; Cobbe et al., 2019; Juliani et al., 2019) are even more specialised.

## 7 CONCLUSION AND FUTURE WORK

We introduced a low-cost benchmark to test RL algorithms in environments with varying and controllable dimensions of hardness. We demonstrated their effects on well-known RL algorithms. The platform allows us to disentangle various factors that make RL benchmarks hard by providing fine-grained control over various dimensions. We further demonstrated how the performance of the studied agents is adversely affected by some dimensions of hardness of environments which are under the control of researchers even in real world environments, such as the time-unit. We will open-source our code to facilitate better, cheaper, more reproducible, and more directed benchmarking in the RL community. We also evaluated the transfer of the dimensions to more complex environments and showed how similar challenges are present there as well. While we tried to exhaustively identify dimensions of hardness, it is unlikely that we have captured *all* orthogonal dimensions of hardness in RL. We welcome more dimensions that readers think will help us encapsulate further challenges in RL.

We believe providing theoretical and practical researchers such a powerful tool is like providing a programming language. Not every part of the MDP *search space* may be interesting to a user, but they have the power to define their own toy MDPs easily with high-level *primitives* and not be limited to the glimpse we have provided here. Future work should try to research a stronger link for toy MDPs with real environments. We are convinced that MDP Playground can have a great impact on RL research and is an important ingredient towards an era with more reproducible, generalisable and better understood RL research.

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

## A    DIMENSIONS OF HARDNESS IN MDP PLAYGROUND

We list here the hardness dimensions for *MDP Playground*. Details on each meta-feature can be found in the documentation for the class `mdp_playground.envs.RLToyEnv` in the accompanying code.

- Reward Delay
- Rewardable Sequence Length
- Reward Sparsity
- $P$ Noise
- $R$ Noise
- Irrelevant Features
- Transforms for Representation Learning
- Reward Shift
- Reward Scale
- State space size/dimensionality
- Action space size/dimensionality
- Terminal State Density
- Terminal State Reward
- Relevant Dimensions (for both state and action spaces)
∗ Only for discrete environments:

  - Image Representations
  ∗ Only for Image Representations:
    - Shift Quantisation
    - Scale Range
    - Rotation Quantisation
∗ Only for continuous environments:

  - Target Point
  - Target Radius
  - Time Unit
  - Inertia
  - State Space Max
  - Action Space Max
  - Transition Dynamics Order
  - Reward Function
∗ Currently fixed hardness dimensions:

  - Initial State Distribution

### A.1    ADDITIONAL SPARSITY OPTION FOR SEQUENCES

With regard to sparsity, recall the tennis serve again. The point received by serving an ace would be a sparse reward. We as humans know to reward ourselves for executing only a part of the sequence correctly. Rewards in continuous control tasks to reach a target point (e.g. in Mujoco, Todorov et al., 2012), are usually dense (such as the negative squared distance from the target). This lets the algorithm obtain a dense signal in space to guide learning, and it is well known (Sutton & Barto, 2018) that it would be much harder for the algorithm to learn if it only received a single reward at the target point. The environments in MDP Playground have a configuration option, $make\_denser$, to allow this kind of reward shaping to make the reward denser and observe the effects on algorithms. To achieve this, when $make\_denser$ is $True$, the environment gives a fractional reward if a fraction of a rewardable sequence is achieved in discrete environments. For continuous environments, for the move to a target point reward function, this option toggles between giving a dense reward as described in the main paper and giving a sparse reward when the agent is within the *target radius*.

# B  ALGORITHM FOR GENERATING MDPS

---

**Algorithm 1** Generating MDPs with MDP Playground

---

1: **Input:**
2: number of states $|S|$,
3: number of actions $|A|$,
4: reward delay $d$,
5: length of rewardable sequences $n$,
6: density of rewardable sequences $rd$,
7: transition noise $t\_n$ or $\sigma_{t\_n}$,
8: reward noise $\sigma_{r\_n}$,
9: `reward_scale`,
10: `reward_shift`,
11: `term_state_reward`,
12: `make_denser`,
13: `terminal_state_density`
14: `relevant_dimensions`
15: Hardness dimensions specific to continuous environments:
16: `target_point`
17: `target_radius`
18: `transition_dynamics_order`
19: `time_unit`
20: `inertia`
21:
22: **function** INIT_TRANSITION_FUNCTION():
23:     **if** discrete environment **then**
24:         **for** each state $s$ **do**                          ▷ For generating a completely connected $P$
25:             Set possible successor states: $S' = S$
26:             **for** each action $a$ **do**
27:                 Set $P(s, a) = s'$ sampled uniformly from $S'$ and remove $s'$ from $S'$
28:         **if** irrelevant features **then**
29:             Generate dynamics $P_{irr}$ of irrelevant part of state space as was done for $P$
30:     **else**
31:         Do nothing as continuous environments have a fixed parameterisation
32:
33: **function** INIT_REWARD_FUNCTION($n$):
34:     **if** discrete environment **then**
35:         Randomly sample $rd * \frac{(|S|-|T|)!}{(|S|-|T|-n)!}$ sequences and store in `rewardable_sequences`
36:     **else**
37:         Do nothing as continuous environments have fixed options for the reward function
38:

---

39: **function** TRANSITION_FUNCTION($s, a$):
40:     **if** discrete environment **then**
41:         $s' = P(s, a)$
42:         **if** $\mathcal{U}(0, 1) < t\_n$ **then**
43:             $s' =$ a random state in $S \setminus \{P(s, a)\}$                              ▷ Inject noise
44:         **if** irrelevant features **then**
45:             Execute dynamics $P_{irr}$ of irrelevant part of state space and concatenate with $s'$
46:         **if** representation learning **then**
47:             $s' =$ image of corresponding polygon(s) with applied selected transforms
48:     **else**
49:         Set $n = $ `transition_dynamics_order`
50:         Set $a^n = a$                              ▷ Superscript $n$ represents $n^{th}$ derivative
51:         Set $s^n = a^n / inertia$      ▷ Each state dimension is controlled by each action dimension
52:         **for** i in reversed(range(n)) **do**
53:             Set $s^i_{t+1} = \sum\limits_{j=0}^{n-i} s^{i+j}_t \cdot \frac{1}{j!} \cdot time\_unit^j$                              ▷ $t$ is current time step.
54:         $s_{t+1} += \mathcal{N}(0, \sigma^2_{t\_n})$
55:         $s' = s_{t+1}$
56:     **return** $s'$
57:
58: **function** REWARD_FUNCTION($s, a$):
59:     $r = 0$
60:     **if** irrelevant features **then**
61:         $s = s[$ `relevant_dimensions` $]$        ▷ Sub-select the part of state space relevant to reward
62:     **if** discrete environment **then**
63:         **if** not `make_denser` **then**
64:             **if** state sequence of $n$ states ending $d$ steps in the past is in `rewardable_sequences` **then**
65:                 $r = $ `reward_scale`
66:         **else**
67:             **for** $i$ in range($n$) **do**
68:                 **if** sequence of $i$ states ending $d$ steps in the past is a prefix sub-sequence of a sequence in `rewardable_sequences` **then**
69:                     $r += $ i/n
70:     **else**
71:         $r = $ Distance moved towards the `target_point`
72:     $r += \mathcal{N}(0, \sigma^2_{r\_n})$
73:     $r* = $ `reward_scale`
74:     $r += $ `reward_shift`
75:     **if** reached terminal state **then**
76:         $r += $ `term_state_reward`
77:     **return** $r$
78:
79: **function** MAIN():
80:     INIT_TERMINAL_STATES()                ▷ Set $T$ according to `terminal_state_density`
81:     INIT_INIT_STATE_DIST()                ▷ Set $\rho_o$ to uniform distribution over non-terminal states
82:     INIT_TRANSITION_FUNCTION()
83:     INIT_REWARD_FUNCTION()

## C    Design Decisions

### C.1    Regular structure of transition function for experiments

It is always possible to design *adversarial $P$*s (Nau, 1983; Ramanujan et al., 2010) which can be made arbitrarily hard to solve. So, we imposed a regular structure on $P$. If we have an environment, say the Earth, where a big reward (e.g. hidden treasure) is placed in an unknown and deliberately unexpected location. Then evaluating the intelligence of an agent on such an environment clearly does not give us a true measure of agent intelligence. This is also a problem with many benchmark environments, e.g., *qbert* has a bug which allows the agent to achieve a very large number of points (Chrabaszcz et al., 2018) and breakout has a scenario where, if an agent creates a hole through the bricks, it can achieve a very large number of points. Even though the latter *is* a sign of intelligent behaviour, it skews the distribution of rewards and introduces variance in the evaluation. In keeping with the low-level and high bias nature of our toy environments, we leave these to more complex benchmarks with more variant environments and instead impose a regular structure on $P$. We do allow completely random generation of $P$s which do not have a regular structure and can have much more variance in their connections, but as we noticed in initial experiments this produces variant environments and leads to noisy evaluations. We also plan to let users also generate or provide their own $P$ so that they might play with toy versions of environments they are interested in.

# D EFFECT OF DIMENSIONS ON MORE COMPLEX BENCHMARKS

We tested the trends of the dimensions on more complex Atari and Mujoco tasks. For Atari, we ran the agents on *beam_rider*, *breakout*, *qbert* and *space_invaders* when varying the dimensions *delay* and *transition noise*. For Mujoco, we ran the agents on *HalfCheetah*, *Pusher* and *Reacher* using *mujoco-py* when varying the dimensions *time unit* and *action range*. We evaluated $5$ seeds for $500k$ steps for *Pusher* and *Reacher*, $3M$ for *HalfCheetah* and $10M$ ($40M$ frames) for Atari. The values shown for *action range* and *time unit* are relative to the ones used in Mujoco.

**Varying *transition noise*** We observe similar trends for injecting transition noise into Atari environments for all three agents as for the toy environments. We also observe that for some of the environments, transition noise actually helps improve performance. This has also been observed in prior work (Wang et al., 2019). This happens when the exploration policy was not tuned optimally since inserting transition noise is almost equivalent to $\epsilon$-greedy exploration for low values of noise. We also observed a similar effect for the toy environments in Figure 13 in Appendix G. However, we also observe that performance drop is different for different environments. This is to be expected as there are other dimensions of hardness which we cannot control or measure for these environments.

**Varying *reward delay*** We see that performance drops for the delay experiments when more delay is inserted. For *qbert* (Figure 6c), these drops are greater on average across the agents. However, for *breakout* (Figure 6b), in many instances, we don't even see performance drops. In *beam_rider* (Figure 6a) and *space_invaders* (Figure 6d), the magnitude of these effects are intermediate to *breakout* and *qbert*. This trend becomes clearer when we also look at Figures 82b-l in Appendix K. We believe this is because large delays from played action to reward are already present in *breakout*, which means that inserting more delays does not have as large an effect as in *qbert* (Figures 6c). Agents are strongest affected in qbert which, upon looking at gameplay, we believe has the least delays from rewarding action to reward compared to the other games. The trends for delay however were noisier. Many considered environments tend to also have repetitive sequences which would dilute the effect of injecting delays. Many of the learning curves in Appendix L, with delays inserted, are indistinguishable from normal learning curves. We believe that, in addition to the motivating examples, this is empirical evidence that delays are already present in these environments and so inserting them does not cause the curves to look vastly different. In contrast, when we see learning curves for transition noise, we observe that, as we inject more and more noise, training tends to a smoother curve as the agent tends towards becoming a completely random agent.

To analyse transfer of dimensions between toy and complex benchmarks, we use the Spearman rank correlation coefficient between corresponding toy and complex experiments for performance across different values of the dimension of hardness. The Spearman correlation was $>= 0.7$ for 19 out of 24 experiments and a positive correlation for four of the remaining five. DQN with delays added on breakout was the only experiment with correlation 0.

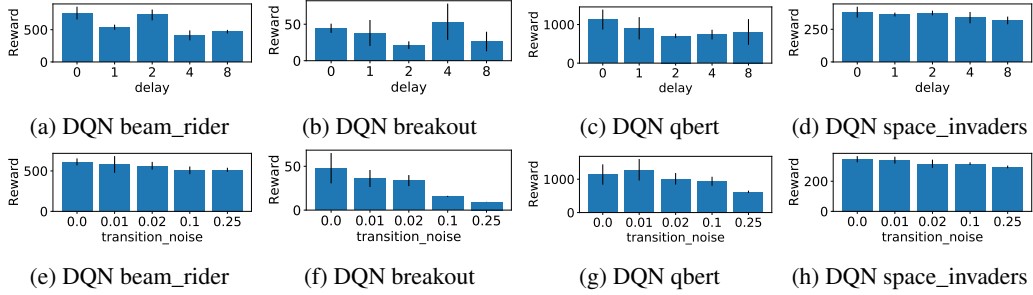

Figure 6: AUC of episodic reward for DQN on various environments at the end of training. Error bars represent 1 standard deviation. Note the different y-axis scales.

**Varying *action range*** We observed similar trends as discussed prior, in that there was an optimal value of *action range* with decreasing performances on either side of this optimum. Figure **??** shows this for all considered agents on HalfCheetah (for SAC and DDPG, runs for *action range* values $>= 2$ and $>= 4$ crashed and are absent from the plot). Qualitatively similar results on the other environments are given in Appendix K. This supports the insight gained on our simpler environment

that tuning this value may lead to significant gains for an agent. For already tuned environments, such as the ones in *Gym*, this dimension is easily overlooked but when faced with new environments setting it appropriately can lead to substantial gains. In fact, even in the tuned environment setting of *Gym*, we found that all three algorithms performed best for an *action range* 0.25 times the value found in *Gym* for *Reacher* (Figures 83c, 83k, 83g in Appendix K). This observation is representative for the types of insight our benchmark can yield for RL algorithm design, as ideally an agent would adaptively set these bounds since these are under its control. Moreover, the learning curves in Appendix L further show that for increasing *action range* the training gets more variant. The difference in performances across the different values of *action range* is much greater in the complex environments. We believe this is due to correlations within the multiple degrees of freedom as opposed to a rigid object in the toy environment. Due to the high bias nature of the toy environments, the high level trend of an optimal action range transfers to more complex ones.

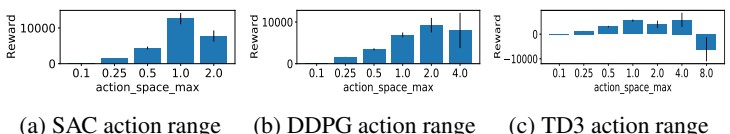

(a) SAC action range     (b) DDPG action range     (c) TD3 action range

Figure 7: AUC of episodic reward at the end of training on HalfCheetah **varying action range**. Error bars represent 1 standard deviation. Note the different y-axis scales.

# E MORE ON RELATED WORK

Our benchmark is also designed with long term AGI in mind. Dimensions like identifying delays and sequences will be essential to solving AGI. Most current algorithms lack such capability (Pearl, 2018).

Many of the other benchmarks mentioned in the main paper are largely vision-based, which means that a large part of their problem solving receives benefits from advances in the vision community while our benchmarks try to tackle pure RL problems in their most toy form. This also means that our experiments are extremely cheap, making them a good platform to test out new algorithms' robustness to different challenges in RL.

A parallel and independent work along similar lines as the MDP Playground, which was released a month before ours on arXiv, is the Behaviour Suite for RL (bsuite, Osband et al. (2019)). In contrast to our *generated* benchmarks, that suite *collects* simple RL benchmarks from the literature that are representative of various types of problems which occur in RL and tries to characterise RL algorithms. Unlike their framework, where currently there is no toy environment for Hierarchical RL (HRL) algorithms, the rewardable sequences that we describe would also fit very well with HRL. Additionally, we also have a toy continuous environment whereas bsuite currently only has discrete environments. An important distinction between the two platforms could be summed up by saying that they try to characterise *algorithms* while we try to characterise *environments* with the aim that new adaptable algorithms can be developed that can tackle environments of desired difficulty. They also do not generate completely random $P$ and $R$ for their environments like we do, which would help avoid algorithms overfitting to certain benchmarks.

For some readers, it might feel obvious that injecting many of these dimensions causes difficulties for agents. But to the best of our knowledge no other work has tried to collect all *orthogonal* dimensions in one place and study them comprehensively and what aspects transfer from toy to more complex environments.

The nature of the toy environments is one of high bias. We believe that the *transfer* of the hardness dimensions from toy to complex environments occurs because the algorithms we have tested are environment agnostic and usually do not take aspects of the environment into account. Q-learning for instance is based on TD-errors and the Bellman equation. The equation is agnostic to the environment and while adding deep learning may help agents learn representations better, it does not remove the problems inherent in deep learning. While it's nice to have general algorithms that may be applied in a black box fashion, by studying the dimensions we have listed and their effects on environments, we will gain deeper insights into what is needed to design better agents.

An additional comment can be made comparing the continuous and discrete complex environments comparisons to the toy benchmarks. The noise in comparing the discrete environments was higher and we believe this is due to the discrete environments being much more sparse and having many more *lucky areas* that can be exploited as with the *qbert* bug and *breakout* strategy mentioned. In comparison, continuous environments usually employ a dense reward formulation in which case the value functions are likely to be continuous.

Algorithms like DQN (Mnih et al., 2015) were applied to many varied environments and produce very variable performance across these. In some simple environments, DQN's performance exceeds human performance by large amounts, but in other environments, such as Montezuma's revenge, performance is very poor. For some of these environments, e.g. Montezuma's revenge, we need a very specific sequence of actions to get a reward. For others, there are different delays in rewards. A problem with evaluating on these environments is that we have either no control over their difficulty or little control such as having different difficulty levels. But even these difficulty levels, do not isolate the confounding factors that are present at the same time and do not allow us to control the confounding factors *individually*. We make that possible with our Dimensions.

When designing the platform, we went over the components of an MDP and tried to exhaustively add as many parameterisable dimensions as possible with the condition that they are all *orthogonal* and can be applied independently of each other. In a sense, this is an attempt to capture fundamental dimensions of hardness in the same way that human cognition is founded, in part, on four different systems and endow humans with abstract reasoning abilities (Spelke & Kinzler, 2007). We don't try to capture, say credit assignment or generalisation as dimensions. These are to be dealt with

at a higher level the same way that intelligent behaviour and reasoning arise from the interplay of different underlying cognitive systems but which process objects or space at a lower sensory level.

### E.1 DEBUGGING WITH MDP PLAYGROUND

We discuss here further 2 more examples of how the toy environments helped us debug RL algorithms in practice.

When merging some of our environments into bsuite, we noticed that when we varied sparsity, the performance of their DQN agent would go down in proportion to the environment's sparsity. This did not occur for other agents. This seemed to suggest something different for the DQN agent and when we looked at DQN's hyperparameters we realised that it had a fixed $\epsilon$ schedule. While that may be desirable in some situations, we felt it hurt DQN's performance because it was not allowed to explore enough early on nor exploit what it learnt fully later. When we use regular structured environments, the agent performances are freed of the "noise" that is present due to irregular transition functions and this makes it easy to see high-level trends.

In the Ray version we used, we observed that the noisy nets (Fortunato et al., 2018) implementation was broken on the toy environments and then we observed the same on more complex benchmarks. This makes it easy to debug if something major is broken.

## F  Sample states used for Representation Learning

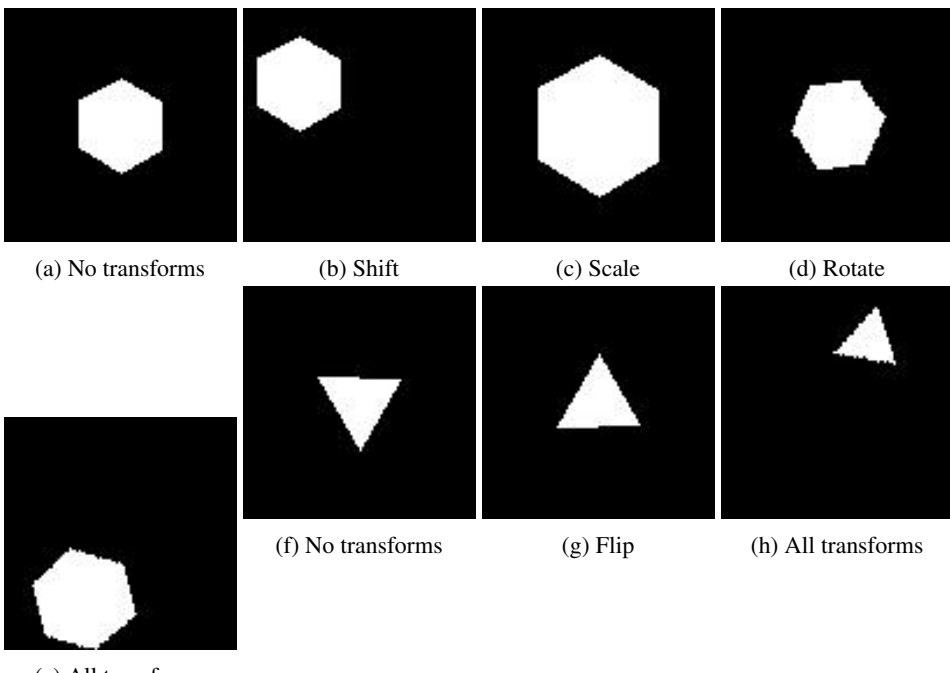

(a) No transforms      (b) Shift      (c) Scale      (d) Rotate

(f) No transforms      (g) Flip      (h) All transforms

(e) All transforms

Figure 8: When using the meta-feature *representation learning* in discrete environments, each categorical state corresponds to an image of a polygon (if the states were numbered beginning from 0, each state $n$ corresponds to a polygon with $n + 3$ sides). Various transforms can be applied to the polygons randomly at each time step. Samples shown correspond to states 3 and 0

# G    MORE EXPERIMENTS AND ADDITIONAL REWARD PLOTS

We continue with the experiments and results from the main paper here.

## G.1    RESULTS FOR VARYING TRANSITION AND REWARD NOISES

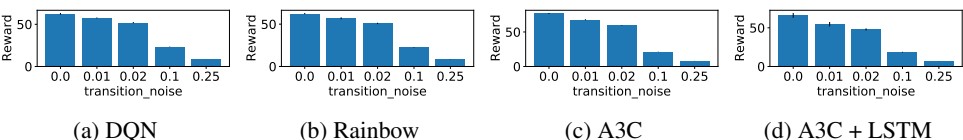

(a) DQN      (b) Rainbow      (c) A3C      (d) A3C + LSTM

Figure 9: Mean episodic reward at the end of training for the different algorithms **when varying transition noise**. Error bars represent 1 standard deviation. Note the different reward scales.

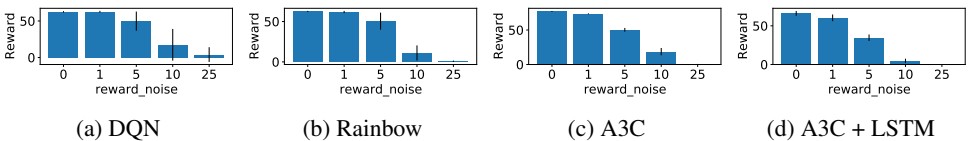

(a) DQN      (b) Rainbow      (c) A3C      (d) A3C + LSTM

Figure 10: Mean episodic reward at the end of training for the different algorithms **when varying reward noise**. Error bars represent 1 standard deviation. Note the different reward scales.

We see a similar trend during *training*, as for delays and sequences, when we vary the *transition noise* in Figure 9 and the *reward noise* in Figure 10. Performance degrades gradually as more and more noise is injected. It is interesting that, during training, all the algorithms seem to be more sensitive to noise in the transition dynamics compared to the reward dynamics: transition noise values as low as 0.02 lead to a clear handicap in learning while for the reward dynamics (with the *reward scale* being 1.0) reward noise standard deviation of $\sigma_{r\_n} = 1$ still resulted in learning progress.



(a) DQN      (b) Rainbow      (c) A3C      (d) A3C + LSTM

Figure 11: Mean episodic reward for evaluation rollouts (max 100 timesteps) at the end of training for the different algorithms **when varying transition noise**. Error bars represent 1 standard deviation.

Interestingly, when we plot the *evaluation* performances[3] in Figures 11 and 12, we see, on comparing with the training plots, that the training performance of the algorithms is more sensitive to noise in the transition dynamics (Figure 9) than the eventual evaluation performance is (Figure 11). While it is obvious that the mean episodic reward during training would be perturbed when noise is injected into the *reward* function, it is non-trivial that injecting noise into the *transition* function still leads to good learning (as displayed in the evaluation rollout plots). An additional seeming anomaly is that the evaluation rollouts for A3C variants especially (and DQN to a small extent), suggest that it performs *better* in the presence of transition noise. This might indicate that A3C in the presence of no transition noise does not explore enough (as was also conjectured in the unexpected results for varying the sparsity meta-feature) and is actually helped when transition noise is present during training.

As we mentioned earlier, one of the advantages of our platform is that it allows us to introduce all the hardness dimensions on the same base environment at the same time. This is helpful to understand interaction effects between them. We plot the most interesting interaction effects in Figure 13 where we varied both transition and reward noise over respective grid values. This plot shows that our

---

[3]Here, for evaluation, and not for training because training is *in* the noisy environment, we evaluated in the corresponding environment without noise to assess how well the *true* learning is proceeding.

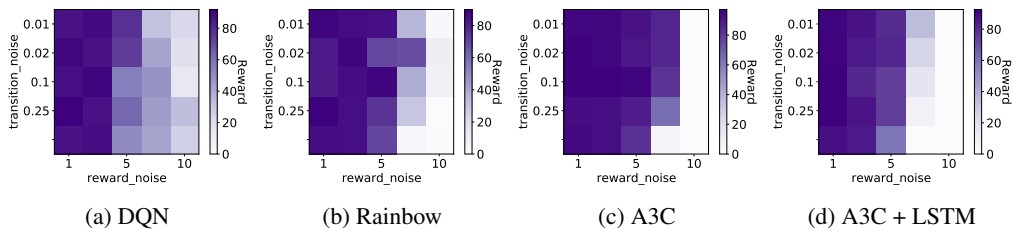

Figure 12: Mean episodic reward for evaluation rollouts (max 100 timesteps) at the end of training for the different algorithms **when varying reward noise**. Error bars represent 1 standard deviation. Note the different reward scales.

observation, that transition noise helps A3C out during *evaluation*, is only clearly valid when the reward noise is not so high ($\sigma_{r\_n} <= 1$) as to disrupt training. The corresponding heatmap plot for training when varying the noises and additional ones for jointly varying delay and rewardable sequence length are present in the Appendix (Figures 37 - 39).

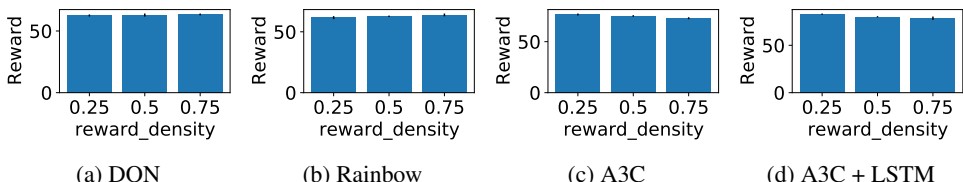

Figure 13: Mean episodic reward for evaluation rollouts (max 100 timesteps) at the end of training for the different algorithms **when varying transition and reward noise**.

### G.2 RESULTS FOR VARYING REWARD SPARSITY

Figure 14: Mean episodic reward at the end of training for the algorithms **when varying reward sparsity**. Error bars represent 1 standard deviation. Note the different reward scales.

Figure 14 shows the results of controlling the meta-feature *sparsity* in the environment. The DQN variants were able to learn the important rewarding states in the environment even when these were sparse while the behaviour of A3C was unexpected. One explanation could be that A3C's exploration was not very good, in which case increasing reward density would help as in Figure 14c. But adding in an LSTM to the A3C agent seems to show the opposite trend (Figure 14d) as increasing reward density leads to worsening performance. This could indicate that having a greater density of rewarding states makes it harder for the LSTM to remember one state to stick to. This behaviour of A3C warrants more investigation in the future.

We have observed A3C is more variant in general than its DQN counterparts and this should be expected as it launches and collects data from several instances of the same environment which induces more variance.

The $make\_denser$ configuration option, makes learning smoother and less variant across different runs of an algorithm, as can be seen in Figure 15a for DQN when compared to Figure 17 for corresponding sequence lengths. To evaluate the *true* learning of algorithms, we turn off the $make\_denser$ option in the evaluation rollouts. The learning curves for these can be seen for DQN in Figure 15b. The agent still does not perform as well as might be expected when making the reward signal denser during training. This is probably due to the sequence lengths still *violating* the complete observability

assumption made by the algorithm. The plots for learning curves for the remaining algorithms are present in Figures 70-75. The plots for final mean reward during training and evaluation are given in Figures 14 and 21.

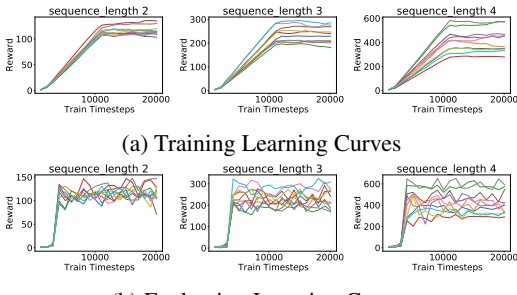

(a) Training Learning Curves

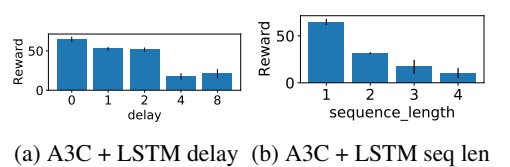

(b) Evaluation Learning Curves

Figure 15: Learning curves for DQN **when *make_denser* is *True* for rewardable sequences**. Please note the different Y-axis scales and the fact that with longer rewardable sequences, a greater number of seeds do not learn anything for the evaluation rollouts (reward $\approx 0$).

## G.3 FURTHER RESULTS FOR VARYING REWARD DELAYS AND SEQUENCES

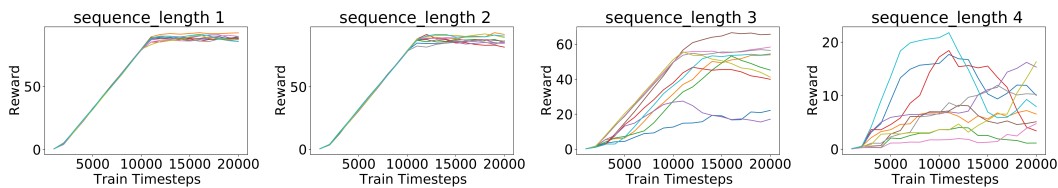

(a) A3C + LSTM delay  (b) A3C + LSTM seq len

Figure 16: Mean episodic reward at the end of training for different agents for varying **delays** (top) and **sequence lengths** (bottom). Error bars represent 1 standard deviation. Note the reward scales.

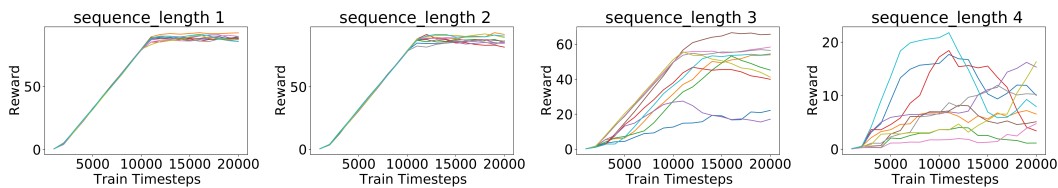

Figure 17: Train Learning Curves for 10 runs with different seeds for DQN **when varying sequence lengths**. Please note that each different colour corresponds to one of 10 seeds in each subplot.

Note that we varied delay on a logarithmic scale and sequence length on a linear one which means that this effect is more pronounced than may first appear when looking at the figures. We additionally also plot the *learning curves*, when varying sequence lengths, in Figure 17. We see how training proceeds much more smoothly and is less variant across different seeds for the vanilla environment (where the sequence length is 1) and that the variance across seeds is very large for sequence length 3.

## G.4 SELECTING TOTAL TIMESTEPS FOR RUNS

We ran the experiments and plot the results for DQN variants up to 20 000 environment timesteps and the ones for A3C variants up to 150 000 time steps since A3C took longer[4] to learn as can be seen in Figure 18. We refrain from fixing a single number of timesteps for our environments (as, e.g., bsuite does), since the study of different trends for different families of algorithms will require different numbers of timesteps. Policy gradient methods such as A3C are slower in general compared to value-based approaches such as DQN. Throughout, we always run 10 seeds of all algorithms to

---

[4]In terms of environment steps. Wallclock time used was still similar.

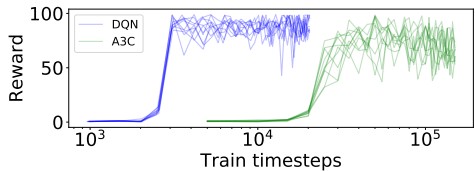

Figure 18: Evaluation rollouts (limited to 100 timesteps per episode) for DQN and A3C in the vanilla environment which shows that DQN learns faster than A3C in terms of the number of timesteps.

obtain reliable results. We repeated many of our experiments with an independent set of 10 seeds and obtained the same qualitative results.

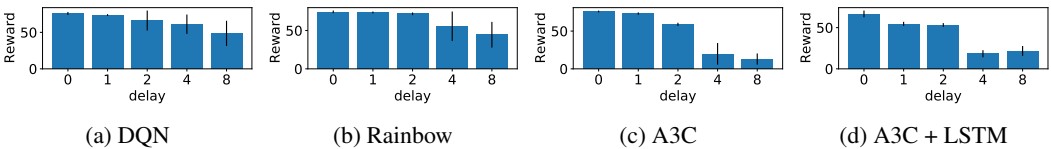

| (a) DQN | (b) Rainbow | (c) A3C | (d) A3C + LSTM |

Figure 19: Mean episodic reward for evaluation rollouts (limited to 100 timesteps) at the end of training for the different algorithms **when varying delay**. Error bars represent 1 standard deviation.

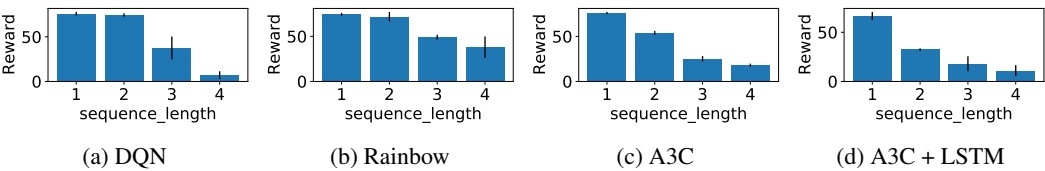

| (a) DQN | (b) Rainbow | (c) A3C | (d) A3C + LSTM |

Figure 20: Mean episodic reward for evaluation rollouts (limited to 100 timesteps) at the end of training for the different algorithms **when varying sequence lengths**. Error bars represent 1 standard deviation.

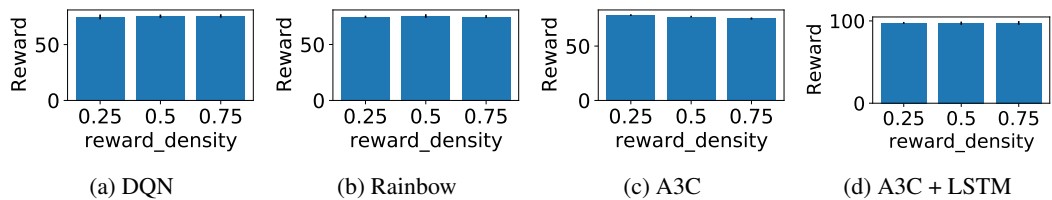

| (a) DQN | (b) Rainbow | (c) A3C | (d) A3C + LSTM |

Figure 21: Mean episodic reward for evaluation rollouts (limited to 100 timesteps) at the end of training for the different algorithms **when varying reward sparsity**. Error bars represent 1 standard deviation.

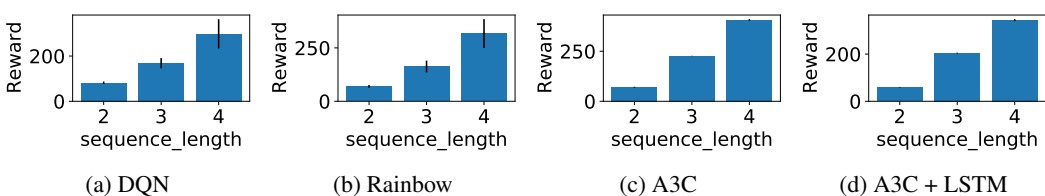

| (a) DQN | (b) Rainbow | (c) A3C | (d) A3C + LSTM |

Figure 22: Mean episodic reward at the end of training for the different algorithms **when *make_denser* is *True* for rewardable sequences**. Error bars represent 1 standard deviation.

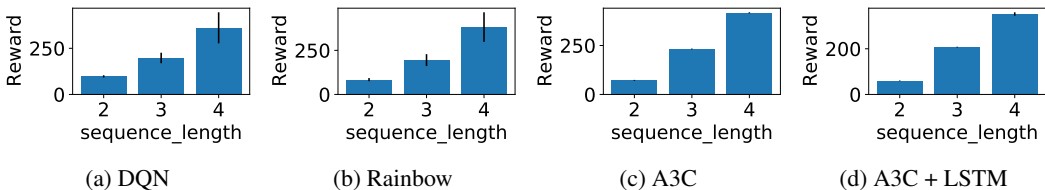

(a) DQN      (b) Rainbow      (c) A3C      (d) A3C + LSTM

Figure 23: Mean episodic reward for evaluation rollouts (limited to 100 timesteps) at the end of training for the different algorithms **when *make_denser* is *True* for rewardable sequences**. Error bars represent 1 standard deviation.

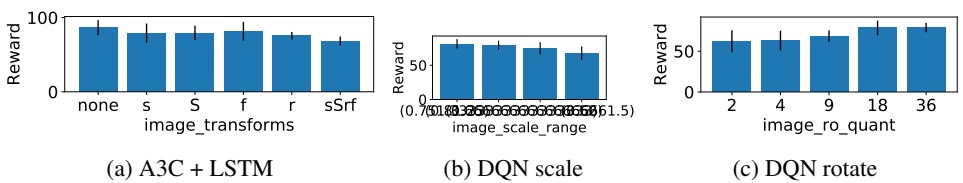

(a) A3C + LSTM      (b) DQN scale      (c) DQN rotate

Figure 24: Mean episodic reward at the end of training for the different algorithms **when varying representation learning**. 's' represents *shift*, 'S' represents *scale*, 'f' represents *flip* and 'r' represents *rotate* in the labels in the first subfigure. *scale_range* represents *scaling* ranges in the second subfigure. *image_ro_quant* is represents quantisation of the *rotation*s in the third subfigure. Error bars represent 1 standard deviation.

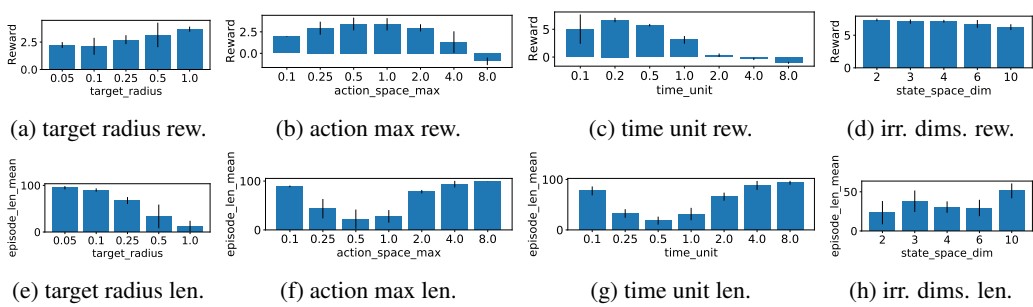

(a) target radius rew.    (b) action max rew.    (c) time unit rew.    (d) irr. dims. rew.

(e) target radius len.    (f) action max len.    (g) time unit len.    (h) irr. dims. len.

Figure 25: Mean episodic reward (above) and lengths (below) for TD3 at the end of training. Error bars represent 1 standard deviation.

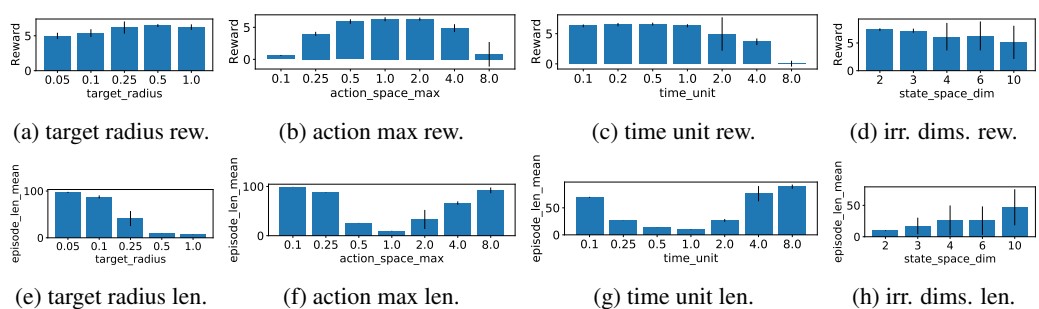

(a) target radius rew.    (b) action max rew.    (c) time unit rew.    (d) irr. dims. rew.

(e) target radius len.    (f) action max len.    (g) time unit len.    (h) irr. dims. len.

Figure 26: Mean episodic reward (above) and lengths (below) for SAC at the end of training. Error bars represent 1 standard deviation.

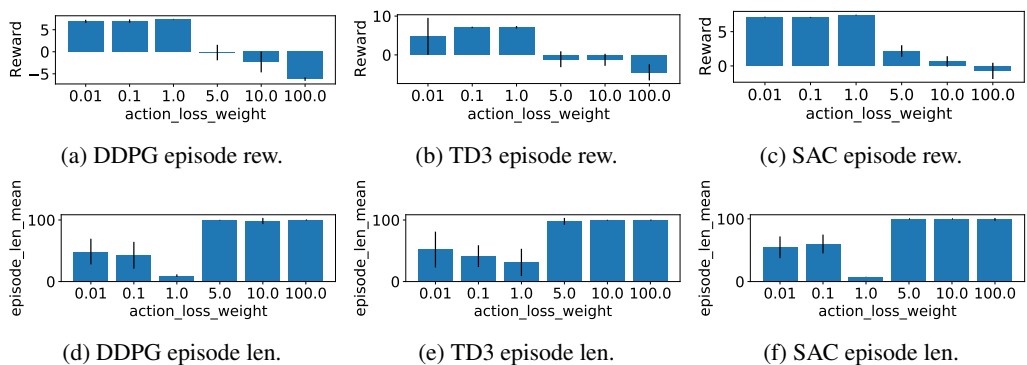

(a) DDPG episode rew.  (b) TD3 episode rew.  (c) SAC episode rew.

(d) DDPG episode len.  (e) TD3 episode len.  (f) SAC episode len.

Figure 27: Mean episodic reward (above) and lengths (below) at the end of training for evaluation rollouts for DDPG, TD3 and SAC when varying **action_loss_weight**. Error bars represent 1 standard deviation.

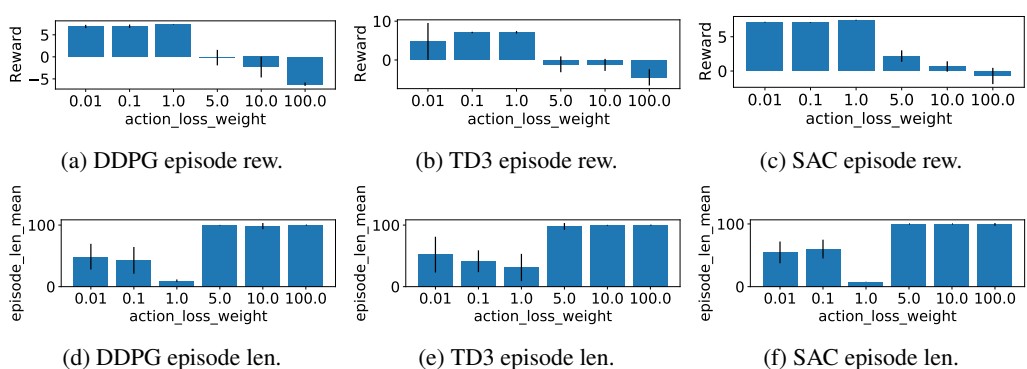

(a) DDPG episode rew.  (b) TD3 episode rew.  (c) SAC episode rew.

(d) DDPG episode len.  (e) TD3 episode len.  (f) SAC episode len.

Figure 28: Mean episodic reward (above) and lengths (below) at the end of training for evaluation rollouts for DDPG, TD3 and SAC when varying **action_loss_weight**. Error bars represent 1 standard deviation.

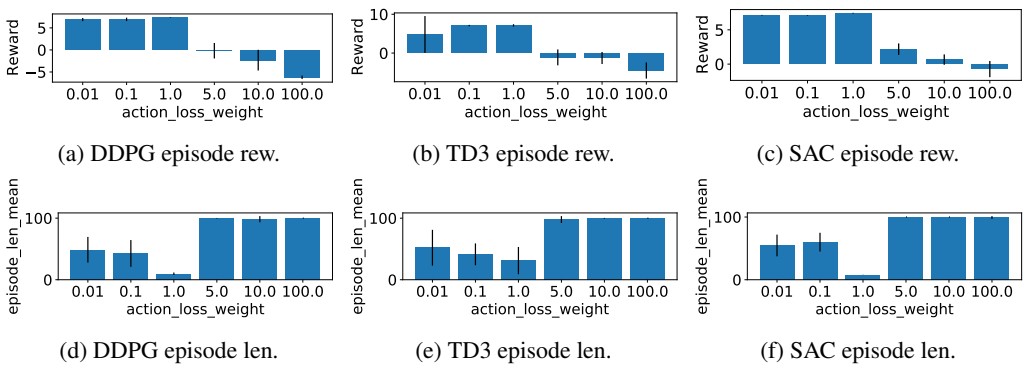

(a) DDPG episode rew.  (b) TD3 episode rew.  (c) SAC episode rew.

(d) DDPG episode len.  (e) TD3 episode len.  (f) SAC episode len.

Figure 29: Mean episodic reward (above) and lengths (below) at the end of training for evaluation rollouts for DDPG, TD3 and SAC when varying **action_loss_weight**. Error bars represent 1 standard deviation.

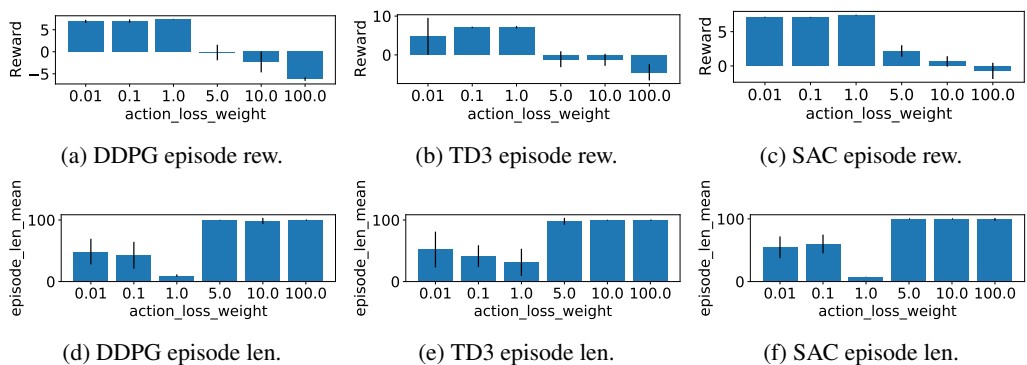

(a) DDPG episode rew.  (b) TD3 episode rew.  (c) SAC episode rew.

(d) DDPG episode len.  (e) TD3 episode len.  (f) SAC episode len.

Figure 30: Mean episodic reward (above) and lengths (below) at the end of training for evaluation rollouts for DDPG, TD3 and SAC when varying **action_loss_weight**. Error bars represent 1 standard deviation.

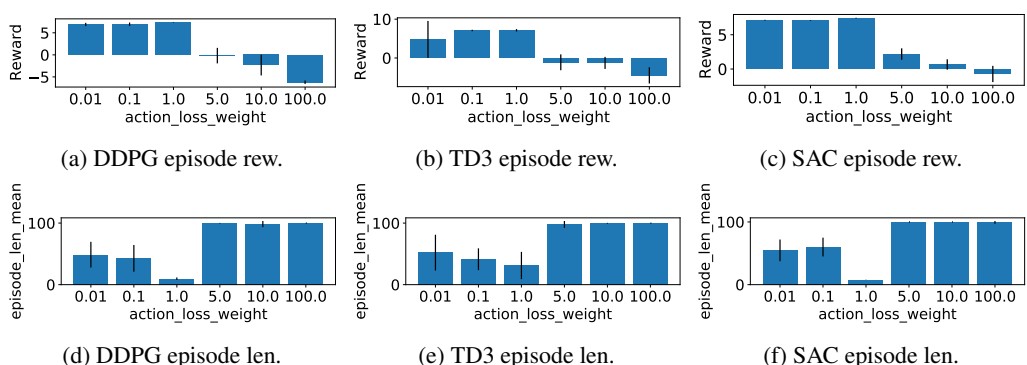

(a) DDPG episode rew.  (b) TD3 episode rew.  (c) SAC episode rew.

(d) DDPG episode len.  (e) TD3 episode len.  (f) SAC episode len.

Figure 31: Mean episodic reward (above) and lengths (below) at the end of training for evaluation rollouts for DDPG, TD3 and SAC when varying **action_loss_weight**. Error bars represent 1 standard deviation.

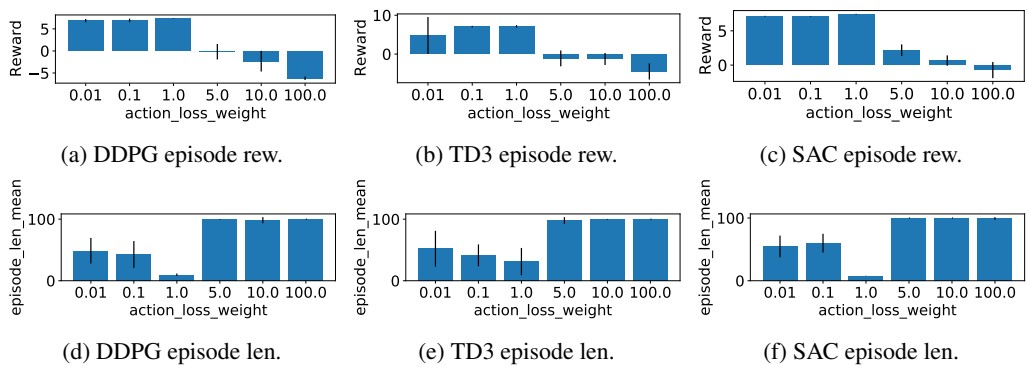

(a) DDPG episode rew.  (b) TD3 episode rew.  (c) SAC episode rew.

(d) DDPG episode len.  (e) TD3 episode len.  (f) SAC episode len.

Figure 32: Mean episodic reward (above) and lengths (below) at the end of training for evaluation rollouts for DDPG, TD3 and SAC when varying **action_loss_weight**. Error bars represent 1 standard deviation.

# H PLOTS FOR TABULAR BASELINES

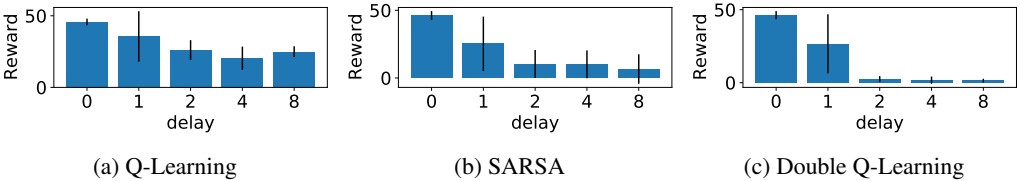

(a) Q-Learning      (b) SARSA      (c) Double Q-Learning

Figure 33: Mean episodic reward (limited to 100 timesteps) at the end of training for three different tabular baseline algorithms **when varying reward delay**. Error bars represent 1 standard deviation.

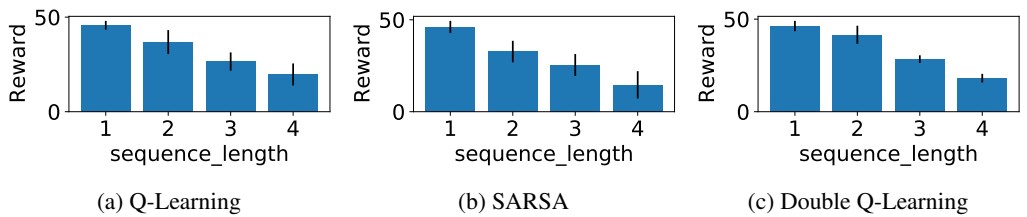

(a) Q-Learning      (b) SARSA      (c) Double Q-Learning

Figure 34: Mean episodic reward (limited to 100 timesteps) at the end of training for three different tabular baseline algorithms **when varying sequence length**. Error bars represent 1 standard deviation.

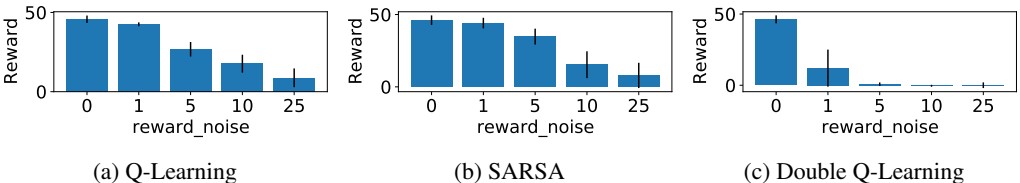

(a) Q-Learning      (b) SARSA      (c) Double Q-Learning

Figure 35: Mean episodic reward (limited to 100 timesteps) at the end of training for three different tabular baseline algorithms **when varying reward noise**. Error bars represent 1 standard deviation.

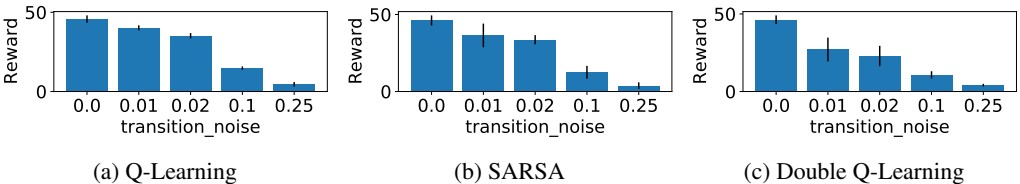

(a) Q-Learning      (b) SARSA      (c) Double Q-Learning

Figure 36: Mean episodic reward (limited to 100 timesteps) at the end of training for three different tabular baseline algorithms **when varying transition noise**. Error bars represent 1 standard deviation.

# I PLOTS FOR VARYING 2 HARDNESS DIMENSIONS TOGETHER

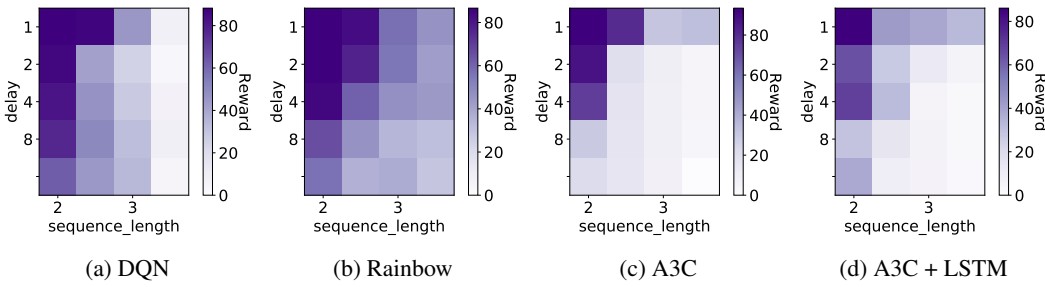

Figure 37: Mean episodic reward at the end of training for the different algorithms **when varying delay and sequence lengths**. Please note the different colorbar scales.

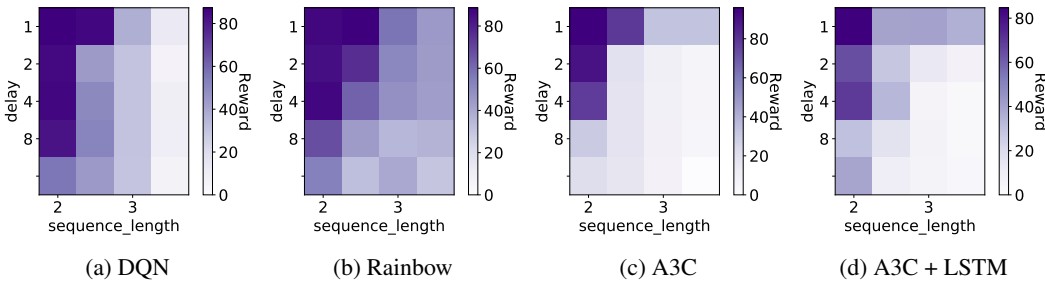

Figure 38: Mean episodic reward for evaluation rollouts (limited to 100 timesteps) at the end of training for the different algorithms **when varying delay and rewardable sequence lengths**. Please note the different colorbar scales.

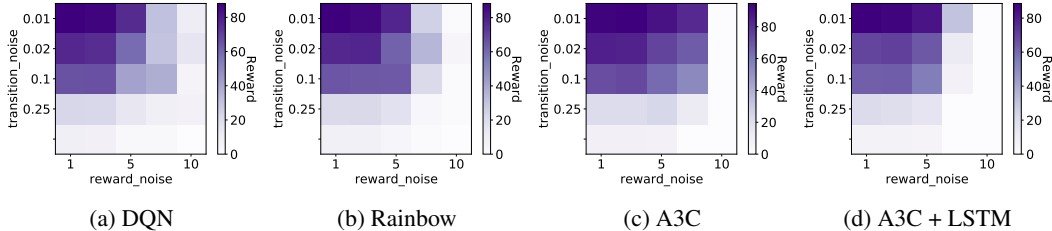

Figure 39: Mean episodic reward at the end of training for the different algorithms **when varying transition noise and reward noise**. Please note the different colorbar scales.

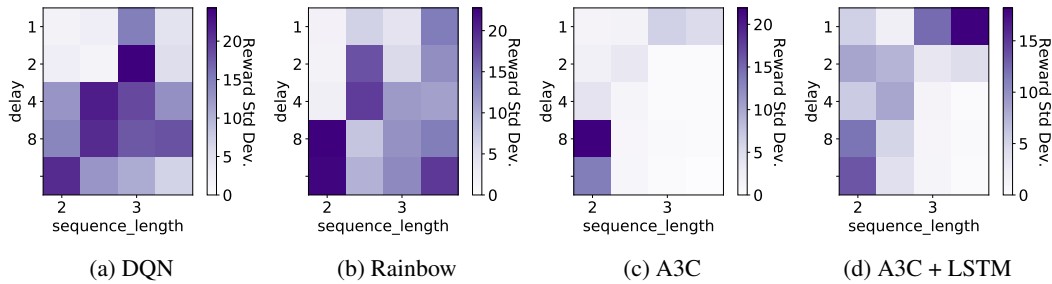

Figure 40: Standard deviation of mean episodic reward at the end of training for the different algorithms **when varying delay and sequence lengths**. Please note the different colorbar scales.

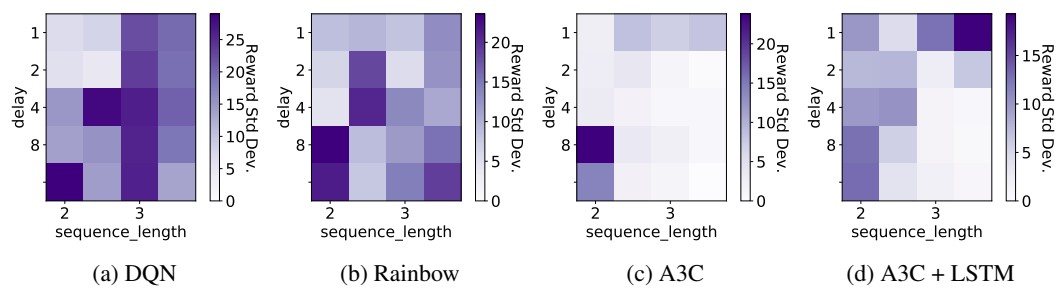

Figure 41: Standard deviation of mean episodic reward for evaluation rollouts (limited to 100 timesteps) at the end of training for the different algorithms **when varying delay and rewardable sequence lengths**. Please note the different colorbar scales.

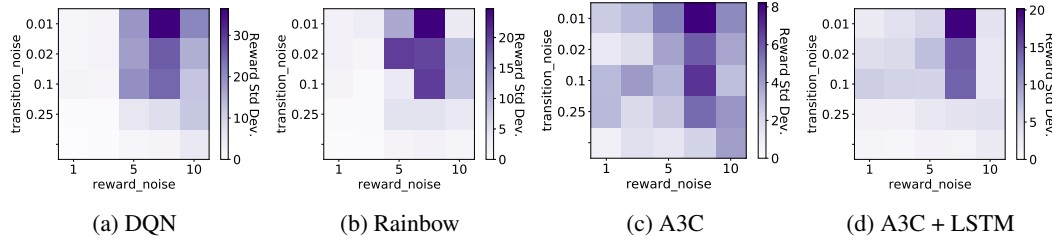

Figure 42: Standard deviation of mean episodic reward at the end of training for the different algorithms **when varying transition noise and reward noise**. Please note the different colorbar scales.

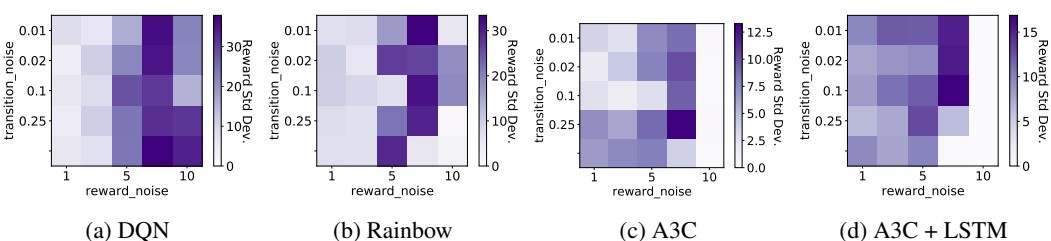

Figure 43: Standard deviation of mean episodic reward at the end of training for evaluation rollouts (limited to 100 timesteps) at the end of training for the different algorithms **when varying transition noise and reward noise**. Please note the different colorbar scales.

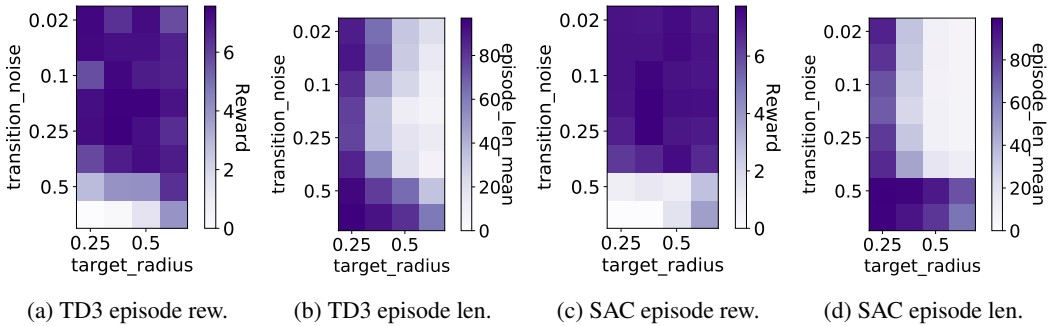

(a) TD3 episode rew.  (b) TD3 episode len.  (c) SAC episode rew.  (d) SAC episode len.

Figure 44: Mean episodic reward and lengths at the end of training for the different algorithms **when varying** $P$ **noise and** *target radius*.

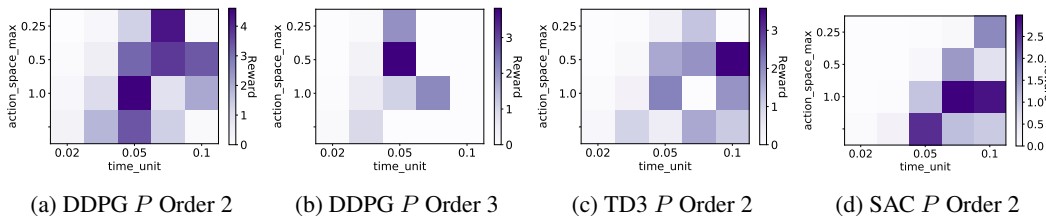

(a) DDPG $P$ Order 2  (b) DDPG $P$ Order 3  (c) TD3 $P$ Order 2  (d) SAC $P$ Order 2

Figure 45: Mean episodic reward at the end of training for the different algorithms **when varying action space max and time unit for a given** $P$ **order**.

## J    ADDITIONAL LEARNING CURVES

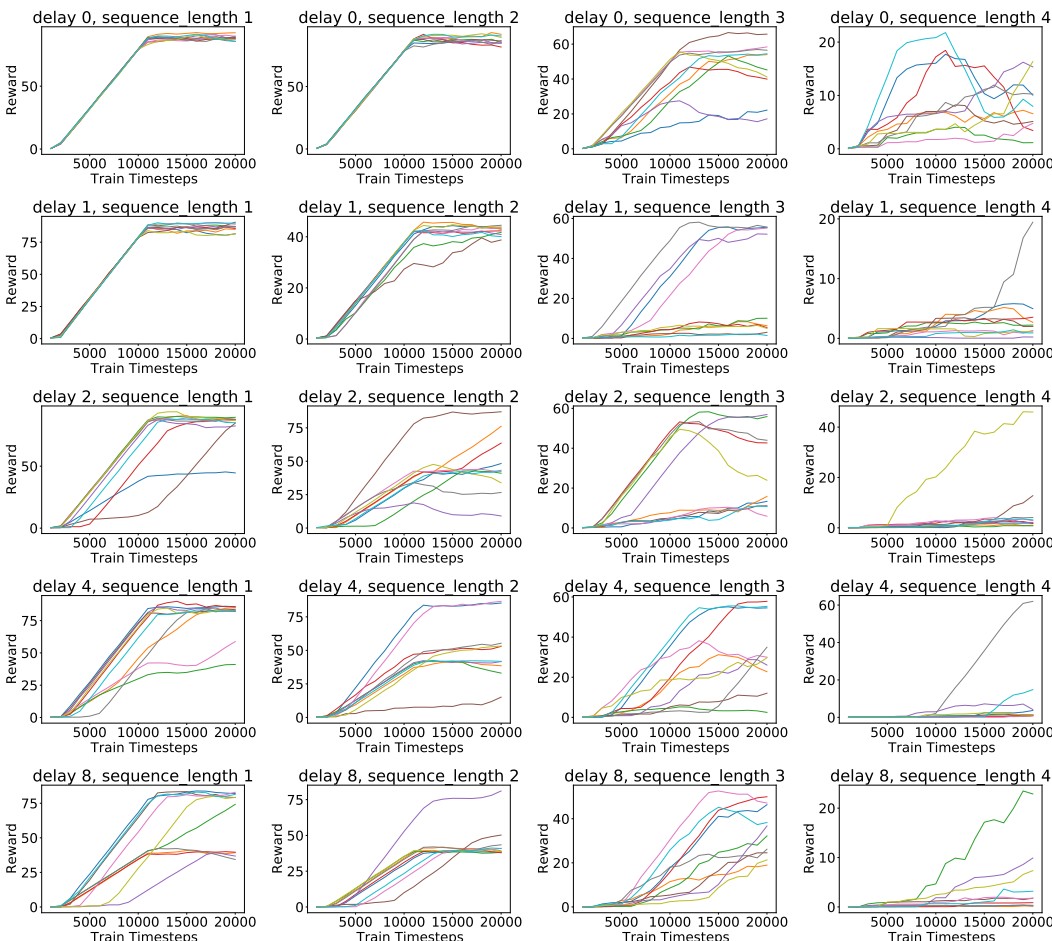

Figure 46: Training Learning Curves for DQN **when varying delay and sequence lengths**. Please note the different colorbar scales.

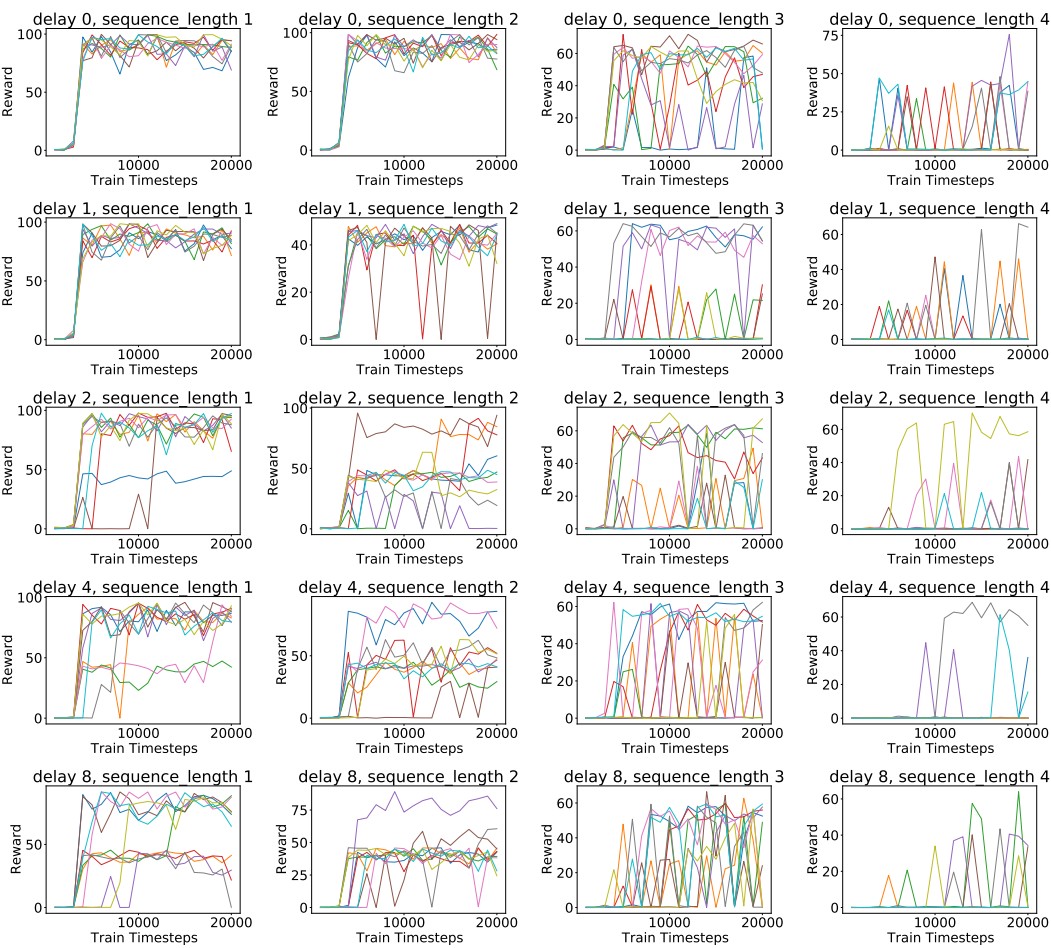

Figure 47: Evaluation Learning Curves for DQN **when varying delay and sequence lengths**. Please note the different colorbar scales.

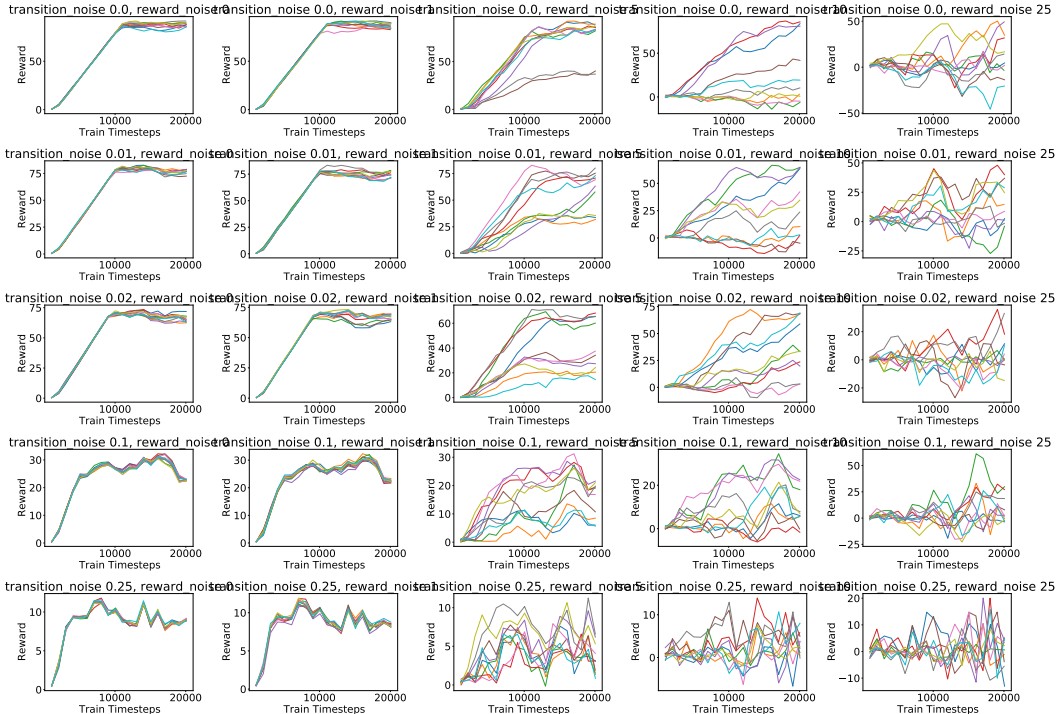

Figure 48: Training Learning Curves for DQN **when varying transition noise and reward noise**.

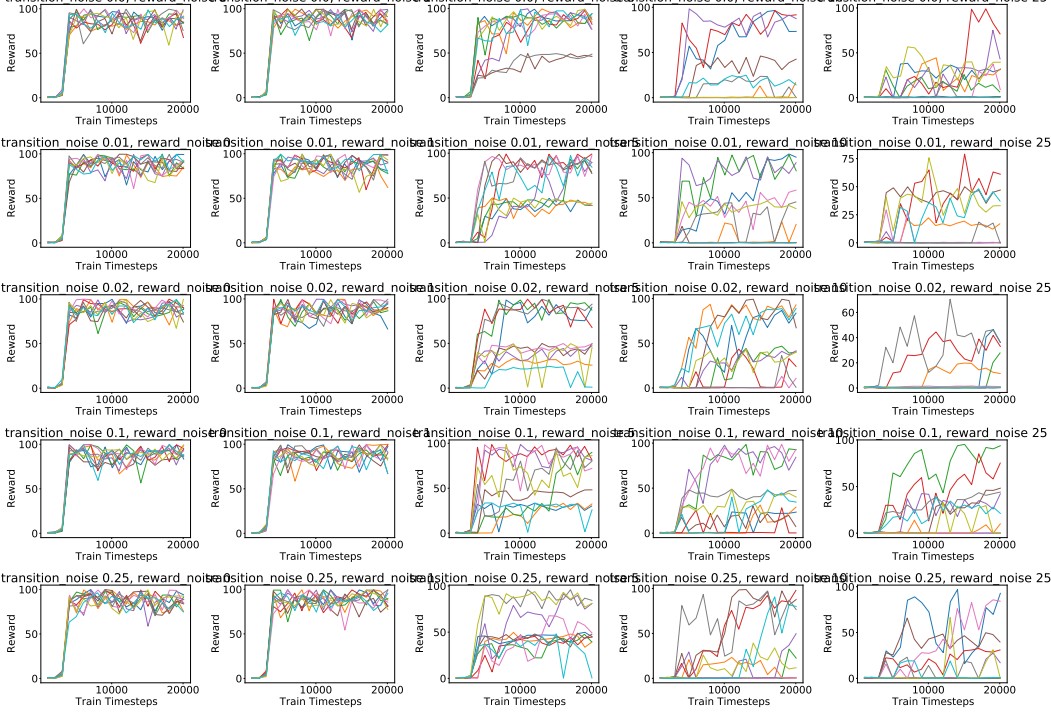

Figure 49: Evaluation Learning Curves for DQN **when varying transition noise and reward noise**. Please note the different Y-axis scales.

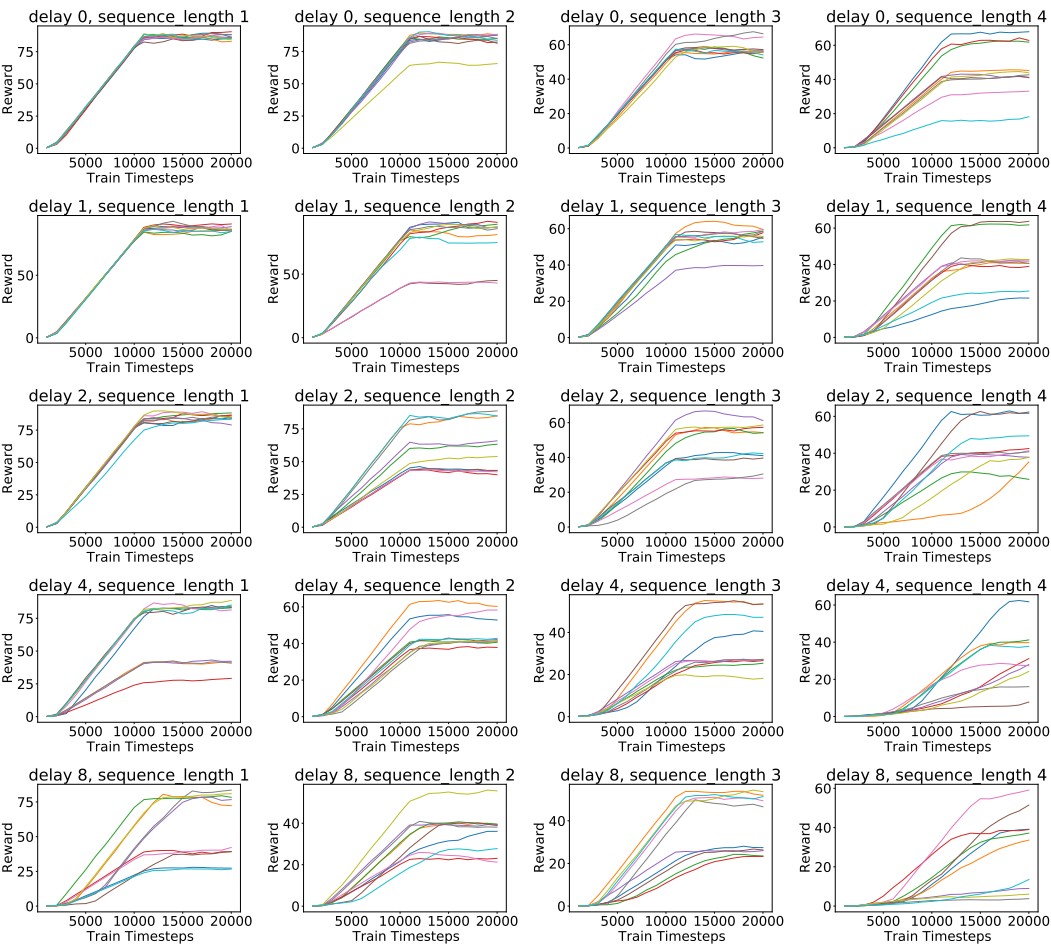

Figure 50: Training Learning Curves for Rainbow **when varying delay and sequence lengths**. Please note the different Y-axis scales.

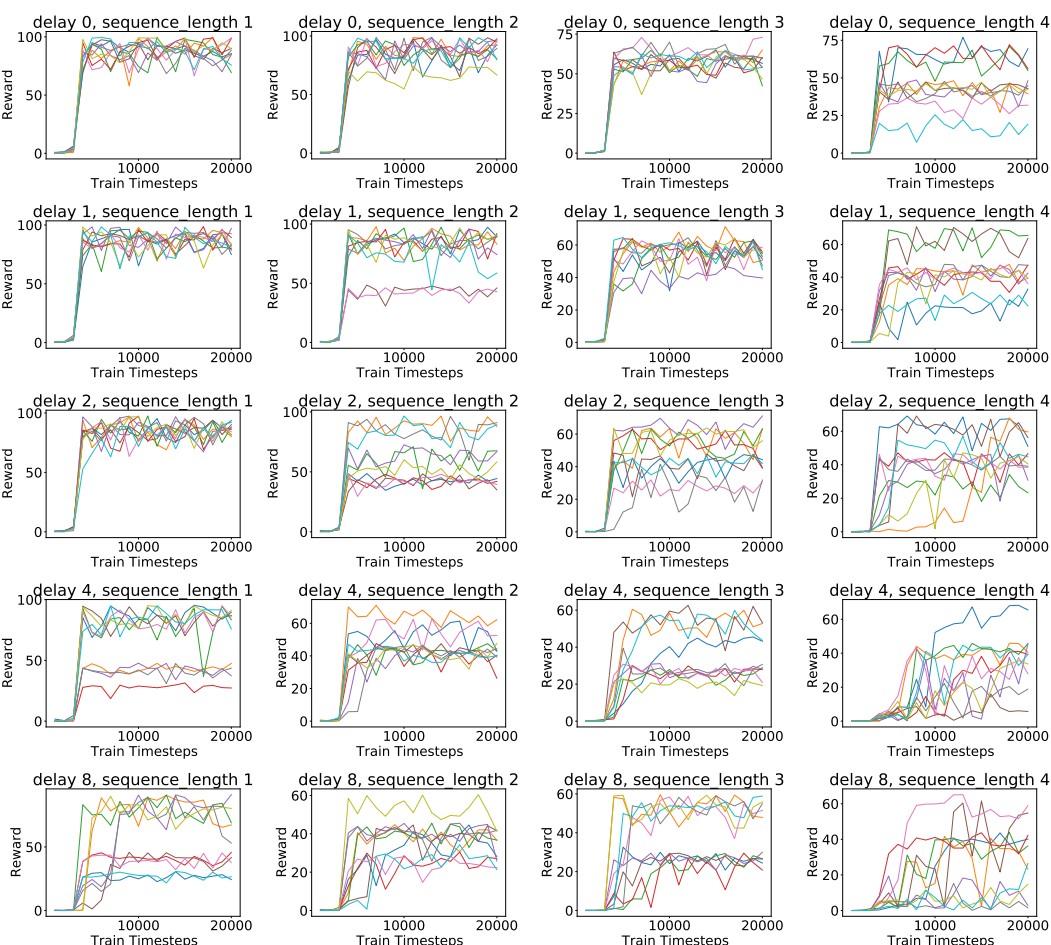

Figure 51: Evaluation Learning Curves for Rainbow **when varying delay and sequence lengths**. Please note the different Y-axis scales.

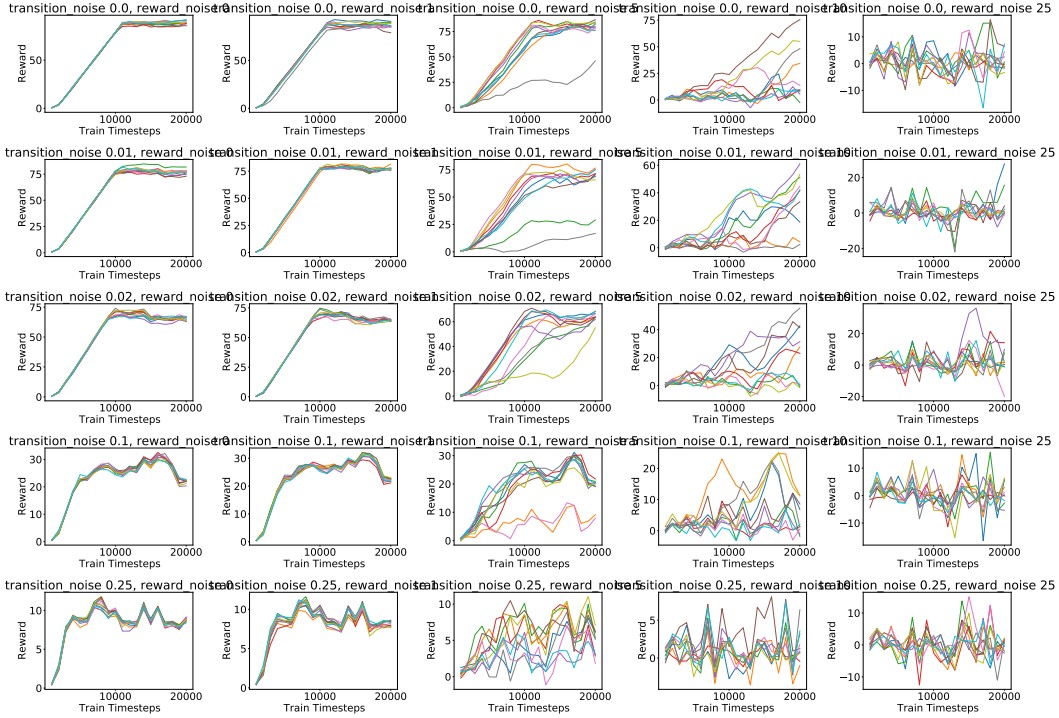

Figure 52: Training Learning Curves for Rainbow **when varying noises**. Please note the different Y-axis scales.

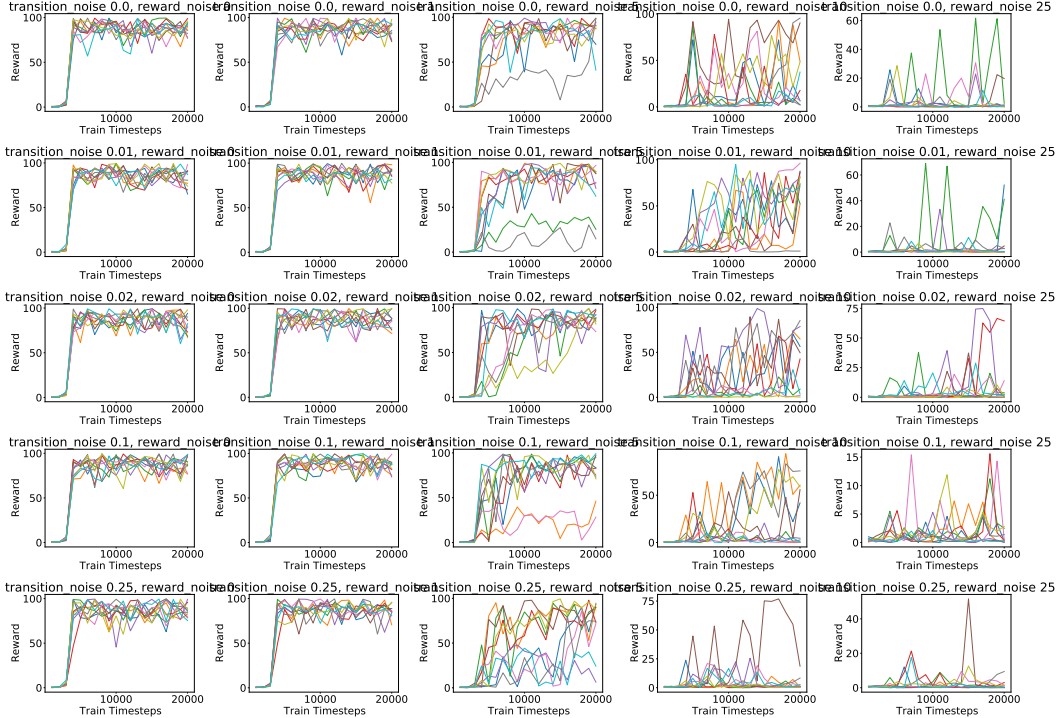

Figure 53: Evaluation Learning Curves for Rainbow **when varying noises**. Please note the different Y-axis scales.

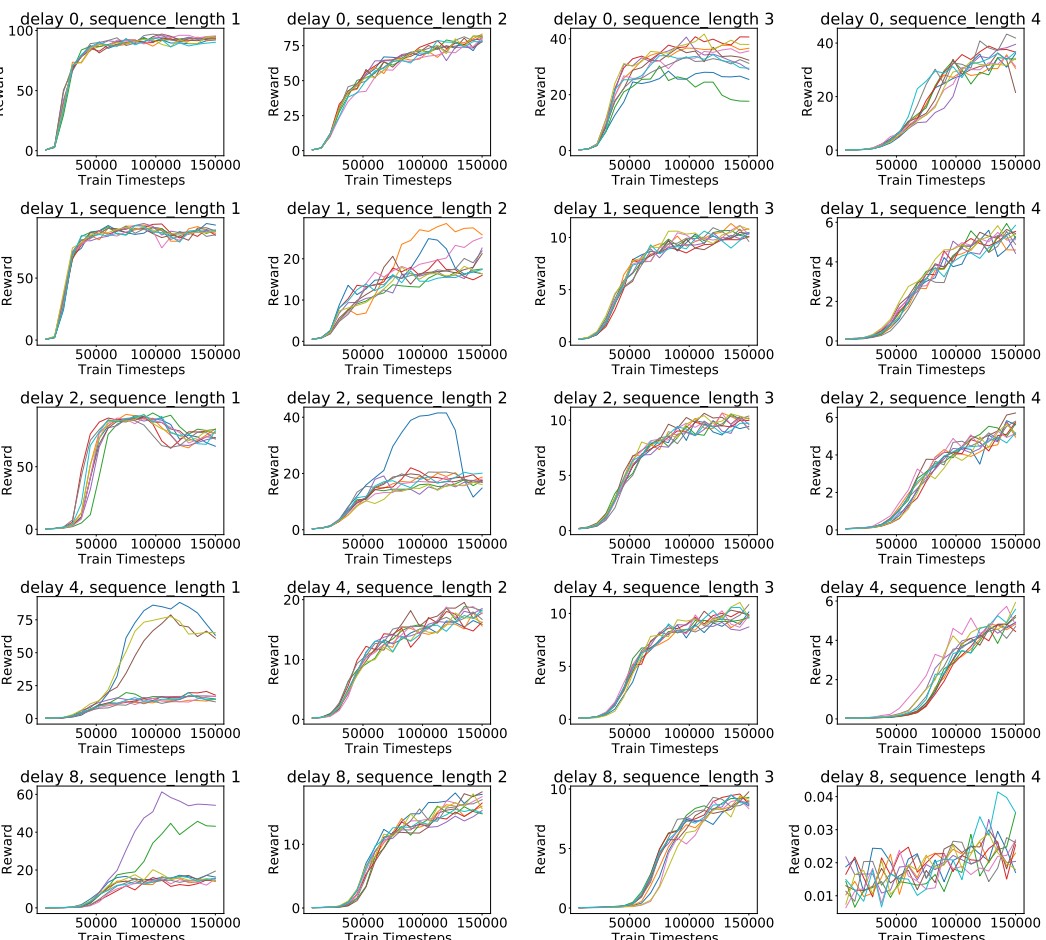

Figure 54: Training Learning Curves for A3C **when varying delay and sequence lengths**. Please note the different Y-axis scales.

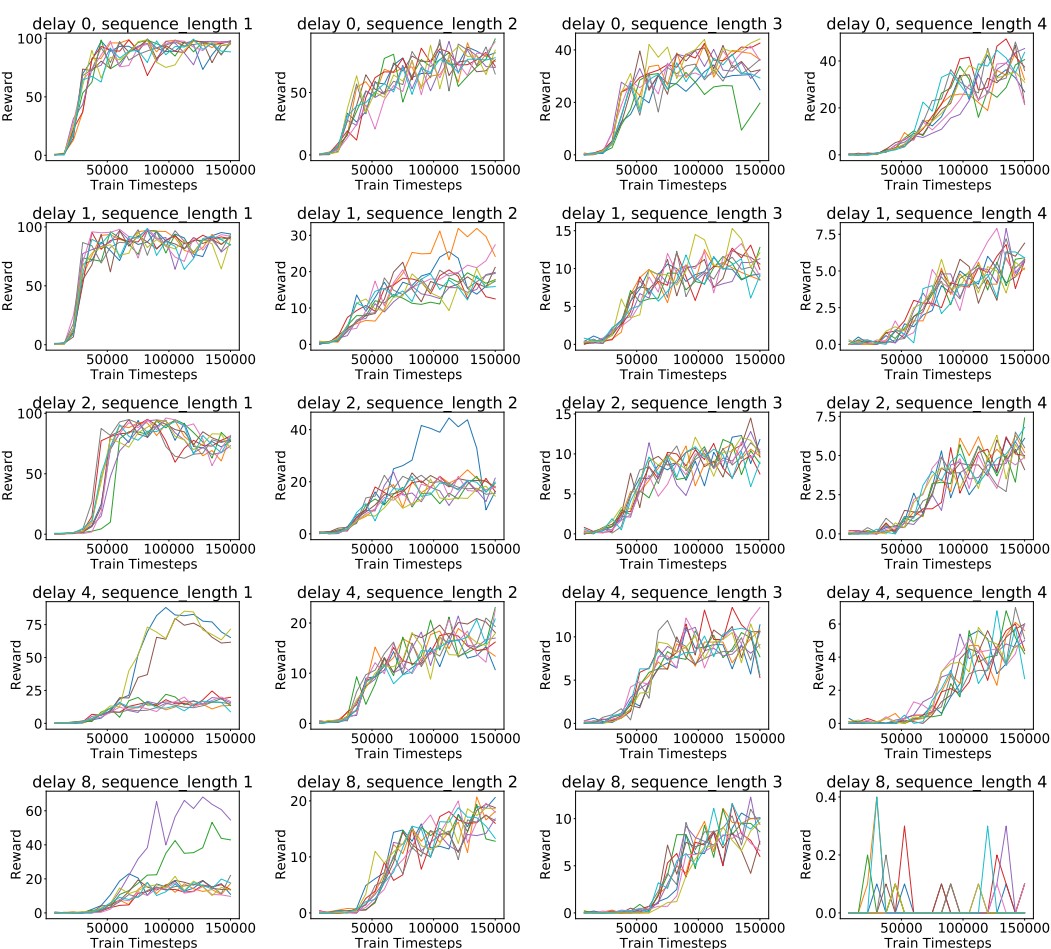

Figure 55: Evaluation Learning Curves for A3C **when varying delay and sequence lengths**. Please note the different Y-axis scales.

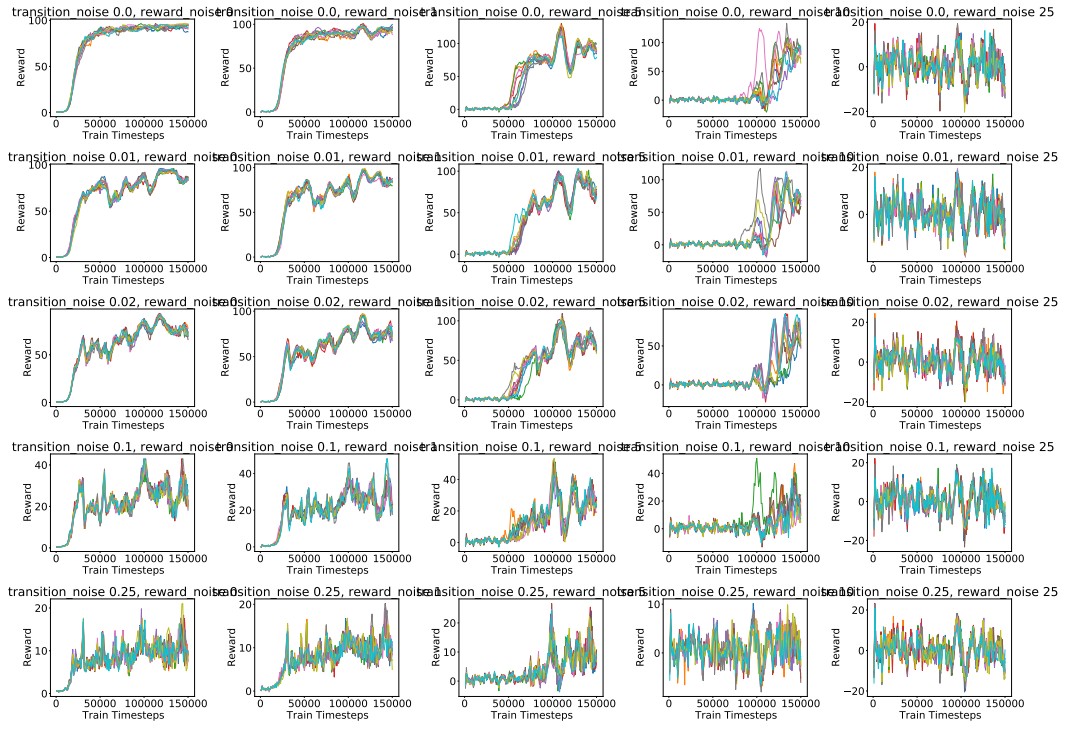

Figure 56: Training Learning Curves for A3C **when varying noises**. Please note the different Y-axis scales.

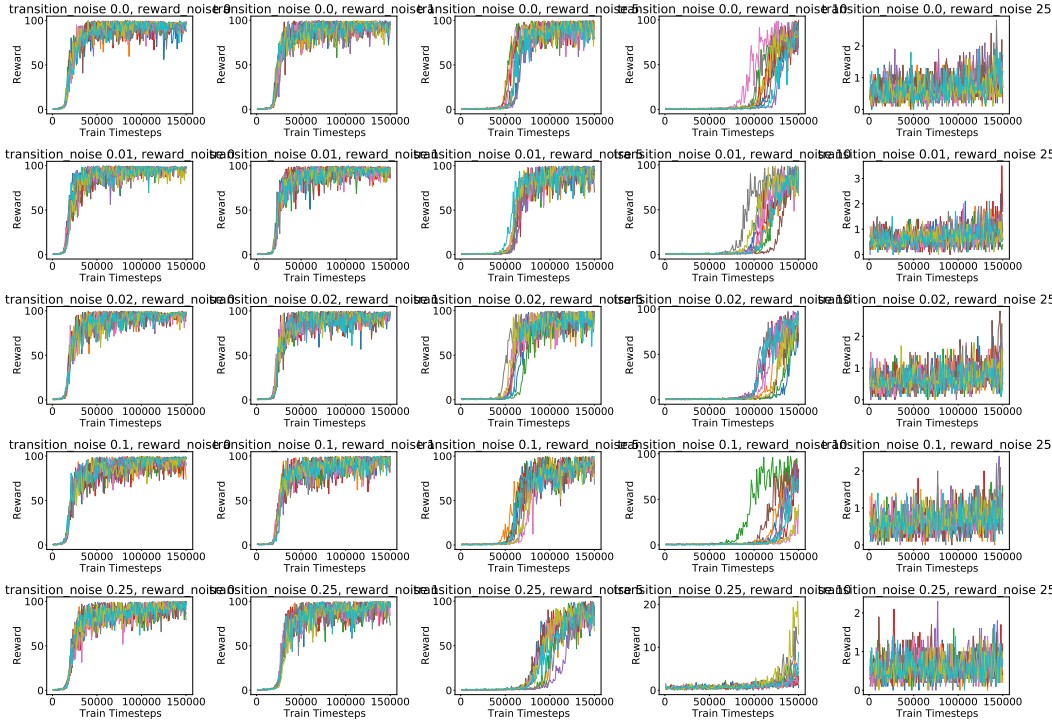

Figure 57: Evaluation Learning Curves for A3C **when varying noises**. Please note the different Y-axis scales.

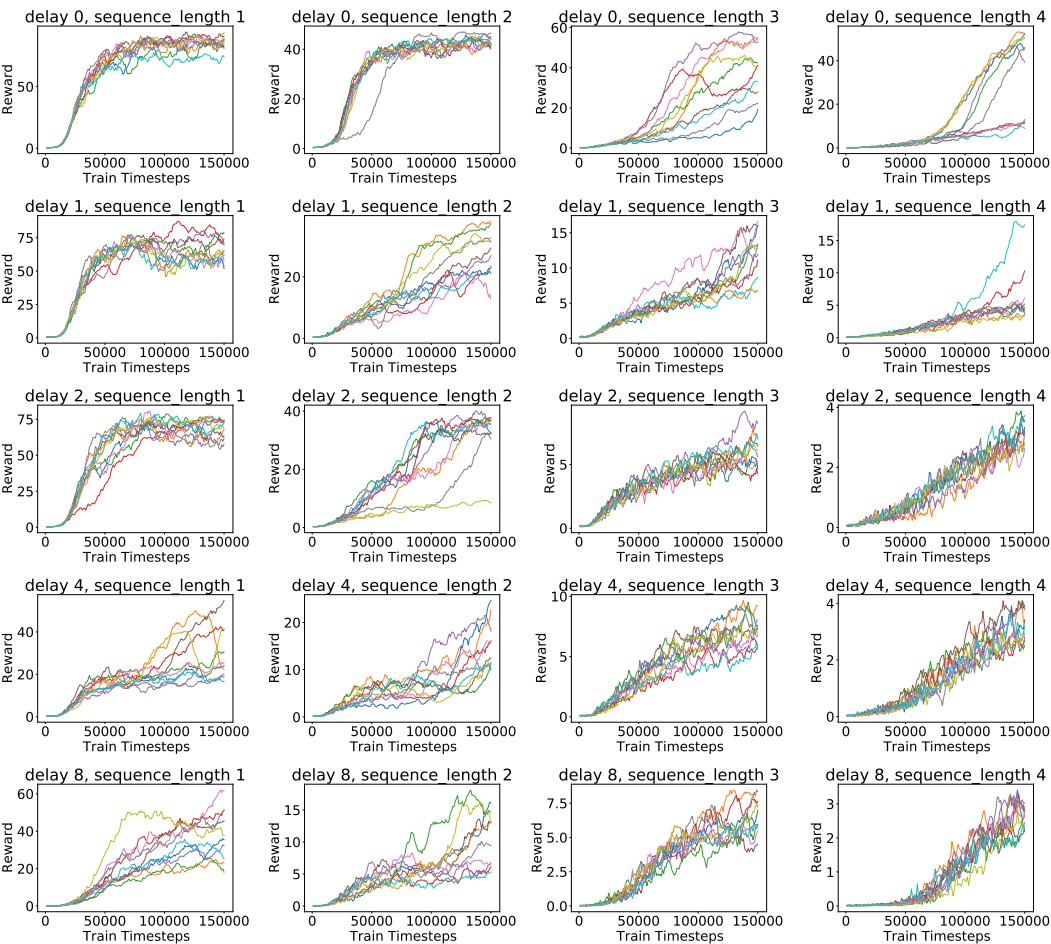

Figure 58: Training Learning Curves for A3C with LSTM **when varying delay and sequence lengths**. Please note the different Y-axis scales.

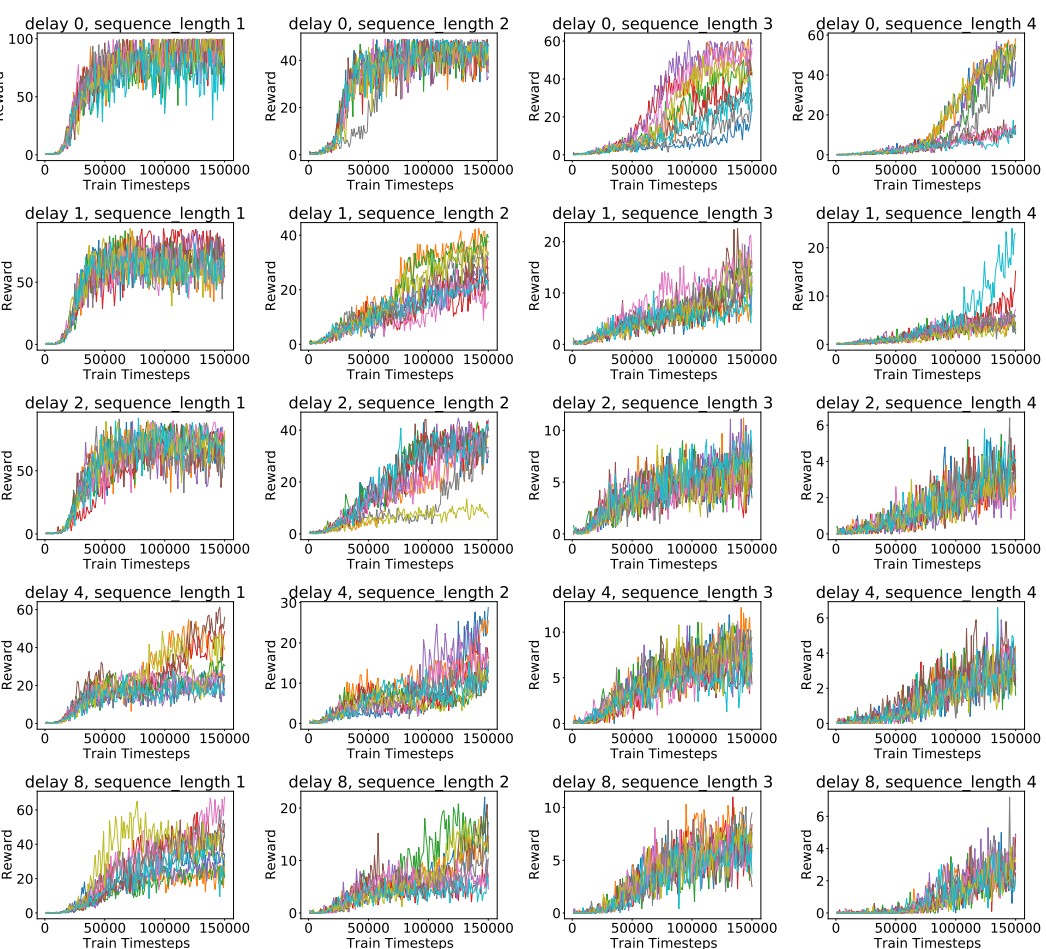

Figure 59: Evaluation Learning Curves for A3C with LSTM **when varying delay and sequence lengths**. Please note the different Y-axis scales.

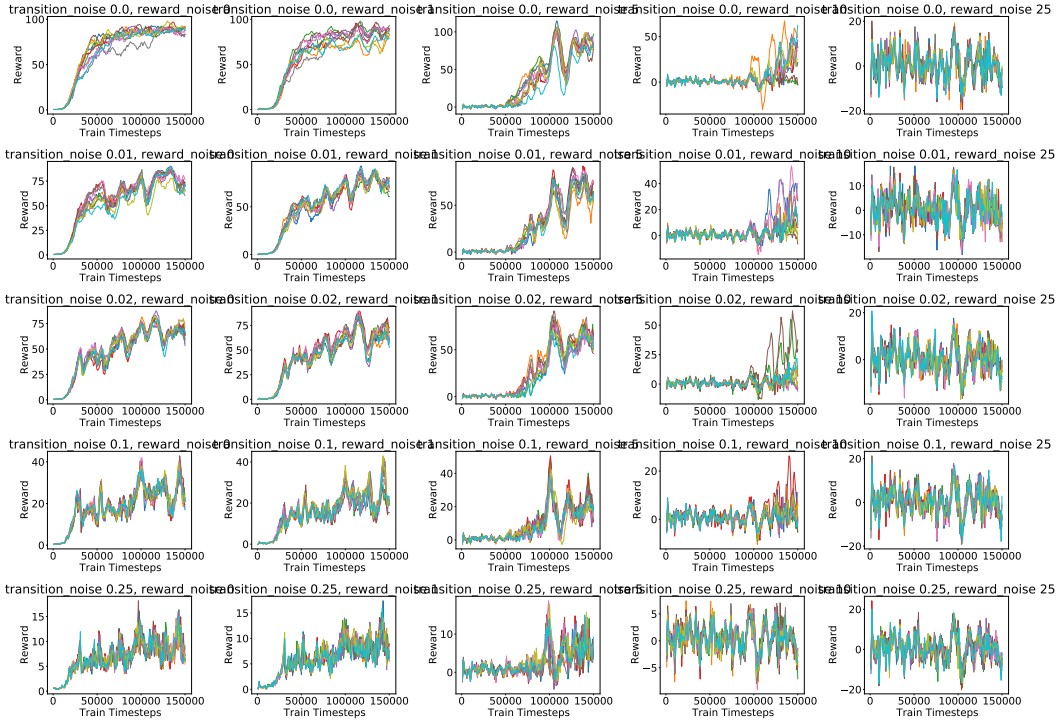

Figure 60: Training Learning Curves for A3C with LSTM **when varying noises**. Please note the different Y-axis scales.

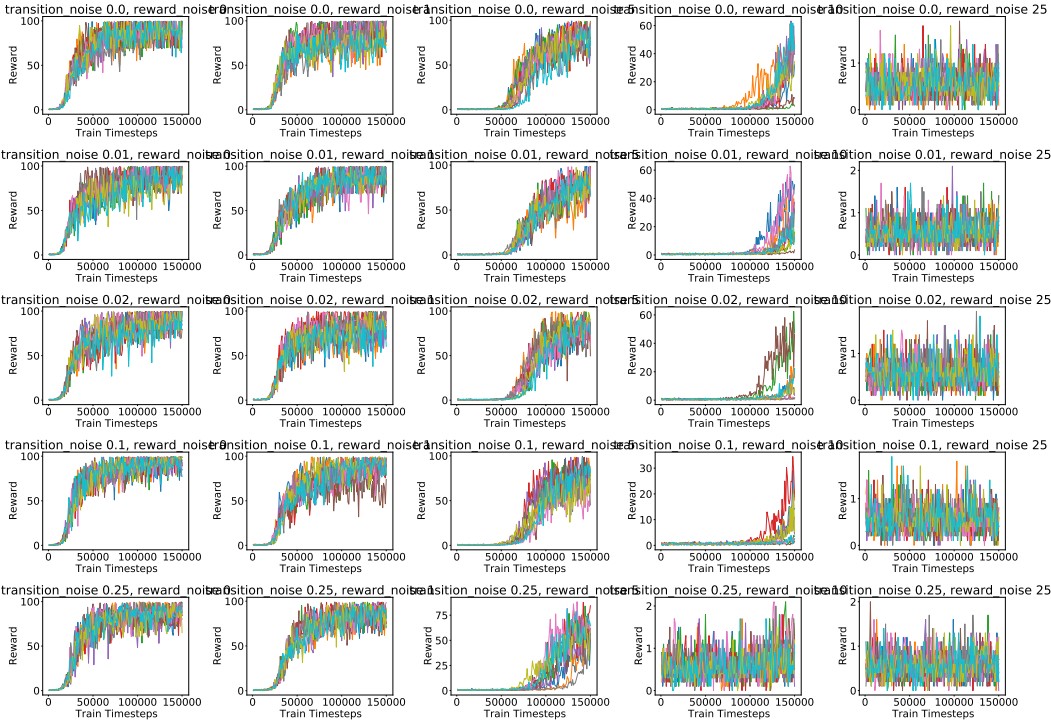

Figure 61: Evaluation Learning Curves for A3C with LSTM **when varying noises**. Please note the different Y-axis scales.

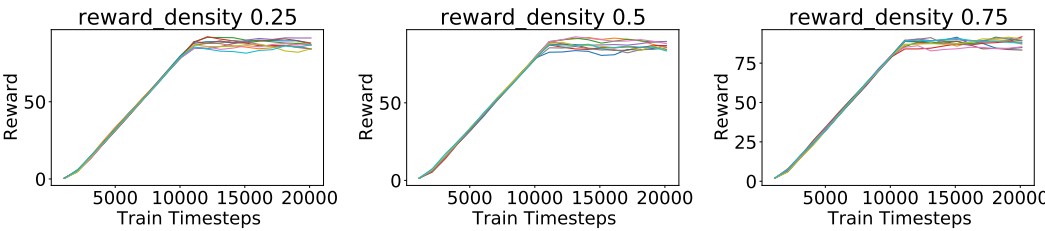

Figure 62: Training Learning Curves for DQN **when varying reward sparsity**. Please note the different Y-axis scales.

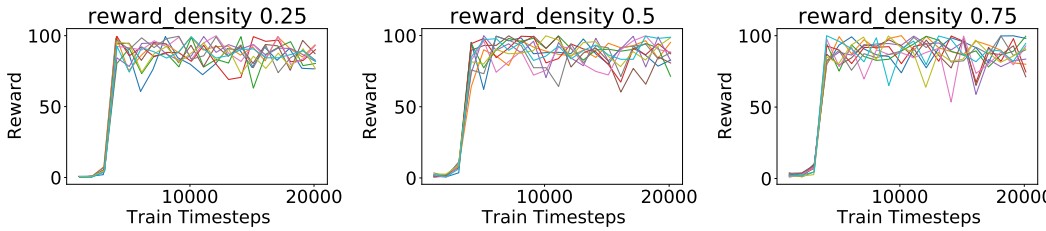

Figure 63: Evaluation Learning Curves for DQN **when varying reward sparsity**. Please note the different Y-axis scales.

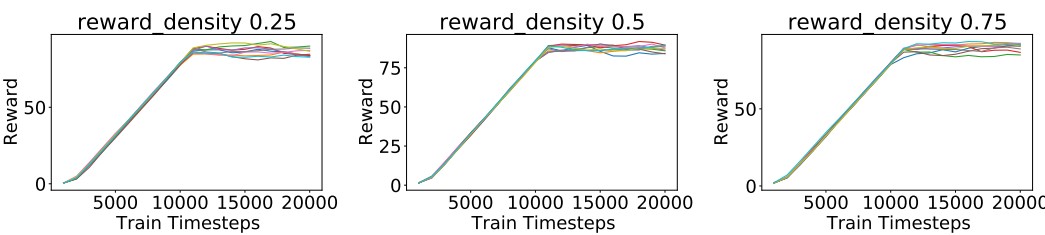

Figure 64: Training Learning Curves for Rainbow **when varying reward sparsity**. Please note the different Y-axis scales.

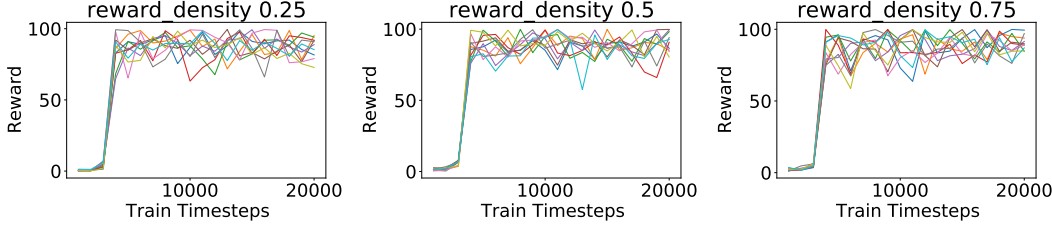

Figure 65: Evaluation Learning Curves for Rainbow **when varying reward sparsity**. Please note the different Y-axis scales.

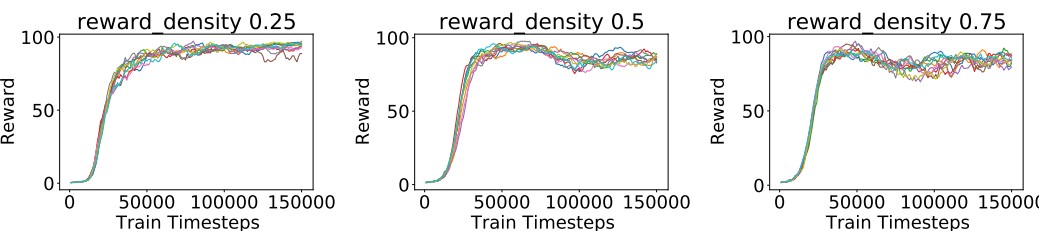

Figure 66: Training Learning Curves for A3C **when varying reward sparsity**. Please note the different Y-axis scales.

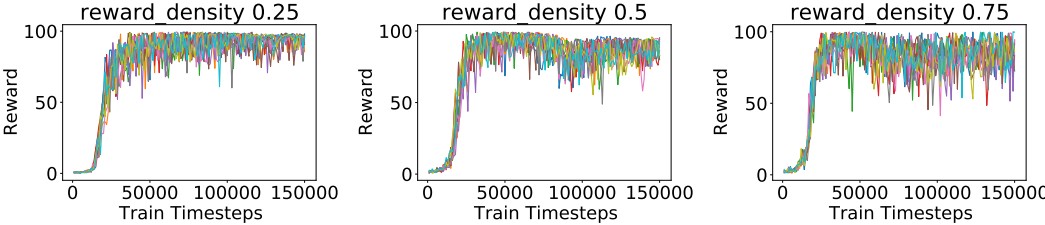

Figure 67: Evaluation Learning Curves for A3C **when varying reward sparsity**. Please note the different Y-axis scales.

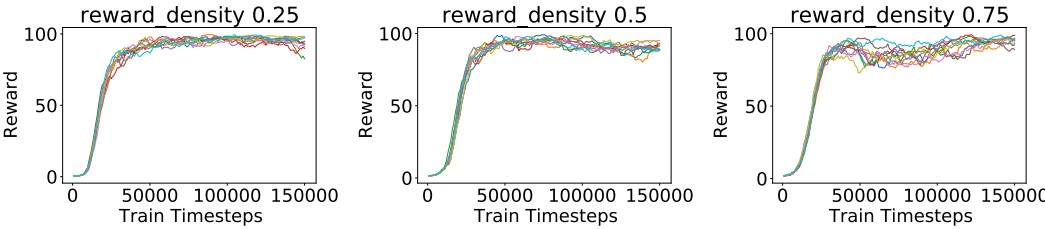

Figure 68: Training Learning Curves for A3C + LSTM **when varying reward sparsity**. Please note the different Y-axis scales.

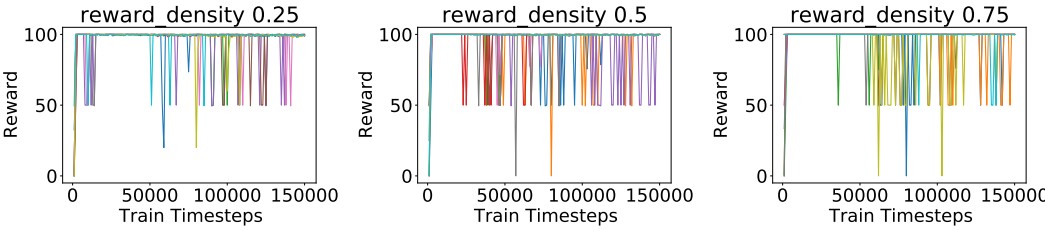

Figure 69: Evaluation Learning Curves for A3C + LSTM **when varying reward sparsity**. Please note the different Y-axis scales.

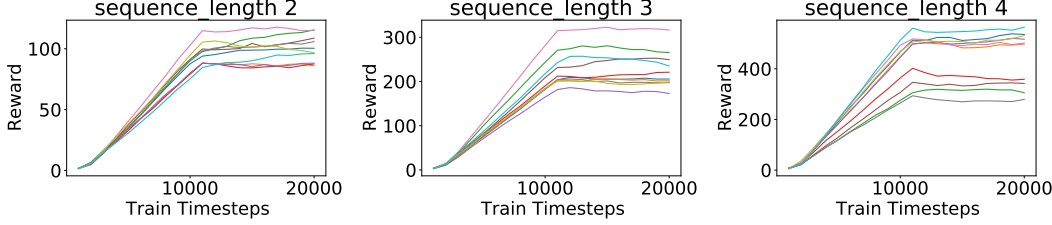

Figure 70: Training Learning Curves for Rainbow **when *make_denser* is *True* for rewardable sequences**. Please note the different Y-axis scales.

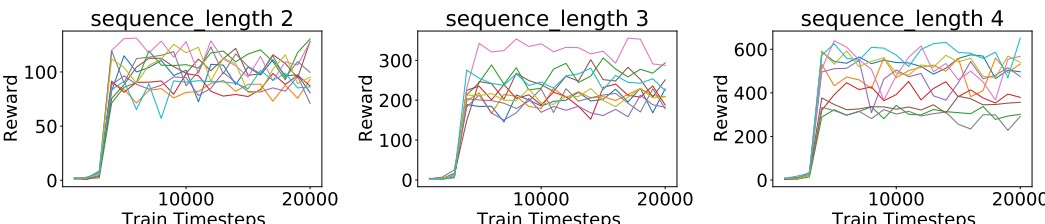

Figure 71: Evaluation Learning Curves for Rainbow **when *make_denser* is *True* for rewardable sequences**. Please note the different Y-axis scales.

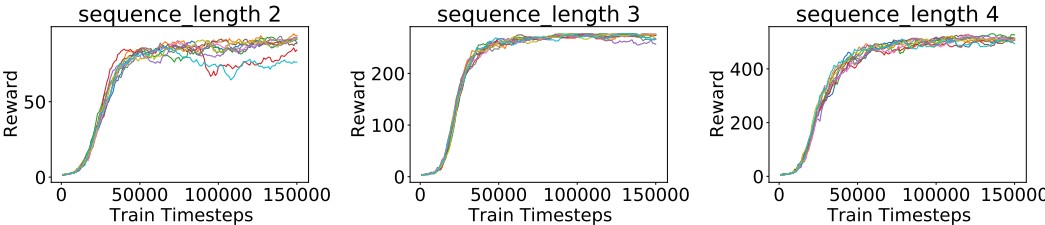

Figure 72: Training Learning Curves for A3C **when *make_denser* is *True* for rewardable sequences**. Please note the different Y-axis scales.

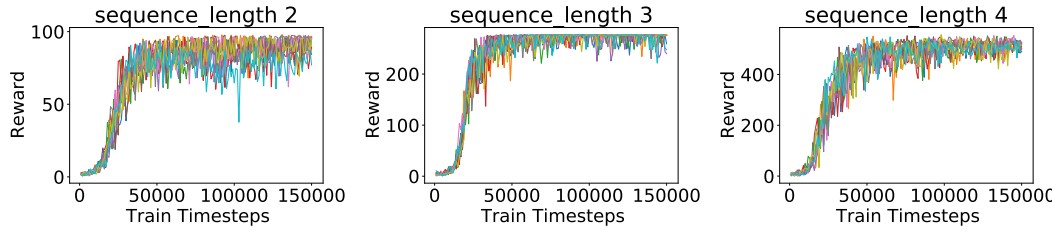

Figure 73: Evaluation Learning Curves for A3C **when *make_denser* is *True* for rewardable sequences**. Please note the different Y-axis scales.

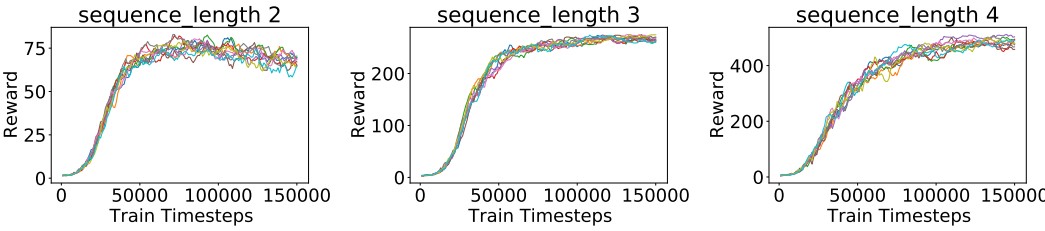

Figure 74: Training Learning Curves for A3C + LSTM **when *make_denser* is *True* for rewardable sequences**. Please note the different Y-axis scales.

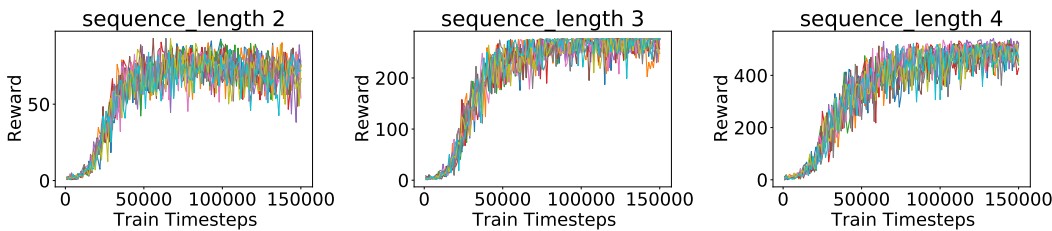

Figure 75: Evaluation Learning Curves for A3C + LSTM **when *make_denser* is *True* for rewardable sequences**. Please note the different Y-axis scales.

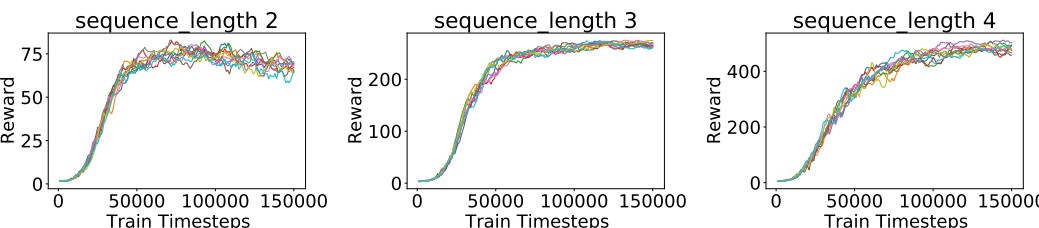

Figure 76: Training Learning Curves for A3C + LSTM **when *make_denser* is *True* for rewardable sequences**. Please note the different Y-axis scales.

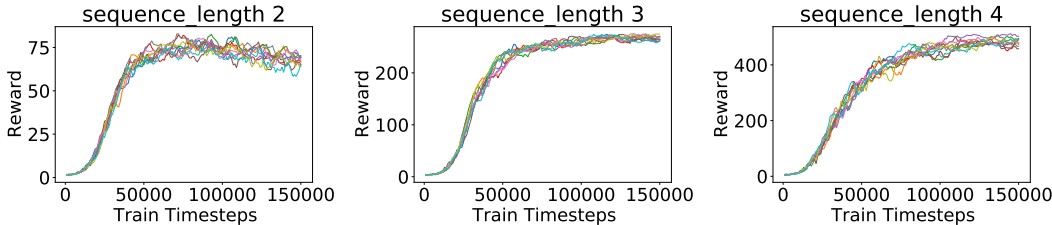

Figure 77: Training Learning Curves for A3C + LSTM **when *make_denser* is *True* for rewardable sequences**. Please note the different Y-axis scales.

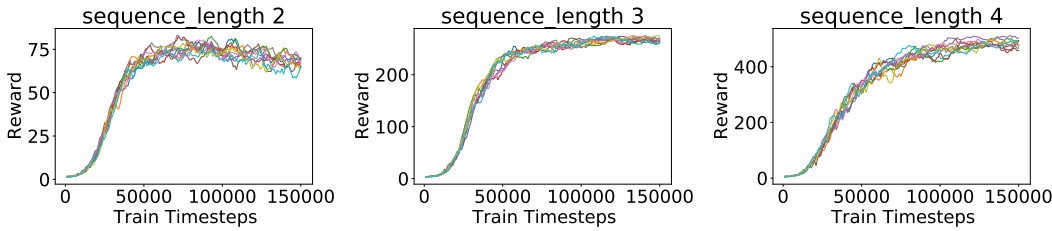

Figure 78: Training Learning Curves for A3C + LSTM **when *make_denser* is *True* for rewardable sequences**. Please note the different Y-axis scales.

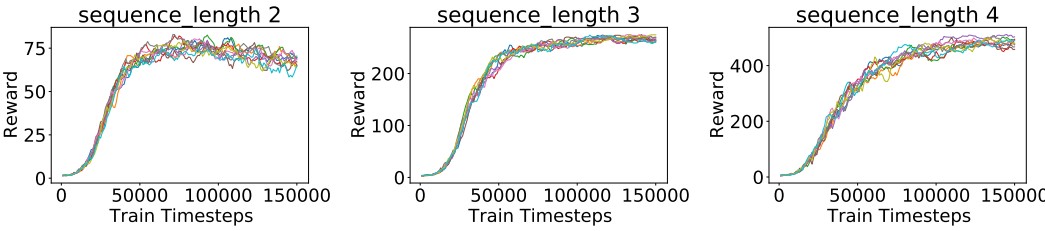

Figure 79: Training Learning Curves for A3C + LSTM **when *make_denser* is *True* for rewardable sequences**. Please note the different Y-axis scales.

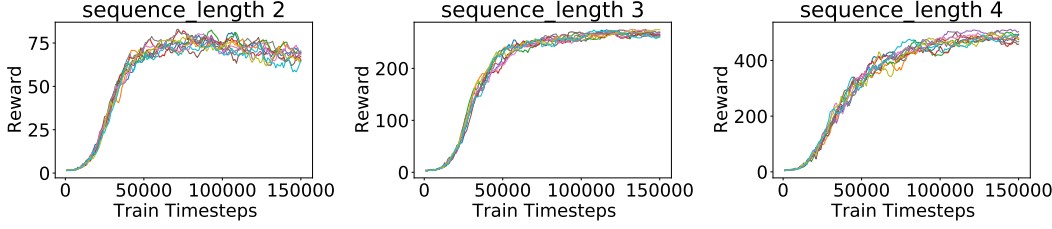

Figure 80: Training Learning Curves for A3C + LSTM **when *make_denser* is *True* for rewardable sequences**. Please note the different Y-axis scales.

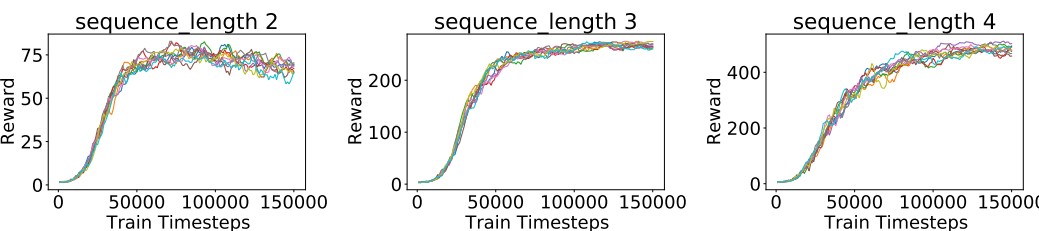

Figure 81: Training Learning Curves for A3C + LSTM **when *make_denser* is *True* for rewardable sequences**. Please note the different Y-axis scales.

# K  PERFORMANCE ON COMPLEX ENVIRONMENTS

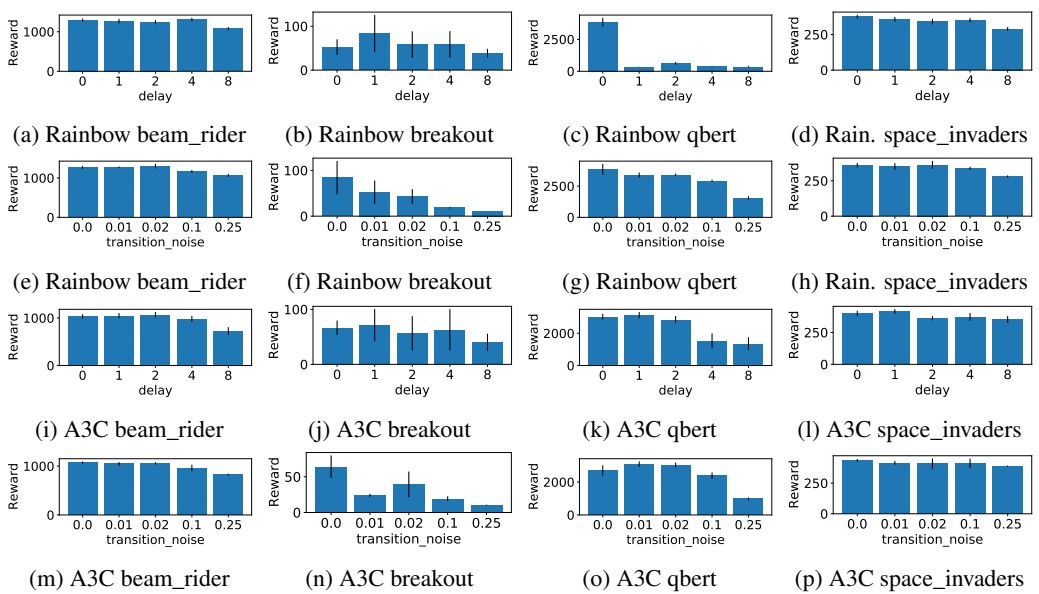

Figure 82: AUC of episodic reward for agents at the end of training. Error bars represent 1 standard deviation. Note the different y-axis scales.

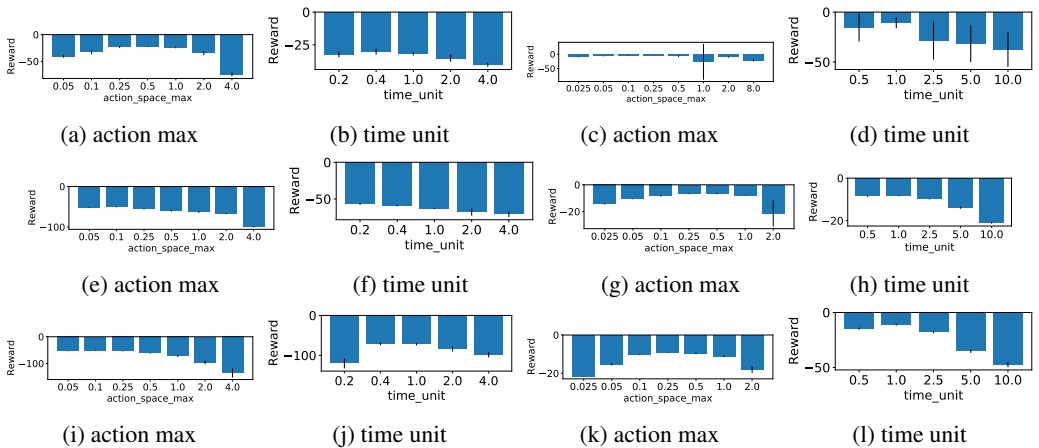

Figure 83: AUC of episodic reward at the end of training on HalfCheetah **varying action max** (top) and **time unit** (bottom). Error bars represent 1 standard deviation. Note the different y-axis scales.

## L  LEARNING CURVES FOR COMPLEX ENVIRONMENTS

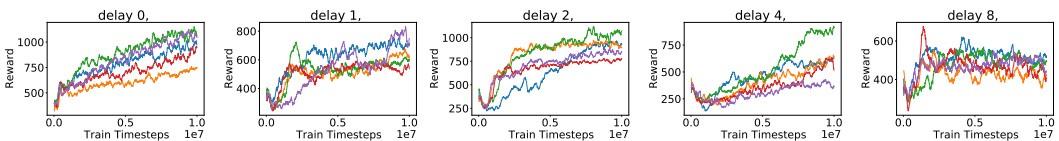

Figure 84: Training Learning Curves for DQN on beam_rider **when varying delay**. Please note the different Y-axis scales.

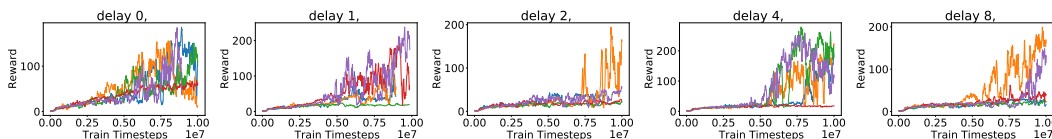

Figure 85: Training Learning Curves for DQN on breakout **when varying delay**. Please note the different Y-axis scales.

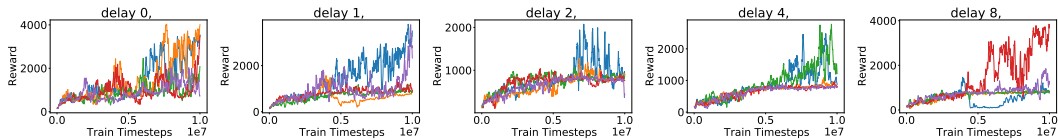

Figure 86: Training Learning Curves for DQN on qbert **when varying delay**. Please note the different Y-axis scales.

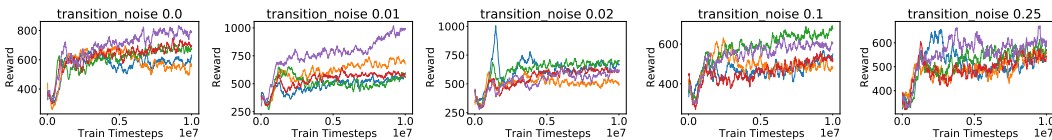

Figure 87: Training Learning Curves for DQN on space_invaders **when varying delay**. Please note the different Y-axis scales.

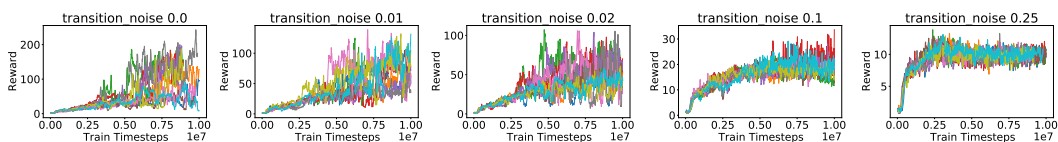

Figure 88: Training Learning Curves for DQN on beam_rider **when varying transition noise**. Please note the different Y-axis scales.

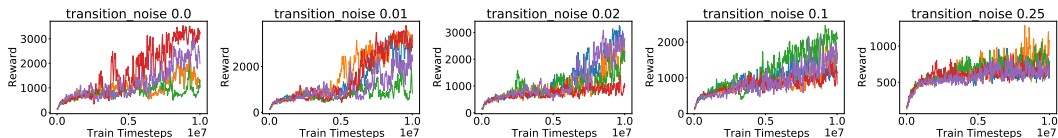

Figure 89: Training Learning Curves for DQN on breakout **when varying transition noise**. Please note the different Y-axis scales.

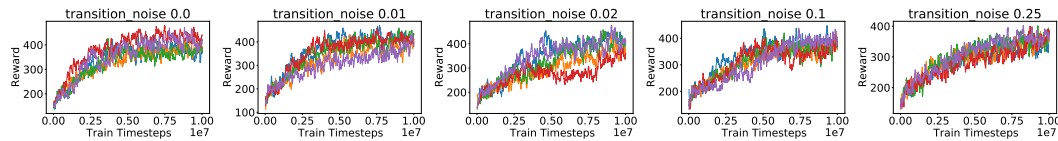

Figure 90: Training Learning Curves for DQN on qbert **when varying transition noise**. Please note the different Y-axis scales.

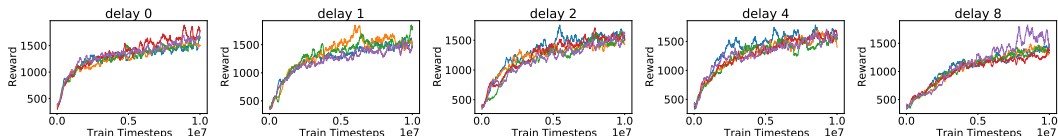

Figure 91: Training Learning Curves for DQN on space_invaders **when varying transition noise**. Please note the different Y-axis scales.

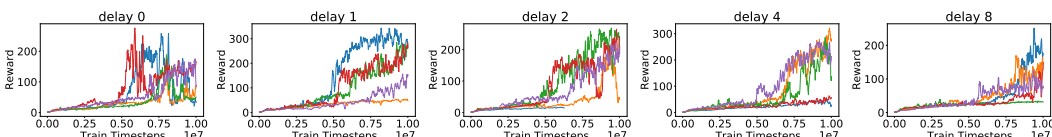

Figure 92: Training Learning Curves for Rainbow on beam_rider **when varying delay**. Please note the different Y-axis scales.

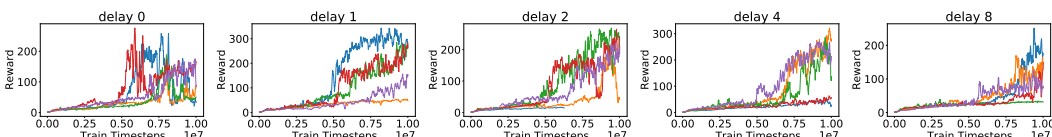

Figure 93: Training Learning Curves for Rainbow on breakout **when varying delay**. Please note the different Y-axis scales.

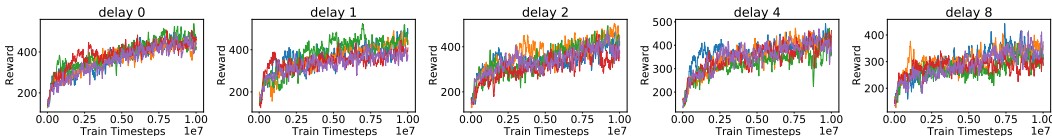

Figure 94: Training Learning Curves for Rainbow on qbert **when varying delay**. Please note the different Y-axis scales.

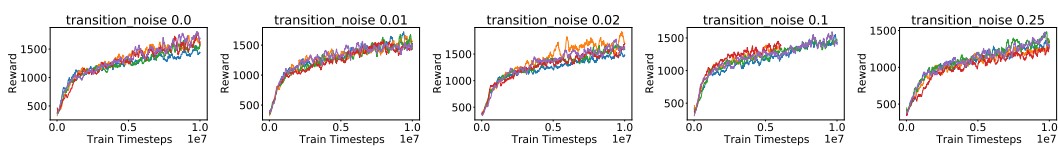

Figure 95: Training Learning Curves for Rainbow on space_invaders **when varying delay**. Please note the different Y-axis scales.

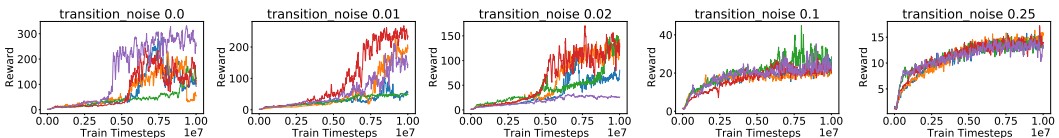

Figure 96: Training Learning Curves for Rainbow on beam_rider **when varying transition noise**. Please note the different Y-axis scales.

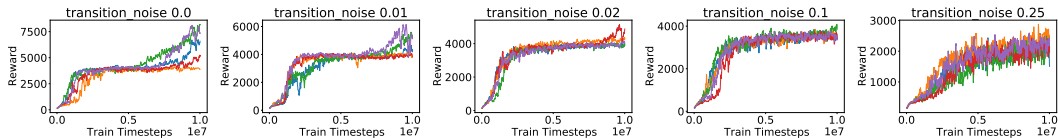

Figure 97: Training Learning Curves for Rainbow on breakout **when varying transition noise**. Please note the different Y-axis scales.

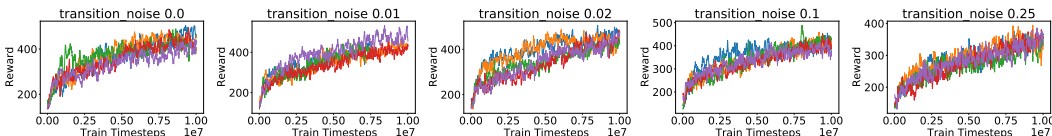

Figure 98: Training Learning Curves for Rainbow on qbert **when varying transition noise**. Please note the different Y-axis scales.

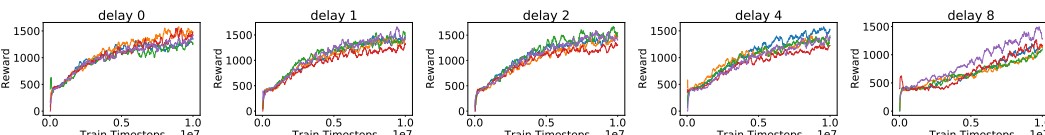

Figure 99: Training Learning Curves for Rainbow on space_invaders **when varying transition noise**. Please note the different Y-axis scales.

Figure 100: Training Learning Curves for A3C on beam_rider **when varying delay**. Please note the different Y-axis scales.

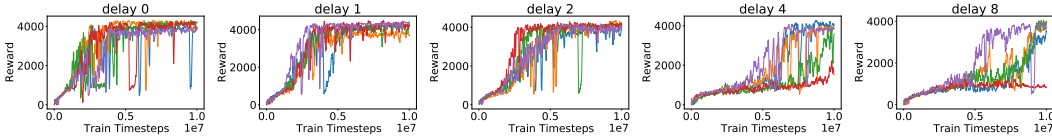

Figure 101: Training Learning Curves for A3C on breakout **when varying delay**. Please note the different Y-axis scales.

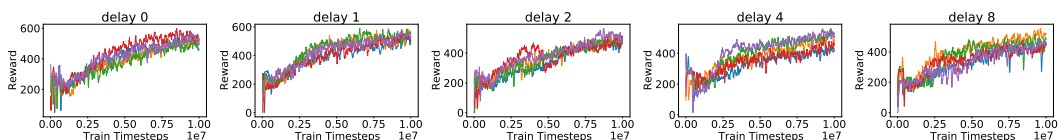

Figure 102: Training Learning Curves for A3C on qbert **when varying delay**. Please note the different Y-axis scales.

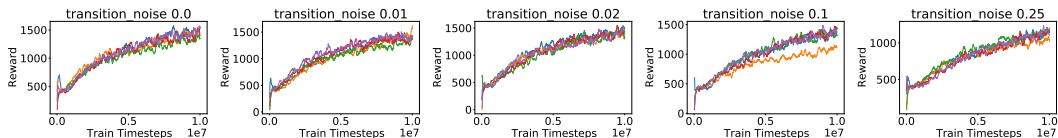

Figure 103: Training Learning Curves for A3C on space_invaders **when varying delay**. Please note the different Y-axis scales.

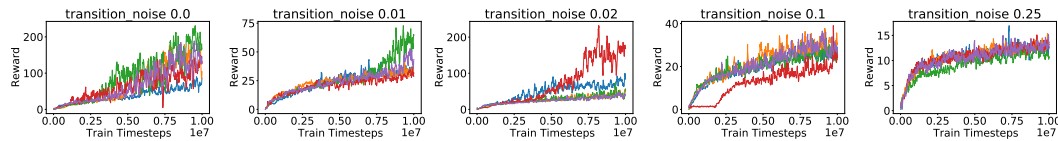

Figure 104: Training Learning Curves for A3C on beam_rider **when varying transition noise**. Please note the different Y-axis scales.

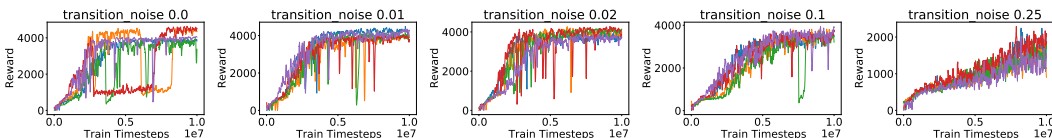

Figure 105: Training Learning Curves for A3C on breakout **when varying transition noise**. Please note the different Y-axis scales.

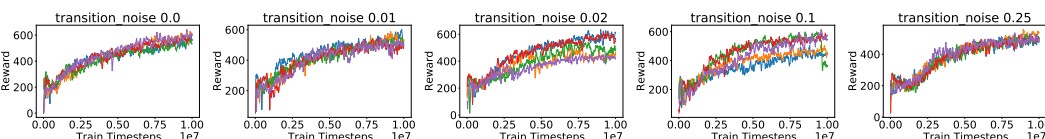

Figure 106: Training Learning Curves for A3C on qbert **when varying transition noise**. Please note the different Y-axis scales.

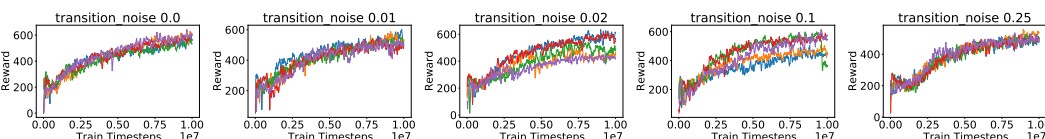

Figure 107: Training Learning Curves for A3C on space_invaders **when varying transition noise**. Please note the different Y-axis scales.

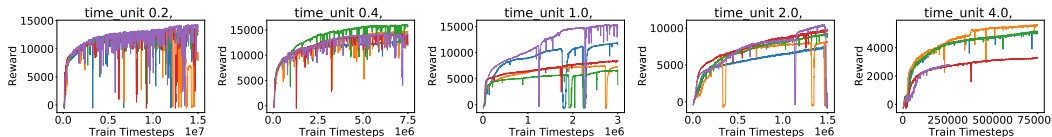

Figure 108: Training Learning Curves for SAC **when varying action max**. Please note the different Y-axis scales.

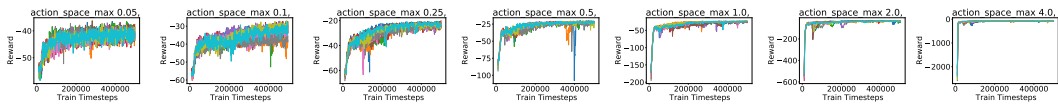

Figure 109: Training Learning Curves for SAC **when varying time unit**. Please note the different Y-axis scales.

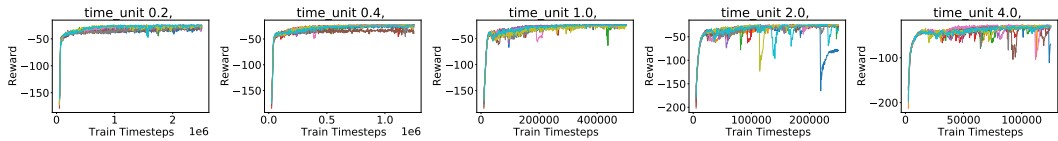

Figure 110: Training Learning Curves for SAC **when varying action max**. Please note the different Y-axis scales.

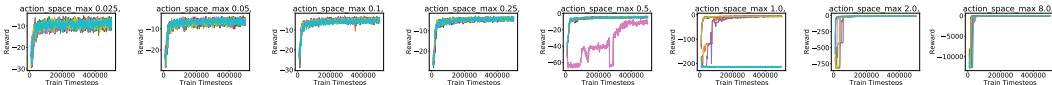

Figure 111: Training Learning Curves for SAC **when varying time unit**. Please note the different Y-axis scales.

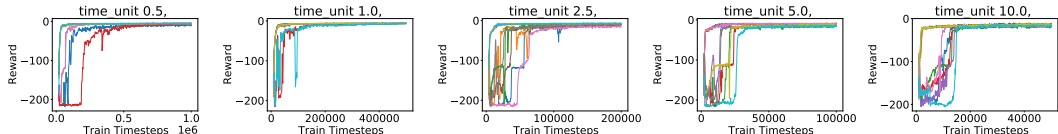

Figure 112: Training Learning Curves for SAC **when varying action max**. Please note the different Y-axis scales.

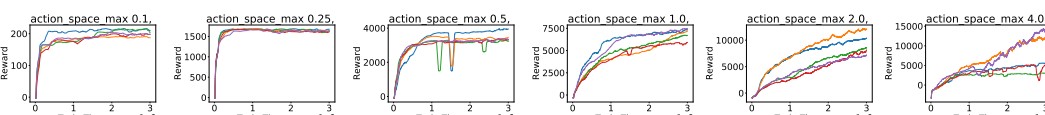

Figure 113: Training Learning Curves for SAC **when varying time unit**. Please note the different Y-axis scales.

Figure 114: Training Learning Curves for DDPG **when varying action max**. Please note the different Y-axis scales.

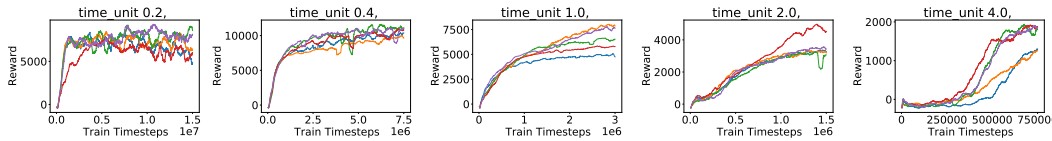

Figure 115: Training Learning Curves for DDPG **when varying time unit**. Please note the different Y-axis scales.

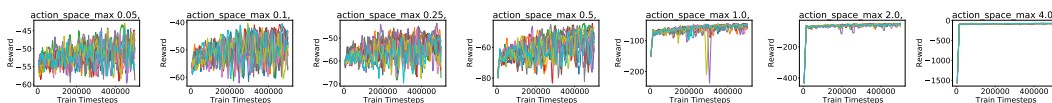

Figure 116: Training Learning Curves for DDPG **when varying action max**. Please note the different Y-axis scales.

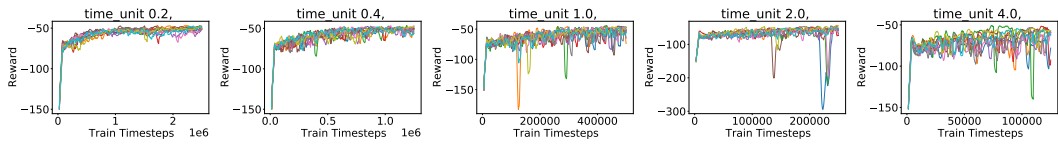

Figure 117: Training Learning Curves for DDPG **when varying time unit**. Please note the different Y-axis scales.

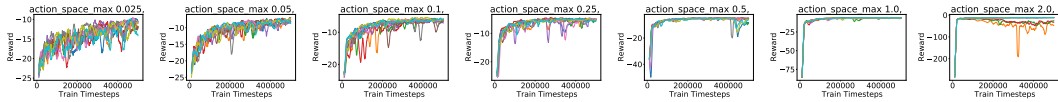

Figure 118: Training Learning Curves for DDPG **when varying action max**. Please note the different Y-axis scales.

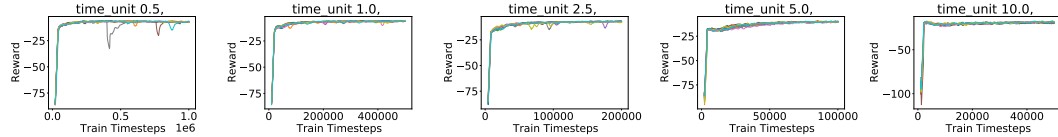

Figure 119: Training Learning Curves for DDPG **when varying time unit**. Please note the different Y-axis scales.

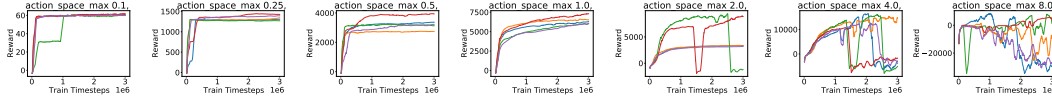

Figure 120: Training Learning Curves for TD3 **when varying action max**. Please note the different Y-axis scales.

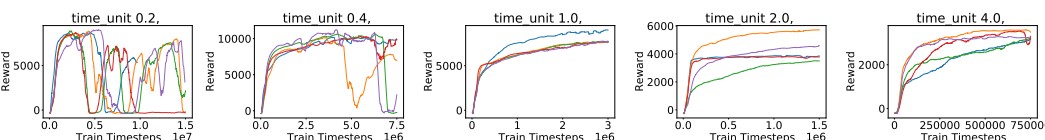

Figure 121: Training Learning Curves for TD3 **when varying time unit**. Please note the different Y-axis scales.

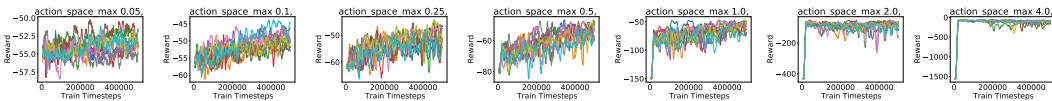

Figure 122: Training Learning Curves for TD3 **when varying action max**. Please note the different Y-axis scales.

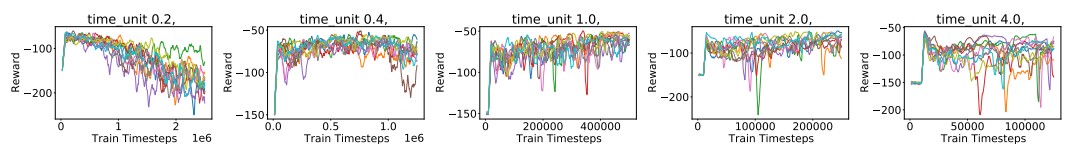

Figure 123: Training Learning Curves for TD3 **when varying time unit**. Please note the different Y-axis scales.

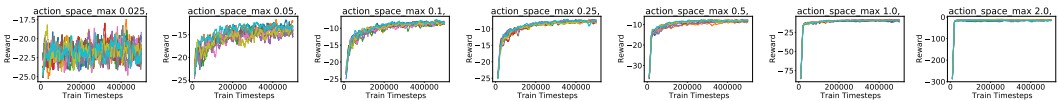

Figure 124: Training Learning Curves for TD3 **when varying action max**. Please note the different Y-axis scales.

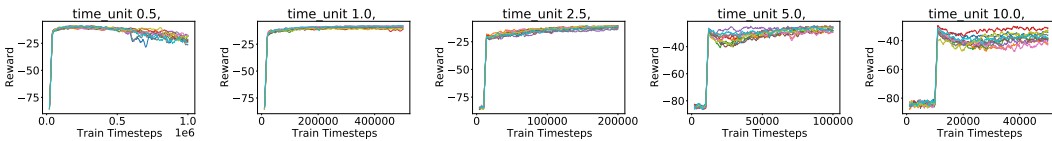

Figure 125: Training Learning Curves for TD3 **when varying time unit**. Please note the different Y-axis scales.

## M    HYPERPARAMETER TUNING

We gained some interesting insights into the significance of certain hyperparameters while tuning them for the different algorithms. Thus, our toy environments might in fact be good test beds for researching hyperparameters in RL, too. For instance, *target network update frequency* turned out to be very significant for learning and sub-optimal values led to very noisy and unreliable training and unexpected results such as networks with greater capacity not performing well. Once we tuned it, however, training was much more reliable and, as expected, networks with greater capacity did well. We now describe the tuning process and an example insight in more detail.

Hyperparameters were tuned for the vanilla environment; we did so manually in order to obtain good intuition about them before applying automated tools. We tuned the hyperparameters in sets, loosely in order of their significance and did 3 runs over each setting to get a more robust performance estimate. We describe a small part of our hyperparameter tuning for DQN next. All hyperparameter settings for tuned agents can be found in Appendix N.

We expected that quite small neural networks would already perform well for such toy environments and we initially grid searched over small network sizes (Figure 126a). However, the variance in performance was quite high (Figure 126b). When we tried to tune DQN hyperparameters *learning starts* and *target network update frequency*, however, it became clear that the target network update frequency was very significant (Figure 126c and 126d) and when we repeated the grid search over network sizes with a better value of 800 for the target network update frequency (instead of the oldd 80) this led to both better performance and lower variance (Figure 126e and 126f).

We then changed the network number of neurons grid to [128, 256, 512] and changed target network update frequency grid to [80, 800, 8000] and continued with further tuning using the grid values specified in Appendix N.

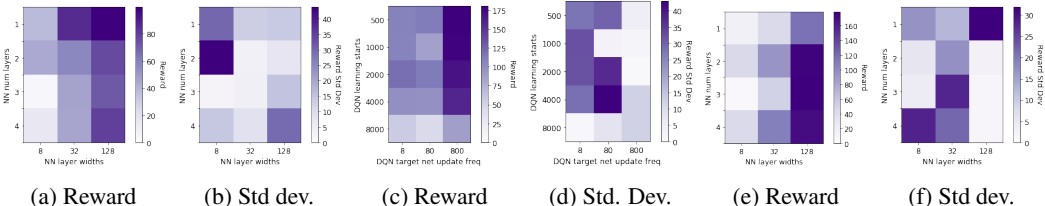

| (a) Reward | (b) Std dev. | (c) Reward | (d) Std. Dev. | (e) Reward | (f) Std dev. |

Figure 126: Mean episodic reward at the end of training for different hyperparameter sets for DQN. Please note the different colorbar scales.

# N  TUNED HYPERPARAMETERS

The code for corresponding experiments for both discrete and continuous environments can be found in the accompanying code for the paper. The README describes how to run the experiments using *config* files and which *config* files correspond to which experiments. Experiments on the discrete environments were run with Ray 0.7.3, while for the continuous environments, they were run with Ray 0.9.0. We had to use Ray 0.7.3 for the discrete environments and Ray 0.9.0 for the continuous ones because we had run the discrete cases for a previous version of the paper on 0.7.3. DDPG was not working and SAC was not implemented in Ray at that time. We tried to use Ray 0.9.0 also for the discrete version but found for the 1st algorithms we tested that, for the same hyperparameters, the results did not transfer even across implementations of the same library. This further makes our point about using our platform to unit test algorithms.

Since we did not save the hyperparameter grids for discrete environments in separate files, they are provided here. The names of the hyperparameters for the algorithms will match those used in Ray 0.7.3.

## N.1  DQN

```
num_layerss = [1, 2, 3, 4]
layer_widths = [8, 32, 128] # at first
layer_widths = [128, 256, 512] # after setting
    target_net_update_freq = 800 showed that 128 was the best
    number of the old 3, we changed search grid for number of
    neurons
fcnet_activations = ["tanh", "relu", "sigmoid"]
learning_startss = [500, 1000, 2000, 4000, 8000]
target_network_update_freqs = [8, 80, 800] # at first
target_network_update_freqs = [80, 800, 8000] # after seeing
    target_net_update_freq = 800 is much better than 80, changed
    the grid for it
double_dqn = [False, True]
learning_rates = [1e-2, 1e-3, 1e-4, 1e-5, 1e-6]
adam_epsilons = [1e-3, 1e-4, 1e-5, 1e-6] # also tried [1e-1, 1e-4,
    1e-7, 1e-10]

tune.run(
    "DQN",
    stop={
        "timesteps_total": 20000,
            },
    config={
      "adam_epsilon": 1e-4,
      "beta_annealing_fraction": 1.0,
      "buffer_size": 1000000,
      "double_q": False,
      "dueling": False,
      "exploration_final_eps": 0.01,
      "exploration_fraction": 0.1,
      "final_prioritized_replay_beta": 1.0,
      "hiddens": None,
      "learning_starts": 1000,
      "lr": 1e-4,
      "n_step": 1,
      "noisy": False,
      "num_atoms": 1,
      "prioritized_replay": False,
      "prioritized_replay_alpha": 0.5,
      "sample_batch_size": 4,
```

```
      "schedule_max_timesteps": 20000,
      "target_network_update_freq": 800,
      "timesteps_per_iteration": 100,
      "train_batch_size": 32,

      "env": "RLToy-v0",
      "env_config": {
        'dummy_seed': dummy_seed,
        'seed': 0,
        'state_space_type': 'discrete',
        'action_space_type': 'discrete',
        'state_space_size': state_space_size,
        'action_space_size': action_space_size,
        'generate_random_mdp': True,
        'delay': delay,
        'sequence_length': sequence_length,
        'reward_density': reward_density,
        'terminal_state_density': terminal_state_density,
        'repeats_in_sequences': False,
        'reward_unit': 1.0,
        'make_denser': False,
        'completely_connected': True
        },
    "model": {
        "fcnet_hiddens": [256, 256],
        "custom_preprocessor": "ohe",
        "custom_options": {},
        "fcnet_activation": "tanh",
        "use_lstm": False,
        "max_seq_len": 20,
        "lstm_cell_size": 256,
        "lstm_use_prev_action_reward": False,
        },

            "callbacks": {
              "on_episode_end": tune.function(on_episode_end),
              "on_train_result": tune.function(on_train_result),
          },
        "evaluation_interval": 1,
        "evaluation_config": {
        "exploration_fraction": 0,
        "exploration_final_eps": 0,
        "batch_mode": "complete_episodes",
        'horizon': 100,
          "env_config": {
            "dummy_eval": True,
            }
      },
      },
      )
```

## N.2 RAINBOW

```
num_layerss = [1, 2, 3, 4]
layer_widths = [128, 256, 512]
fcnet_activations = ["tanh", "relu", "sigmoid"]
learning_rates = [1e-2, 1e-3, 1e-4, 1e-5, 1e-6]
learning_startss = [500, 1000, 2000, 4000, 8000]
target_network_update_freqs = [80, 800, 8000]
```

```
double_dqn = [False, True]

tune.run(
    "DQN",
    stop={
        "timesteps_total": 20000,
            },
    config={
      "adam_epsilon": 1e-4,
      "buffer_size": 1000000,
      "double_q": True,
      "dueling": True,
      "lr": 1e-3,
      "exploration_final_eps": 0.01,
      "exploration_fraction": 0.1,
      "schedule_max_timesteps": 20000,
      "learning_starts": 500,
      "target_network_update_freq": 80,
      "n_step": 4,
      "noisy": True,
      "num_atoms": 10,
      "prioritized_replay": True,
      "prioritized_replay_alpha": 0.75,
      "prioritized_replay_beta": 0.4,
      "final_prioritized_replay_beta": 1.0,
      "beta_annealing_fraction": 1.0,

      "sample_batch_size": 4,
      "timesteps_per_iteration": 1000,
      "train_batch_size": 32,
      "min_iter_time_s": 1,

      "env": "RLToy-v0",
      "env_config": {
        'dummy_seed': dummy_seed,
        'seed': 0,
        'state_space_type': 'discrete',
        'action_space_type': 'discrete',
        'state_space_size': state_space_size,
        'action_space_size': action_space_size,
        'generate_random_mdp': True,
        'delay': delay,
        'sequence_length': sequence_length,
        'reward_density': reward_density,
        'terminal_state_density': terminal_state_density,
        'repeats_in_sequences': False,
        'reward_unit': 1.0,
        'make_denser': False,
        'completely_connected': True
        },
    "model": {
        "fcnet_hiddens": [256, 256],
        "custom_preprocessor": "ohe",
        "custom_options": {},
        "fcnet_activation": "tanh",
        "use_lstm": False,
        "max_seq_len": 20,
        "lstm_cell_size": 256,
        "lstm_use_prev_action_reward": False,
```

```
        },
            "callbacks": {
               "on_episode_end": tune.function(on_episode_end),
               "on_train_result": tune.function(on_train_result),
            },
            "evaluation_interval": 1,
            "evaluation_config": {
            "exploration_fraction": 0,
            "exploration_final_eps": 0,
            "batch_mode": "complete_episodes",
            'horizon': 100,
              "env_config": {
                 "dummy_eval": True,
                 }
            },
        },
    },
 )
```

## N.3 A3C

```
Grids of value for the hyperparameters over which they were tuned:

num_layerss = [1, 2, 3, 4]
layer_widths = [64, 128, 256]

learning_rates = [1e-2, 1e-3, 1e-4, 1e-5, 1e-6]
fcnet_activations = ["tanh", "relu", "sigmoid"]

lambdas = [0, 0.5, 0.95, 1.0]
grad_clips = [10, 30, 100]

vf_loss_coeffs = [0.1, 0.5, 2.5]
entropy_coeffs = [0.001, 0.01, 0.1, 1]

tune.run(
    "A3C",
    stop={
        "timesteps_total": 150000,
          },
    config={
            "sample_batch_size": 10,
            "train_batch_size": 100,
            "use_pytorch": False,
            "lambda": 0.0,
            "grad_clip": 10.0,
            "lr": 0.0001,
            "lr_schedule": None,
            "vf_loss_coeff": 0.5,
            "entropy_coeff": 0.1,
            "min_iter_time_s": 0,
            "sample_async": True,
            "timesteps_per_iteration": 5000,
            "num_workers": 3,
            "num_envs_per_worker": 5,

            "optimizer": {
                "grads_per_step": 10
            },
```

```
        "env": "RLToy-v0",
        "env_config": {
          'dummy_seed': dummy_seed,
          'seed': 0,
          'state_space_type': 'discrete',
          'action_space_type': 'discrete',
          'state_space_size': state_space_size,
          'action_space_size': action_space_size,
          'generate_random_mdp': True,
          'delay': delay,
          'sequence_length': sequence_length,
          'reward_density': reward_density,
          'terminal_state_density': terminal_state_density,
          'repeats_in_sequences': False,
          'reward_unit': 1.0,
          'make_denser': False,
          'completely_connected': True
          },
      "model": {
          "fcnet_hiddens": [128, 128, 128],
          "custom_preprocessor": "ohe",
          "custom_options": {},
          "fcnet_activation": "tanh",
          "use_lstm": False,
          "max_seq_len": 20,
          "lstm_cell_size": 256,
          "lstm_use_prev_action_reward": False,
          },

              "callbacks": {
                "on_episode_end": tune.function(on_episode_end),
                "on_train_result": tune.function(on_train_result),
              },
          "evaluation_interval": 1,
          "evaluation_config": {
          "exploration_fraction": 0,
          "exploration_final_eps": 0,
          "batch_mode": "complete_episodes",
          'horizon': 100,
            "env_config": {
              "dummy_eval": True,
              }
      },
      },
 )
```

## N.4   A3C + LSTM

```
Grids of value for the hyperparameters over which they were tuned:

num_layerss = [1, 2, 3, 4]
layer_widths = [64, 128, 256]

learning_rates = [1e-2, 1e-3, 1e-4, 1e-5, 1e-6]
fcnet_activations = ["tanh", "relu", "sigmoid"]

lambdas = [0, 0.5, 0.95, 1.0]
grad_clips = [10, 30, 100]
```

```
vf_loss_coeffs = [0.1, 0.5, 2.5]
entropy_coeffs = [0.001, 0.01, 0.1, 1]

lstm_cell_sizes = [64, 256, 512]
lstm_use_prev_action_rewards = [False, True]

tune.run(
    "A3C",
    stop={
        "timesteps_total": 150000,
            },
    config={
            "sample_batch_size": 10,
            "train_batch_size": 100,
            "use_pytorch": False,
            "lambda": 0.0,
            "grad_clip": 10.0,
            "lr": 0.0001,
            "lr_schedule": None,
            "vf_loss_coeff": 0.1,
            "entropy_coeff": 0.1,
            "min_iter_time_s": 0,
            "sample_async": True,
            "timesteps_per_iteration": 5000,
            "num_workers": 3,
            "num_envs_per_worker": 5,

            "optimizer": {
                "grads_per_step": 10
            },

      "env": "RLToy-v0",
      "env_config": {
        'dummy_seed': dummy_seed,
        'seed': 0,
        'state_space_type': 'discrete',
        'action_space_type': 'discrete',
        'state_space_size': state_space_size,
        'action_space_size': action_space_size,
        'generate_random_mdp': True,
        'delay': delay,
        'sequence_length': sequence_length,
        'reward_density': reward_density,
        'terminal_state_density': terminal_state_density,
        'repeats_in_sequences': False,
        'reward_unit': 1.0,
        'make_denser': False,
        'completely_connected': True
        },
    "model": {
        "fcnet_hiddens": [128, 128, 128],
        "custom_preprocessor": "ohe",
        "custom_options": {},
        "fcnet_activation": "tanh",
        "use_lstm": True,
        "max_seq_len": delay + sequence_length,
        "lstm_cell_size": 64,
        "lstm_use_prev_action_reward": True,
```

```
        },

            "callbacks": {
              "on_episode_end": tune.function(on_episode_end),
              "on_train_result": tune.function(on_train_result),
            },
        "evaluation_interval": 1,
        "evaluation_config": {
        "exploration_fraction": 0,
        "exploration_final_eps": 0,
        "batch_mode": "complete_episodes",
        'horizon': 100,
          "env_config": {
            "dummy_eval": True,
            }
    },
    },
)
```

## O  MORE ON CONCLUSION AND FUTURE WORK

Among the continuous environments, we have a toy task of moving along a line. Here, we hand out greater rewards the closer a point object is to moving along a line. This is also a better task to test exploration than the completely random discrete environments. It already gave some interesting results and further work will follow. We are in the process of implementing plug and play *model-based* metrics to evaluate model-based algorithms, such as the Wasserstein metric (likely a sampled version because analytical calculation would be intractable in many cases) between the true dynamics models and the learnt one to keep track of how model learning is proceeding. Our Environments plan to allow using their transition and reward functions to perform *imaginary* rollouts without affecting the current state of the system.

Another significant meta-feature is *reachability* in the transition graph. We believe a lot of insights can be gained from graph theory to model toy environments which try to mimic specific real life situations at a very high level. We plan that users can specify their own transition graphs and also plan to add random generation of specific types of transition graphs.

Even though we have a playground to generate environments where the dimensions such as sequence length are constant, being able to solve environments with variable delay and sequence lengths and identifying them (i.e., segmentation of events in the time domain) is another area we are currently working on with attention-based agents and various other ideas.

The fine-grained control of dimensions allows relating these to good hyperparameter choices. So, our playground could also be used to learn a mapping from hardness dimensions to hyperparameters for different types of environments and even to warm-start hyperparameter optimisation for environments with similar hardness dimensions. This holds promise for future meta-learning algorithms. In a similar vein, it could also be used to perform Combined Algorithm Selection and Hyperparameter Optimisation (Thornton et al., 2013), since it's clear that currently different RL algorithms do well in different kinds of environments.

Further interesting toy experiments which are already possible with our platform are varying the terminal state densities to have environments for benchmarking *safe RL*.

The states and actions contained in a rewardable sequence could just be a single *compound* state and *compound* action if we discretised time in a suitable manner. This brings us to the idea of learning at multiple timescales. HRL algorithms with formulations like the options framework (Sutton et al., 1999), could try to identify these rewardable sequences at the higher level and then carry out *atomic* actions at the lower level.

We also hope to benchmark other algorithms like PPO[5] (Schulman et al., 2017), Rudder (Arjona-Medina et al., 2019), MCTS (Silver et al., 2016), DDPG[6] (Lillicrap et al., 2016) on continuous tasks and table-based algorithms and to show theoretical results match with practice on toy benchmarks.

We also aim to promote reproducibility in RL as in (Henderson et al., 2018) and hope our benchmark helps with that goal. To this end, we have already improved the Gym `Box` and `Discrete Spaces` to allow their seeds to be controlled at initialization time as well.

We need different RL algorithms for different environments. Aside from some basic heuristics such as applying DDPG (Lillicrap et al., 2016) to continuous environments and DQN to discrete environments, it is not very clear when to use which RL algorithms. We hope this will be a first step to being able to identify from the environment what sort of algorithm to use and to help build adaptive algorithms which adapt to the environment at hand. Additionally, aside from being a great benchmark for RL algorithms, it is also a great didactic tool for teaching how RL algorithms work in different environments.

---

[5]We tried PPO but could not get it to learn

[6]We tried DDPG also but there seemed to be a bug in the implementation and it crashed even on tuned examples from Ray

## P CPU SPECIFICATIONS

```
processor       : 0
vendor_id       : GenuineIntel
cpu family      : 6
model           : 158
model name      : Intel(R) Core(TM) i7-8850H CPU @ 2.60GHz
stepping        : 10
microcode       : 0xb4
cpu MHz         : 900.055
cache size      : 9216 KB
physical id     : 0
siblings        : 12
core id         : 0
cpu cores       : 6
apicid          : 0
initial apicid  : 0
fpu             : yes
fpu_exception   : yes
cpuid level     : 22
wp              : yes
flags           : fpu vme de pse tsc msr pae mce cx8 apic sep mtrr
    pge mca cmov pat pse36 clflush dts acpi mmx fxsr sse sse2 ss
   ht tm pbe syscall nx pdpe1gb rdtscp lm constant_tsc art
   arch_perfmon pebs bts rep_good nopl xtopology nonstop_tsc
   cpuid aperfmperf tsc_known_freq pni pclmulqdq dtes64 monitor
   ds_cpl vmx smx est tm2 ssse3 sdbg fma cx16 xtpr pdcm pcid
   sse4_1 sse4_2 x2apic movbe popcnt tsc_deadline_timer aes xsave
    avx f16c rdrand lahf_lm abm 3dnowprefetch cpuid_fault epb
   invpcid_single pti ssbd ibrs ibpb stibp tpr_shadow vnmi
   flexpriority ept vpid ept_ad fsgsbase tsc_adjust bmi1 hle avx2
    smep bmi2 erms invpcid rtm mpx rdseed adx smap clflushopt
   intel_pt xsaveopt xsavec xgetbv1 xsaves dtherm ida arat pln
   pts hwp hwp_notify hwp_act_window hwp_epp md_clear flush_l1d
bugs            : cpu_meltdown spectre_v1 spectre_v2
   spec_store_bypass l1tf mds swapgs
bogomips        : 5184.00
clflush size    : 64
cache_alignment : 64
address sizes   : 39 bits physical, 48 bits virtual
power management:
```

