# OpenReview forum: "MDP Playground: Controlling Orthogonal Dimensions of Hardness in Toy Environments"
_ICLR.cc/2021/Conference — Reject_

### Official Review · AnonReviewer2 · 2020-10-15
**What insights can be gained?**

**Rating:** 4
**Confidence:** 4

**Review:**

This paper proposes a suite of benchmark tasks designed to test (and possibly debug) reinforcement learning algorithms. Deemed the MDP Playground, these environments are applicable to both discrete and continuous RL agents and allow tuning of various dimensions of complexity - reward delays, reward sparsity, stochasticity, etc. The authors demonstrate their framework by evaluating the performance of many well-known RL agents across a variety of these playground environments. Additionally they conduct similar experiments on Atari and Mujoco tasks and observe similar trends in agent performance when injecting noise, reward delays, and varying action max values. Finally, the MDP Playground is very quick to run and facilitates fast experimentation.

It's my view that the efficacy of a testing and debugging suite like MDP Playground is measured by the actionable insights that can be generated with it. To this end the authors describe some findings that may be applicable in the design of new environments (such as action max needing tuning in continuous action environments), but little is shown about new insights gained toward understanding shortcomings of existing algorithms or routes for building better RL agents.

Additionally, why do we believe that the structure of the MDPs generated by MDP Playground will resemble that of the problems that RL practitioners in the community are interested in solving? Specifically for discrete environments having a completely connected transition function consisting of 8 states and 8 actions seems like it may not resemble more complicated environments like Atari. Similarly, moving a pointmass in a 2D plane likely has many differences from learning how to locomote a multi-jointed robot. It's not clear that insights gained from Playground environments will transfer to more complex environments.

Tangentially, it might be interesting to have a tool that could analyze a particular complex environment and automatically generate a corresponding Playground MDP that would somehow capture the same measures of difficulty, such that agents could be debugged/optimized in this low-cost environment before being transferred back to the complex environment.

I have read the author response and stand by my original score of the paper.

---

> ### Author Response · Authors · 2020-11-17
> **Response to AnonReviewer2**
>
> >but little is shown about new insights gained toward understanding shortcomings of existing algorithms or routes for building better RL agents.
>
> Thank you for pointing that out. We can discuss more in detail about shortcomings of existing algorithms, e.g., how most of the algorithms we have surveyed cannot identify these dimensions in environments but we as humans can give a rough measure of each of these dimensions in the situations we are faced with. And we could also discuss some new research directions (we’ve already done a bit of this in 3.1 in common response to reviewers), however, all the breadth and diversity of RL is a very open research area with (too) many dimensions and we believe putting these together in a highly tunable platform is itself a very important tool that promotes quick development of algorithms. Identifying the dimensions and allowing fine-grained control over them is not an easy undertaking. Further, we would like to ask you to read the other reviewers’ comments about wanting a more detailed description of the dimensions themselves. So, if you would like we can discuss insights in more detail but we would kindly request you to discuss with the other reviewers whether we should rather add more insights or more description of the dimensions in the main paper.
>
>
> >… more complicated environments like Atari. Similarly, moving a pointmass in a 2D plane likely has many differences from learning how to locomote a multi-jointed robot.It's not clear that insights gained from Playground environments will transfer to more complex environments.
>
> We’re sorry that you feel this way, we have performed extensive experiments showing high-level transfer of trends to more complex environments in Section 4.3. We have further discussed what kinds of trends we think will not transfer to more complex environments. For instance, we believe our environment cannot capture complex dynamics interactions between different joints and have said in the text “We believe this is due to correlations within the multiple degrees of freedom as opposed to a rigid object in the toy environment.”. The cases you have mentioned are too specific for a toy benchmark and our platform wasn’t designed with those in mind. Like we have said in 3 in the common response to all reviewers, MDP Playground is meant as a bridge between theory and practice. We apologise if that was not clear enough. We’re not trying to claim that it will capture complex/high variance aspects like multi-jointed robots but it does capture general aspects like the significance of the time unit which we have shown transfer experiments for.

---

### Official Review · AnonReviewer1 · 2020-10-26
**Tries to address an important issue but falls short. The presented analysis is unprincipled and incomplete.**

**Rating:** 5
**Confidence:** 4

**Review:**

--------------------------------------

POST-REBUTTAL COMMENTS

As a result of the discussion the paper has improved, so I'm increasing my score. However, the core issue, that the proposed benchmarks don't seem to capture the difficulty structure of either real problems or more complex benchmarks, remains.

--------------------------------------


SUMMARY:

The paper introduces MDP Playground, a parameterized suite of MDPs with low computational requirements that nonetheless present significant challenges for existing RL algorithms. The authors argue that MDP Playground is a valuable testbed for developing and evaluating ne RL algorithms. The paper also assess how the identified dimensions of hardness that can be exercised in MDP Playground transfer to more complex benchmarks. The paper also uses MDP Playground to evaluate several rllib algorithms.


HIGH-LEVEL COMMENTS:

This work has a lot of potential. It targets an important problem in RL research: studying the behavior of RL algorithms as an environment changes along various dimensions of hardness. It is valuable to have a benchmark suite of the sort this work aims to deliver, one that allows varying these dimensions in a controlled way and consists of problems that are simple enough for debugging.

However, in its current form this work has non-trivial weaknesses in contribution and presentation that make publication at ICLR or similar conferences premature:


1) MDP Playground's problems are completely unintuitive. They are randomly generated and aren't inspired by any real scenarios. This raises the question of how meaningful they are. In ML, it is understood from results such as the no-free-lunch theorem that no learner can do well across all possible tasks, and the sensible approach is to make ML algorithms work well on meaningful data distributions that are encountered in reality. Unfortunately, RL as a field hasn't been good at sticking to this principle: many of its existing benchmarks, such as videogames, look "interesting" and difficult, but in a very different way that real decision-making scenarios are. MDP Playground exacerbates this issue -- since its MDPs are randomly generated and don't have a natural interpretation, it's difficult to get even an intuition for a good behavior in them, and difficult to understand how to generate problems that that have a combination of hardness along different dimensions that is representative of realistic scenarios.

A case in point is the paper's omission of MDP non-ergodicity (and presence of constraints on the desired policy as a common real factor that causes non-ergodicity) from its list of hardness dimensions. In reality, and even in some existing benchmarks, the learner can reach irrecoverable (absorbing) failure states, such as robots damaging themselves or objects they interact with, unless they are very careful. There are goal-directed MDP models with such states -- see, e.g., reference [a] at the end of the review -- and learning in their presence in realistic scenarios requires costly resets -- see, e.g., reference [b]. MDP Playground doesn't help with researching this aspect, e.g., by having knobs for probabilities of entering such states, penalties for doing so and the cost of recovery, despite it being ubiquitous in reality.

Another major omission is the fact that in reality agent actions have durations that can make it very hard to learn a policy within a given time budget. The existing benchmark doesn't allow studying the effects of action duration on learning time.

Given these gaps, I doubt that MDP Playground in its current form adds much value over the existing RL benchmarks, which have some drawbacks but have interpretability as a big asset for debugging.


2) The paper's analysis of dimensions of hardness is quite unprincipled, contrary to the paper's claims. What makes many decision-making problems (both existing ones and those introduced in MDP Playground) hard is partial observability, but the paper never formally states what POMDPs are, nor even mentions this term. If it defined POMDPs formally, the incorrectness of some of the statements the paper makes about problems with partial observability (see the detailed comments below) would become obvious, and the connections between partial observability and hardness would become much clearer mathematically. See, e.g., reference [c] for a formal but accessible treatment of POMDPs.



TECHNICAL ISSUES/RELATED WORK:

-- In the intro, the paper claims that "partial observability [is] when the underlying environment is assumed to be an MDP, however, the state formulation, i.e., the observation used by the agent is not Markovian". This is an extremely inaccurate statement at best. In POMDPs, observations are Markovian -- their probability depends only on the current (hidden) state. "State formulation" is also Markovian, as it is fully observable MDPs. So are the belief states. What is non-Markovian in POMDPs is the optimal policy w.r.t. the observations: an optimal policy can depend on the entire observation history. However, again, each observation history maps to a belief state, and in the belief state space the optimal policies are Markovian.


-- The same goes for several other statements, e.g. "performance degrades in environments where the delayed reward induces partial observability and hence makes the state used by the algorithm non-Markovian". How can delayed reward introduce partial observability? How can partial observability make "the state used by the algorithm" (an imprecise term in its own right) non-Markovian?


-- Out of the dimensions of hardness the paper does identify, why is "sequence length" a distinct dimension, rather than being an instance of reward delay or reward sparsity?


-- In the related work, a notable omission is reference [d], which looks at aspects that MDP hard and analyzes existing benchmarks w.r.t. those aspects.


-- I wasn't sure what the benefit of the "varying reprsentations" experiment that tries to enforce various kinds of invariances was. First of all, the experiment is about applying various types of data augmentation to images, not about varying representations. Second, the conclusion that "this indicates that shift and other types of invariance do not come for free and that one needs to have sufficient amount of samples for the algorithm to become invariant to the transforms we desire." is rather obvious and well-known.


-- A similar remark goes for several other experiments: varying time units, target variance, irrelevant features. Their results are completely expected, and while they might be a useful sanity check, the results description can be condensed to 1 line each and placed in the figure captions.


-- A similar comment goes for the entire content of Section 4.3 as well-- it is very verbose and can be greatly condensed and structured into a short list with bullet points summraizing the findings.


Typos:

"Very low cost execution" --> "Very low cost of execution"


[a] Kolobov, Mausam, Weld, "A Theory of Goal-Oriented MDPs with Dead Ends", UAI-2012

[b] Eysenbach, Gu, Ibarz, Levine,  "Leave no Trace: Learning to Reset for Safe and Autonomous Reinforcement Learning", ICLR-2018

[c] Kochenderfer, "Decision Making Under Uncertainty: Theory and Application", MIT Press, 2015

[d] Maillard, Mann, Mannor "“How hard is my MDP?” The distribution-norm to the rescue", NIPS-2014

---

> ### Author Response · Authors · 2020-11-17
> **Response to AnonReviewer1 part 1**
>
> >In ML, it is understood from results such as the no-free-lunch theorem that no learner can do well across all possible tasks, and the sensible approach is to make ML algorithms work well on meaningful data distributions that are encountered in reality.
>
> This is true for empirical research on complex environments. However, for theoretical development of algorithms, making algorithms work on meaningful data distributions that are encountered in reality may not always be a requirement. For example, Q-learning is agnostic to the structure in *P* and *R* and should work in theory for **any** kind of *P* and *R*.
>
> >Unfortunately, RL as a field hasn't been good at sticking to this principle: many of its existing benchmarks, such as videogames, look "interesting" and difficult, but in a very different way that real decision-making scenarios are.
>
> >MDP Playground exacerbates this issue -- since its MDPs are randomly generated and don't have a natural interpretation, it's difficult to get even an intuition for a good behavior in them, and difficult to understand how to generate problems that that have a combination of hardness along different dimensions that is representative of realistic scenarios.
>
> We have made a case for how we want to help researchers identify inductive biases in 3.1 in the common response to reviewers. To do this, even random data can work as long as the data have the important features in there. Also as mentioned there, most RL algorithms that we are aware of are agnostic to the structure of *P* and *R*, so we believe it’s not wrong to test them on randomly generated *P*s and *R*s. The bsuite debugging example we have given in Appendix E.1 supports our claim. We only needed the key feature of varying reward sparsity there.
>
> >A case in point is the paper's omission of MDP non-ergodicity (and presence of constraints on the desired policy as a common real factor that causes non-ergodicity) from its list of hardness dimensions. In reality, and even in some existing benchmarks, the learner can reach irrecoverable (absorbing) failure states, such as robots damaging themselves or objects they interact with, unless they are very careful.
>
> >MDP Playground doesn't help with researching this aspect, e.g., by having knobs for probabilities of entering such states, penalties for doing so and the cost of recovery, despite it being ubiquitous in reality.
>
> Thank you, we have a couple of responses to this.
> Firstly, we allow setting *terminal state density* which achieves non-ergodicity by having terminal states which only transition back to themselves. Further, we have *terminal state costs* which can also be specified, so at least in some form non-ergodicity *is* captured. Secondly, as we have said in principle (1) above, we aim to allow users to specify their own *P* and *R* and this is easily ready by the end of the rebuttal.
> Thirdly, we are also happy to consider adding it as a separate dimension according to our commitment (2) above. To that end, could you please point out if you’re not satisfied with the first two proposals here?
>
> >Another major omission is the fact that in reality agent actions have durations that can make it very hard to learn a policy within a given time budget. The existing benchmark doesn't allow studying the effects of action duration on learning time.
>
> We are confused by this statement. We believe from your description, that the included dimension *time unit* is exactly this feature (please see the last paragraph of Section 2 in the paper for the relevant equation). It's the time that the actions last for in the continuous environments. Could you please clarify what you believe to be missing here?
>
> >The paper's analysis of dimensions of hardness is quite unprincipled, contrary to the paper's claims.
>
> We have explicitly stated our principle for selecting the dimensions here. Could you please tell us if that makes it better for you?
>
> >In the intro, the paper claims that "partial observability [is] when the underlying environment is assumed to be an MDP, however, the state formulation, i.e., the observation used by the agent is not Markovian". This is an extremely inaccurate statement at best. In POMDPs, observations are Markovian -- their probability depends only on the current (hidden) state. "State formulation" is also Markovian, as it is fully observable MDPs. So are the belief states. What is non-Markovian in POMDPs is the optimal policy w.r.t. the observations: an optimal policy can depend on the entire observation history. However, again, each observation history maps to a belief state, and in the belief state space the optimal policies are Markovian.
>
> Thank you for pointing this out. We're sorry if our statements were inaccurate. We’re happy to reframe it to be accurate for the rebuttal.

---

> > ### Author Response · Authors · 2020-11-17
> > **Response to AnonReviewer1 part 2**
> >
> > >Out of the dimensions of hardness the paper does identify, why is "sequence length" a distinct dimension, rather than being an instance of reward delay or reward sparsity?
> >
> > We thought we clearly distinguished sequence length from delay in the text. Apologies if it wasn’t clear enough. The rewardable sequences are action sequences that cause a reward. This reward may be handed out after a further delay at the end of the sequence. The percentage of random sequences that result in a reward is the reward density. Let us provide another example here to clarify this: say a rat needs to walk around in a circle and then push a ball down a ramp to achieve a reward and the reward is only handed *after* the ball has rolled down the ramp. The *sequence* of actions that causes the reward is walking in a circle followed by pushing the ball down the ramp. The *delay* is the amount of time it takes for the ball from leaving the rat’s paws to roll down the ramp only *after* which a reward is handed out. Any action taken by the rat while the ball is rolling down doesn't affect the fact that the rat gets the (delayed) reward, so only the sequence of actions (trajectory) up to pushing the ball needs decision making by the rat. Now sparsity is again orthogonal to both sequences and delay. Assume the circle the rat had to move in (before pushing the ball) to get the reward was of a radius in the range [1, 2] cm. Then the reward is much sparser than if the rat would have had to move in a circle with any radius because many more sequences would allow the rat to achieve the reward. We hope this example makes the dimensions clearer.
> >
> > >In the related work, a notable omission is reference [d], which looks at aspects that MDP hard and analyzes existing benchmarks w.r.t. those aspects.
> >
> > Thank you for the reference. We weren't aware of it. We will add it to the list of related work for the rebuttal.
> >
> > >I wasn't sure what the benefit of the "varying reprsentations" experiment that tries to enforce various kinds of invariances was. First of all, the experiment is about applying various types of data augmentation to images, not about varying representations.
> >
> > Since there's a difference between optimising for pixel representations and optimising for feature space representations we wanted to give researchers the ability to have multiple representations for the same underlying state. The first part of the experiment was turning on pixel representations instead of feature space representations. Further, the pixel representations of the same internal state in feature space do vary when we apply what you term data augmentations and what we termed transforms.
> >
> > >Second, the conclusion that "this indicates that shift and other types of invariance do not come for free and that one needs to have sufficient amount of samples for the algorithm to become invariant to the transforms we desire." is rather obvious and well-known.
> >
> > Could you please point out the relevant literature here? If the results are indeed well known, that supports our claim that the dimensions carry over from our toy environments to complex environments and that is what a lot of researchers care about (which again justifies usage of randomly generated transitions). A further aspect that was surprising is that shift was the transform that caused the poorest performance despite the spatial invariance of convolutions, which we found surprising and thus worthwhile to report on.
> >
> >
> > >Typos:
> > "Very low cost execution" --> "Very low cost of execution"
> >
> > To the best of our knowledge, this was a paragraph heading and we don’t need the “of” for such headings.
> >
> > >This work has a lot of potential. It targets an important problem in RL research: studying the behavior of RL algorithms as an environment changes along various dimensions of hardness. It is valuable to have a benchmark suite of the sort this work aims to deliver, one that allows varying these dimensions in a controlled way and consists of problems that are simple enough for debugging.
> >
> > Thank you for the kind words.

---

> > > ### Comment · AnonReviewer1 · 2020-11-19
> > > **R1's  response to REBUTTAL part 2**
> > >
> > > > We thought we clearly distinguished sequence length from delay in the text.
> > >
> > > You have indeed, but my question is: why is this necessary? Why isn't sequence length = n the same as reward delay = n-1 ?
> > >
> > > > Second, the conclusion that "this indicates that shift and other types of invariance do not come for free and that one needs to have sufficient amount of samples for the algorithm to become invariant to the transforms we desire." is rather obvious and well-known. -- Could you please point out the relevant literature here?
> > >
> > > What you are talking about here is an instance of overfitting, so the relevant literature is any ML textbook: learning an invariance needs either a lot of data or a special inductive bias. If you don't have either, you'll overfit.
> > >
> > > > To the best of our knowledge, this was a paragraph heading and we don’t need the “of” for such headings.
> > >
> > > What you seem to be saying is that you are free to drop prepositions in headings/titles. If so, this is simply not true -- you can find plenty of counterexamples in any book by Cambridge University Press or on the BBC website.

---

> > > > ### Author Response · Authors · 2020-11-23
> > > > **Response to R1's response**
> > > >
> > > > >We thought we clearly distinguished sequence length from delay in the text.
> > > >
> > > > >>You have indeed, but my question is: why is this necessary? Why isn't sequence length = n the same as reward delay = n-1 ?
> > > >
> > > > Apologies if our intent was unclear. During the delay period, the agent may play any action but these actions are inconsequential to the reward. For a sequence, the agent needs to be able to play actions that are consequential to the reward. Conceptually, these target different aspects of the problem.  You can have one independently of the other.

---

> > ### Comment · AnonReviewer1 · 2020-11-19
> > **R1's  response to REBUTTAL part 1**
> >
> > > This is true for empirical research on complex environments. However, for theoretical development of algorithms, making algorithms work on meaningful data distributions that are encountered in reality may not always be a requirement. For example, Q-learning is agnostic to the structure in P and R and should work in theory for any kind of P and R.
> >
> > It's indeed not a requirement for theoretical algorithm development, but once theoretical development is complete, researchers usually want to see how the resulting algorithm works on "interesting" problems, i.e., those that resemble some realistic scenarios.
> >
> > > We have made a case for how we want to help researchers identify inductive biases in 3.1 in the common response to reviewers.
> >
> > Please see my response to that common rebuttal -- the lack of realistic "feel" is nonetheless an issue.
> >
> > > Firstly, we allow setting terminal state ....
> >
> > This doesn't quite do it -- modeling failure states requires being able to set different costs for different terminal states: high (possibly infinite) costs for failure states and negative costs  (i.e., rewards) for goal states.
> >
> > > We believe from your description, that the included dimension time unit is exactly this feature...
> >
> > It's not quite the same -- what I meant is that execution of different actions can take a different amount of time, even in discrete-space setting. So, if you have a limited time budget to learn a policy, the RL algorithm needs to be cognizant of this budget and of the duration of various actions.
> >
> > > We have explicitly stated our principle for selecting the dimensions here ...
> >
> > My comment about the analysis being unprincipled referred largely to the numerous inaccurate and incorrect claims about partial observability.

---

> > > ### Author Response · Authors · 2020-11-23
> > > **Response to R1's response**
> > >
> > > >Firstly, we allow setting terminal state ....
> > >
> > > >>This doesn't quite do it -- modeling failure states requires being able to set different costs for different terminal states: high (possibly infinite) costs for failure states and negative costs (i.e., rewards) for goal states.
> > >
> > > Apologies if this is not clear from the text but this *is* currently possible because the goal state reward is handed out regardless of terminal costs. So, terminal states (whether rewarding or failure states) *can* have different costs.

---

> > > > ### Comment · AnonReviewer1 · 2020-11-24
> > > > **--**
> > > >
> > > > OK, sounds good, then it's just a matter of clarifying/emphasizing it in the paper.

---

> > > ### Author Response · Authors · 2020-11-23
> > > **Response to R1's response part 2**
> > >
> > > >We have explicitly stated our principle for selecting the dimensions here ...
> > >
> > > >>My comment about the analysis being unprincipled referred largely to the numerous inaccurate and incorrect claims about partial observability.
> > >
> > > Regarding inaccuracy about partial observability, we were trying to incorporate changes suggested by you but, unfortunately, we couldn’t publicly find the reference you suggested. Further, we went through Richard Sutton and Andrew Barto’s *Reinforcement Learning 2nd edition* (http://incompleteideas.net/book/RLbook2020.pdf) and tried to understand what might have been wrong in our statements. However, having gone through the book, we believe there’s been a misunderstanding about the semantics of *states*, *observations*, etc. that we have used regarding partial observability. When we say *state (formulation) used by the agent*, we don’t mean the *state of the MDP/POMDP* but the *state that the agent uses as its internal representation*. Sutton and Barto’s book also uses similar terminology. For instance, here are excerpts from the book,
> > >
> > > *In practice, the states of real agents will not be Markov but may approach it as an ideal.*
> > >
> > > *A common strategy for finding a Markov state is to look for something compact that is recursively updatable and enables accurate short-term predictions.*
> > >
> > > *The approximate state will play the same role in our algorithms as before, so we continue to use the notation S_t for the state used by the agent, even though it may not be Markov.*
> > >
> > > Like that textbook, we mean the (approximate) state used by the agent and not the MDP state (that is not fully observable by the agent). So, we will try to clarify this in our text and make this distinction clear in every statement. For instance, we will try now to clarify the following concerns you had.
> > >
> > > >How can delayed reward introduce partial observability? How can partial observability make "the state used by the algorithm" (an imprecise term in its own right) non-Markovian?
> > >
> > > Assume we have a fully observable *MDP M* and an agent which uses the observable state *s* as its internal state representation and receives reward *r_t* at time *t*. Now, in the delayed reward setting, we define a new *MDP M’* in which we give the agent the reward *r’_{t+d} = r_t*, where *d* is the delay. The same agent still using internal state representation *s* will then be using a non-Markovian state representation for *M’*. Does that sound better to you?

---

> > > > ### Comment · AnonReviewer1 · 2020-11-24
> > > > **States, observations, etc**
> > > >
> > > > There are a few smaller issues here that are manifestations of a big overarching one: the paper is using highly non-standard and imprecise POMDP terminology. I don't think the way to fix it is by introducing your name for a concept and then explain what you mean by that name. Rather, simply use standard POMDP terminology and you'll be fine. You can explain what it means in the paper just to refresh the reader's memory, but that's optional.
> > > >
> > > > In particular, in brief:
> > > >
> > > > -- The term for the internal state the agent uses to make decisions is "information state". In MDPs, an information state sufficient for computing an optimal policy coincides with the MDP's state. In POMDPs,  an information state sufficient for computing an optimal policy is the belief state, which is equivalent to a full observation history. A belief state is indeed different from the POMDP's latent state (sometimes called just "state"), which the agent doesn't have access to, and different from the concept of observation.
> > > >
> > > > -- The term "Markovian" doesn't apply to any of the above concepts of state (and in a paper, I wouldn't apply it to the concept of observations either). This term applies to transition functions, reward functions, observation functions (in the case of POMDPs), and policies. However, there is a concept of "Markov state", which denote a state formulation in terms of which the reward, transition, and observation functions are Markovian.
> > > >
> > > >
> > > > There are a few recommendations that follow from this:
> > > >
> > > > -- If you want to claim a principled coverage of partial observability, use the term "POMDP". It is the established term for MDPs with partial observability.
> > > >
> > > > -- State a formal definition of POMDP in the paper, the same way you state a formal MDP definition, as a tuple. Use well-established terms for each element of that tuple.
> > > >
> > > > -- State an optimal POMDP solution class -- the class of deterministic policies that are Markovian in the agent's belief state.
> > > >
> > > > -- Check and, if needed, correct everything you say about partial observability in the paper. E.g., as you can see from the above, the statement "the state formulation, i.e., the observation used by the agent is not Markovian" is both imprecise and wrong.

---

> > > > > ### Author Response · Authors · 2020-11-24
> > > > > **Thank you**
> > > > >
> > > > > Thank you very much for the detailed and quick responses! We are incorporating your feedback into the paper for the rebuttal version.

---

> > > > ### Comment · AnonReviewer1 · 2020-11-24
> > > > **Delayed rewards**
> > > >
> > > > > Assume we have a fully observable MDP M and an agent which uses the observable state s as its internal state representation and receives reward r_t at time t. Now, in the delayed reward setting, we define a new MDP M’ in which we give the agent the reward r’_{t+d} = r_t, where d is the delay. The same agent still using internal state representation s will then be using a non-Markovian state representation for M’. Does that sound better to you?
> > > >
> > > > I'm not sure how this relates to (partial) observability. But in any case, two things:
> > > >
> > > > -- Be careful when you say "in the delayed reward setting, we define a new MDP M’ in which we give the agent the reward r’_{t+d} = r_t, where d is the delay." -- if you just take the original MDP and modify it *only* by replacing its reward function with a delayed one, what you get isn't an MDP (or, rather , not a first-order MDP), because the reward function now isn't a function of just the current state. To make the new problem Markov, you need to augment the state space with a time variable.
> > > >
> > > > -- If the agent uses the state of the original MDP as its information state for the "delayed" decision problem, then I'd phrase it as "The agent's information state is non-Markov for the delayed-reward version of the problem."

---

> > > > > ### Author Response · Authors · 2020-11-24
> > > > > **Improvement to explanation**
> > > > >
> > > > > Thank you for pointing that out! We should have mentioned the state spaces of the old MDP M and the new MDP M' explicitly.
> > > > > In the delayed setting, the state space S' of M' would no longer be S, the state space of the original MDP (also equal to the common observation space O of M and M'), but is augmented with additional information to generate the rewards (i.e. M' is a proper MDP and we have implemented it using a similar augmented state as you have suggested). However, crucially, in our example, this additional information in the state of M' is not part of the information state of the agent (which still uses the original s from S). This additional information is exactly the part of the state s' from S' that is not observable (because it is present in S', but not in the observation space O and hence: partially observable).

---

### Official Review · AnonReviewer4 · 2020-10-29
**Interesting framework but choice of perturbations seems arbitrary.**

**Rating:** 4
**Confidence:** 5

**Review:**

This paper presents "MDP Playground", a family of procedurally generated MDPs that can be used to benchmark certain dimension of difficulty considered by the authors to be challenging to current RL algorithms.  The paper presents the effects of the various perturbations to the MDP on state-of-the-art learning algorithms and discusses particular dimension of interest in the paper.  A full and exhaustive analysis of results is presented in the appendix.  The "MDP Playground" is slated to be open-sourced so that the community can benchmark against it.


Pros:
I think that high-level difficulties of MDPs are under-studied and that the over-reliance on benchmarks such as Atari or Mujoco make people look more at per-environment/game performance instead of thinking about high-level issues with the MDPs (exploration, delays et.c). Although this is done in a hand-wavy manner, all attempts at formalising this seem essential to better understand what the actual degrees of difficulty are for particular tasks and whether novel proposed approaches are really tackling the challenge they claim to be tackling.

I find the use of two procedurally generated MDP, one continuous and the other discreet, in a very simple task setup to be a good design choice, and also allows for quick iteration.

Cons:
Although this is an opinionated position, I'm not super happy with the choice of perturbations. Some are very general, such as delays or stochasticity, while others are very specific such as target radius, action max, or action loss weight.  I feel the nomenclature around the dimensions of 'hardness' (nit, perhaps 'difficulty' would be a better word here) is not very clear.  The proposed dimensions seemed to be inspired by some tasks the authors are working with, but in that case it would make more sense to ground their choice by describing the tasks and arguing why these are particularly important.  For example, target radius seems to be completely arbitrary, I could define an infinite number of reward functions that describe a goal and use all sorts of topologies to window my reward and shape it as the agent nears the goal, is this really a general problem for RL though?  I remain unconvinced.

With relation to bsuite I would have also like to see more discussion on why the additional dimensions of difficulty make sense.  For example bsuite already proposes noise as an evaluation dimension, how does MDP Playground's noise differ?

More generally, not all dimensions were clearly described, in particular stochasticity, it wasn't clear to me how this was defined.  I would have appreciated more time spent on describing the challenges rather than the analysis of the results on all sorts of environments, in the end the core contribution here is the framework and its structure, the analysis is slightly out of scope given the length of the paper.

There seems to also be very similar work in this space [https://arxiv.org/abs/2003.11881] that also proposes an open-source benchmark, it would be interesting to compare the choice of hardness dimensions to the ones used here.

Conclusion:
Overall I think this is a good direction of work, but it is a bit too unprincipled, and the paper structure kind of confusing.  I would prefer perhaps less degrees of hardness, or perhaps a couple 'families' to make your thought process easier to understand.  Then, further discussion and grounding for each family of tasks would be great, to understand where these dimensions of hardness would manifest themselves.  Finally, spending more time on describing each hardness dimension clearly instead of compacting it all into the end of Section 2 would also make this an easier read.

I think this will be hard to achieve for the rebuttal phase, but I encourage continued work in this domain and look forward to seeing the authors' response.

---

> ### Author Response · Authors · 2020-11-17
> **Response to AnonReviewer4 part 1**
>
> >I feel the nomenclature around the dimensions of 'hardness' (nit, perhaps 'difficulty' would be a better word here) is not very clear.
>
> It is very important for us to have a clear nomenclature since a central goal of MDP Playground is adoption by the community. We welcome any discussion to find better terminology. We previously discussed between “hardness” and “difficulty”  and decided to go for 'hardness' as it sounded intuitively better to us. But that was a subjective preference on our part. Could you please provide some reasoning as to why you think ‘difficulty’ is a better fit here?
>
>
> >Some are very general, such as delays or stochasticity, while others are very specific such as target radius, action max, or action loss weight. … The proposed dimensions seemed to be inspired by some tasks the authors are working with, but in that case it would make more sense to ground their choice by describing the tasks and arguing why these are particularly important. For example, target radius seems to be completely arbitrary, I could define an infinite number of reward functions that describe a goal and use all sorts of topologies to window my reward and shape it as the agent nears the goal, is this really a general problem for RL though? I remain unconvinced.
>
> It is true that there is likely some bias in our selections because our research is still on a finite number of sources and we apologise for any bias that may have crept in. However, we kindly point out to the reviewer that the proposed dimensions are not inspired by any tasks we work on and we even tried to survey environments we haven’t previously worked with. *target radius*, *action max*, or *action loss weight* were introduced when we surveyed the continuous environments in the literature, most of them here (https://github.com/openai/gym/tree/master/gym/envs/mujoco), and were consistent with our principle of making the environments as parameterisable as possible. They *are*, however, specific to the continuous environments and not there for discrete environments.
>
> Regarding *target radius*, we use just one number specifying the distance from the goal to decide whether we have reached the goal or not. To the best of our knowledge, any other way to decide if we have reached a goal requires more than a single scalar. Consequently, we believe an n-dimensional sphere having a target radius is the only property which may be considered objective. We are curious about how you would define a reward function in this instance and are happy to include your suggestion.
>
> Regarding *action max*, maybe our terminology was confusing. By *action max*, we actually meant the action range *[action_min, action_max]*, just that in our case we decided to keep *action_min = - action_max* because the toy environment is symmetric. The action range itself is always present in continuous systems, so unless there was some confusion due to our poor terminology, we believe it to be general. If you disagree, could you be so kind as to elaborate?
>
> Regarding *action loss weight*, we can see how this might be seen as a subjective inclusion but this parameter is so frequently present in continuous control environments, that we feel omitting it would have left a common use case unaddressed (for instance, almost every environment in the Mujoco link above has the action loss weight). This can easily be turned off by setting it to 0 (the default). While, just like the reviewer, we would also love to be perfectly general and as objective as possible, for practical reasons we have included some dimensions which we felt were very common but these can always be turned off (which is usually the default).
>
> We welcome any constructive discussions and actionable suggestions to make our design decisions and selection of dimensions as objective and general as possible. Assuming that MDP Playground is designed to be improved based on debates like these, we hope the reviewer does not deem the proposed dimensions to be arbitrary but instead can appreciate the nature and the consequences of making design decisions and having to impose subjectivity in some instances.

---

> > ### Author Response · Authors · 2020-11-17
> > **Response to AnonReviewer4 part 2**
> >
> > >With relation to bsuite I would have also like to see more discussion on why the additional dimensions of difficulty make sense.
> >
> > While we very much appreciate bsuite and it was nice to see DeepMind come up with an idea similar to ours around the same time as us, we couldn’t find a principle unifying how they selected their dimensions similar to principle (1) of ours. They include some “higher-level” dimensions which have overlap and are not orthogonal (e.g., this can be seen by the overlap between the environments they used to score these dimensions: in Appendix A of the bsuite paper, see the issues addressed by individual environments). For instance, *basic* and *generalisation* have overlap. *Memory* and *credit assignment* we believe should have overlap. In addition, they also include “lower-level” dimensions like R noise which can be orthogonally applied and over which one can have fine-grained control. However, even R noise is fixed to have a grid of values. Our intention is to allow users fine-grained control by specifying values along orthogonal axes. In addition, bsuite’s dimensions can’t be orthogonally applied in the same base environment, but it is rather a collection of existing environments with one dimension each which can’t always be controlled. We have performed experiments varying 2 dimensions (see, e.g. Appendix I) and these have let us see correlations between performances across different dimensions. Finally, bsuite doesn’t have continuous environments which doesn’t allow testing of continuous agents which we feel excludes many important RL algorithms.
> >
> > >For example bsuite already proposes noise as an evaluation dimension, how does MDP Playground's noise differ?
> >
> > Regarding the noise, bsuite injects only noise in *R* and not noise in *P*. It also only has *R* scale as a dimension while we allow both *R* scale and *R* shift because we tried to exhaustively include everything.
> >
> >
> > >There seems to also be very similar work in this space [https://arxiv.org/abs/2003.11881] that also proposes an open-source benchmark, it would be interesting to compare the choice of hardness dimensions to the ones used here.
> >
> > Thanks for pointing out the paper, we will add a comparison to it. We know there is some overlap in the dimensions at least. However, it seems that that benchmark consists of much more specific environments, and only those that are continuous. Following the curriculum of benchmarks we proposed (last paragraph of section 5 of paper), they are further down in the RL development pipeline than MDP Playground or bsuite which target the toy environment domain.
> >
> > >I think that high-level difficulties of MDPs are under-studied and that the over-reliance on benchmarks such as Atari or Mujoco make people look more at per-environment/game performance instead of thinking about high-level issues with the MDPs (exploration, delays et.c). Although this is done in a hand-wavy manner, all attempts at formalising this seem essential to better understand what the actual degrees of difficulty are for particular tasks and whether novel proposed approaches are really tackling the challenge they claim to be tackling.
> >
> > Thank you for your kind words, this was our intention too.
> >
> > >I find the use of two procedurally generated MDP, one continuous and the other discreet, in a very simple task setup to be a good design choice, and also allows for quick iteration.
> >
> > This is precisely what we intended by using what we term "high bias" toy environments. Thank you for the appreciation!

---

> > > ### Comment · AnonReviewer4 · 2020-11-24
> > > **Response to 1281**
> > >
> > > Thank you for your responses.  In terms of evolutions of this work, I think the main thing to work on is a more clear choice of perturbations.  As another reviewer points out above, just adding a bunch of knobs isn't necessarily very helpful because a large combination of knob settings will be relatively uninteresting (either too hard, or hard in a way that is unrealistic, or maybe even too easy).  I know the choices in bsuite intuitively make sense to me because they are common themes in RL literature, and the selection in the 'challenges of real-world rl' is supported by examples in families of real application. Tasks such as 'target radius' seem very specific, and I'm not sure how I would use such a perturbation to validate a particular experimental insight.
> > >
> > > Perhaps one question is whether you're using these environments for other research projects or simply producing them as they stand.  Starting to write some papers that leverage these particular challenges could provide insights into which ones are really useful, and could help you whittle down the selection some.
> > >
> > > As to bsuite specifically, it was put on arxiv Agust 9th 2019 - I haven't tried to google your title to respect anonymity, but the claim of contemporaneous work does seem perhaps a bit extreme 1.5 years later...
> > >
> > > As it stands I will maintain the review as it is stands but if there is a revised paper I will read it and provide feedback.

---

> > > > ### Author Response · Authors · 2020-11-25
> > > > **Response to AnonReviewer4**
> > > >
> > > > >Thank you for your responses. In terms of evolutions of this work, I think the main thing to work on is a more clear choice of perturbations. As another reviewer points out above, just adding a bunch of knobs isn't necessarily very helpful because a large combination of knob settings will be relatively uninteresting (either too hard, or hard in a way that is unrealistic, or maybe even too easy).
> > > >
> > > > We think the process of developing a powerful search space and putting this in the hands of the user is in itself an important part of the research.
> > > >
> > > > One of the negatives of instantiating specific toy versions which seem interesting is that it biases the conclusions that can be drawn for the real distribution of interesting environments towards the ones instantiated. For instance, the exploration dimension in bsuite includes deep_sea and cartpole. These 2 very specific instantiations cannot possibly reliably capture the rich diversity of exploration problems in reality. However, if we try to capture anything close to the true diversity, we would need many, many more specific instantiations and that is not what our toy environments are meant to capture but more specific complex benchmarks.
> > > >
> > > > We felt the tool itself was like providing a researcher with a programming language for MDPs where they could define and explore their own interesting search spaces. We gave a glimpse into what was possible and tried not to impose what was the more interesting part of the provided search space from our point of view.
> > > >
> > > > >Tasks such as 'target radius' seem very specific, and I'm not sure how I would use such a perturbation to validate a particular experimental insight.
> > > >
> > > > We would highly appreciate if you could provide further details on why *target radius* seems too specific and point us to other ways on how we could define whether a target goal is reached in *continuous* environments.
> > > > We did gain the insight of the SOTA agents we tested not being adaptive near the target. If we designed an agent that was adaptive the closer it was to the target, we could easily verify this on the toy environment. Humans naturally slow down but this is not the case with these SOTA agents.
> > > > Further, since MDPP provides all these knobs, it is straightforward to turn *target radius* off.
> > > >
> > > > >Perhaps one question is whether you're using these environments for other research projects or simply producing them as they stand. Starting to write some papers that leverage these particular challenges could provide insights into which ones are really useful, and could help you whittle down the selection some.
> > > >
> > > > Thank you for your advice. We do use these for debugging, similar to examples we have provided, because when debugging in complex environments we have no clear signal of what is breaking. We also gained high-level insights such as for target_radius that we mentioned. We do not use it for the final testing of agents, for which we still use the complex benchmarks.
> > > > As we have noted in the insights in the paper, if the environments are tuned properly, dimensions such as action range and time unit will not have any clear impact but for a completely new environment, one cannot say in advance which of these dimensions will be most relevant. Another example of such a dimension is reward scaling. A lot of these agents, by performing reward normalisation or reward clipping, lessen or remove the relevance of this dimension. So, it’s hard to narrow down the dimensions, because anyone of them might be important for a given application domain depending on various other factors.
> > > >
> > > > >As to bsuite specifically, it was put on arxiv Agust 9th 2019 - I haven't tried to google your title to respect anonymity, but the claim of contemporaneous work does seem perhaps a bit extreme 1.5 years later…
> > > >
> > > > We apologise if we didn’t mention the exact dates of our paper submissions. We hoped that mentioning the 3 iterations of the paper would provide an idea of how old it is. To be clearer, we presented at the same workshop as the bsuite authors the submission deadline for which was September 9th 2019. Our arxiv date was about a week later. The workshop paper was basically the 1st iteration followed by the other 2 iterations we have mentioned.
> > > >
> > > > >As it stands I will maintain the review as it is stands but if there is a revised paper I will read it and provide feedback.
> > > >
> > > > Thank you very much for your continued support to provide further feedback.

---

### Official Review · AnonReviewer3 · 2020-10-29
**Review -- after rebuttal**

**Rating:** 6
**Confidence:** 4

**Review:**

The paper describes a new benchmark for evaluating reinforcement learning techniques (MDP-playground). It can be seen as a toolbox allowing to generate different MDPs with different characteristics. Each MDP will then be used to probe a particular ability of learning algorithms, resulting in a comparison of methods over multiple dimensions. Proposed dimensions are reward sparsity, stochasticity, delayed reward, etc....  In addition to this toolbox, the authors also evaluate some of the classical algorithms in the domain.

== Comments

First of all, the idea of evaluating RL techniques over multiple dimensions is very interesting, since right now the comparison of RL techniques is very weak, and providing simple tools in that direction is crucial to make  advances in the field. The MDP-playground approach is a good approach toward this goal and proposes a large number of different metrics on both continuous and discrete MDPs, making this platform the most complete in the domain.

But I identify two drawbacks in the proposed toolbox: first of all, if many metrics are proposed, it is very difficult to know which of these metrics are really relevant, and which are not. Said otherwise, the methodology would gain if these metrics could be connected to real use-cases e.g what are the relevant dimensions underlying atari environments, robotics ones, etc... Right now, imagine I evaluate my model over all the different metrics, and compare my model to other models, I still don't know which model I have to choose for solving a concrete use-case. A second drawback is readability: the approach is somehow proposing too many metrics such that being able to understand which model is good and which model is bad is very difficult. I would propose the authors to think about organizing these metrics in a way that they can be easily presented to users (e.g using spider plots ? by using a hierarchy ? ) At last, the paper is just comparing a few models over these metrics, while I would expect to have a more complete benchmark of existing models.

To conclude, if I really like the approach proposed in this paper, and if I think that it is a nice step toward a better evaluation of RL algorithms, I find that the paper is lacking some important characteristics to make MDP-playground really usable: i) a good way to summarize the performance of RL algorithms too many metrics allowing a good understanding of the methods ii) a comparison of more algorithms and iii) a  link between the proposed metrics and classical benchmarks.

==
Considering the modifications made on the paper, I increase my score

---

> ### Author Response · Authors · 2020-11-17
> **Response to AnonReviewer3**
>
> >first of all, if many metrics are proposed, it is very difficult to know which of these metrics are really relevant, and which are not.
>
> This is just the nature of RL - there are too many dimensions and that is why RL is so hard. In different application areas, different dimensions are more relevant and that is the domain of more specific benchmarks - they are meant to capture and instantiate more specific problems. In fact as our qbert example (from 3.1) in common response to all reviewers) shows, different dimensions can be relevant even within the same environment at different points in time and space and this is also true of the real world. Another example we would like to mention is that of Atari. It comes in deterministic and non-deterministic versions. While for some researchers, performing well on deterministic Atari could be a worthwhile cause, for others they might want to only really work with non-determinism. For the former, noise would not be relevant at all. So, any kind of weighting of relevances of metrics is specific to the application at hand and we have refrained from doing it.
>
>
> >Said otherwise, the methodology would gain if these metrics could be connected to real use-cases e.g what are the relevant dimensions underlying atari environments, robotics ones, etc.
>
> We are not really sure whether this concern is regarding textually describing underlying dimensions which we have done by motivating the dimensions how these may be relevant in different domains. We could improve and detail that description if you like. If this concern is regarding being able to measure the “amounts” of these dimensions in the complex environments, then we have addressed it in 3.1 and 4.2 in the common response to reviewers.
>
> >I would propose the authors to think about organizing these metrics in a way that they can be easily presented to users (e.g using spider plots ? by using a hierarchy ? )
>
> Thank you for the suggestion. However, one drawback of the spider plots is that it forces us to impose a subjective valuation of the different dimensions. That's why we refrain from setting a fixed grid of values for a dimension and allow fine-grained control over the amount of R noise (or any other dimension) that a user can inject into the environments. The experiments we have plotted in the paper are just a sample of the experiments possible with MDP Playground. Users could easily choose a different grid for another experiment. We do, however, also see the benefit of including spider plots with user-chosen weighting of relevance of values of the dimensions and are working on adding those for the rebuttal. Would that work for you?
>
> >At last, the paper is just comparing a few models over these metrics, while I would expect to have a more complete benchmark of existing models.
>
> We have compared 3 SOTA deep RL methods and also 3 tabular methods in the appendix. The transfer experiments to more complex environments for the deep RL methods has led to extensive resource usage on our side for the Atari and Mujoco environments and this is apart from the hyperparameter tuning which in itself is expensive. We are happy to run more agents on the toy environments though. Would that be alright for you?
>
> >since right now the comparison of RL techniques is very weak, and providing simple tools in that direction is crucial to make advances in the field. The MDP-playground approach is a good approach toward this goal and proposes a large number of different metrics on both continuous and discrete MDPs, making this platform the most complete in the domain.
>
> Thank you for the kind words.

---

### Author Response · Authors · 2020-11-12
**Thank you message to Reviewers**

Dear Reviewers,

Thank you for taking the time and effort for reviewing our paper and for the detailed feedback. We hope you and your loved ones are all safe and healthy in these uncertain times. We are glad to see that you found our work interesting, that you agree with us that we address an under-studied problem and that you see a lot of potential in our work. We value your detailed feedback and want to take the time to properly address all the concerns. We just wanted to let you know that we have already read your feedback and are preparing consolidated responses where we address common and individual concerns of the reviewers and will upload it soon.

Best wishes,
MDP Playground Authors.

---

### Author Response · Authors · 2020-11-17
**Common response to reviewers part 1**

## Principle of selecting metrics (1)
We apologise if our principle for selecting the metrics wasn’t clearly stated in the paper. The principle was to **exhaustively** come up with orthogonal dimensions (Sections 5 and 6 in the paper; all dimensions are listed in Appendix A with detailed descriptions in Appendix B and algorithm for generating MDPs with these dimensions in Appendix C). The **orthogonal** is very important. This is so that we can apply the different dimensions independently of each other in an environment. A further detail is that we went over each component that defined our MDP (such as *S*, *A*, *P* and *R*) and based on the literature, tried to assess what difficulties are seen regarding these and then exhaustively add them to MDP Playground. With regard to this, we observed that far too many possible interesting functions can define *P* and *R*, so for defining these some subjectivity on our side had to be imposed. After setting the dimensions relevant to *P* and *R*, the remaining aspects related to these are selected randomly so as to not promote overfitting by agents. We do know that some users will want flexibility here as well and so we allow users to use their own custom *P*s and *R*s in which they can specify them using Python functions. This is currently disabled in the code but can easily be enabled with a bit of work and we’re happy to have it ready for the rebuttal. The nice thing about this is that users don’t need to worry about being able to add these dimensions on top of their *P*s and *R*s because our code automatically takes care of it for them. Additionally, we also have wrappers for Atari and Mujoco which allow adding some of these dimensions in these popular benchmarks without having to worry about coding this themselves. It takes a lot of low-level fiddling with these benchmarks to enable this and this is another contribution we make for the community.

## Commitment to adding new dimensions (2)
We are happy to add whatever dimensions users consider as useful parameters of the environment as long as they meet the orthogonality principle we have set out. We could set up a voting system or a committee similar to the bsuite committee to add new dimensions.
In the future we hope that MDP Playground will become a community-driven benchmark, in which new dimensions are added by the community, to suit the needs of the community.

## Application domain of MDP Playground (3)
We believe there are some misunderstandings regarding the application domain of MDP Playground. We apologise if we didn’t clearly state this in the paper but we view the platform as a bridge between theory and practice. We have tried to make it as parameterisable as possible and allows users fine-grained control over defining their own experiments. The focus is on providing a toy benchmark and looking at high-level abstractions of difficulties. The motivation is to standardise the initial part of the RL development pipeline where researchers define their own toy MDPs and **not** to give insights into the more complex parts of environments such as capturing interactions between degrees of freedom for a robotic arm. We cannot hope to capture every specific domain in the community. We also agree with the bsuite authors in this regard. They also accepted in their rebuttal that they cannot hope to capture every different kind of dimension. (e.g., see https://openreview.net/forum?id=rygf-kSYwH&noteId=r1xvfq6WiH and https://openreview.net/forum?id=rygf-kSYwH&noteId=B1gnI0T-iH)

---

### Author Response · Authors · 2020-11-17
**Common response to reviewers part 2**

### Developing new algorithms with MDP Playground (3.1)
We hope our benchmark will help identify the inductive biases needed for designing new RL algorithms without getting bogged down by other sources of "noise" in the evaluation just as MNIST helped to identify the inductive bias of convolutions. MNIST by no means represents the true distribution of image data in the real world but it retains the key properties needed to identify inductive biases needed to perform well on image data. In the same way we believe we retain key dimensions of the problems which need inductive biases to be identified for them. It could be argued that the “deep” NNs in DQN have made use of the inductive bias of convolutions on image data. However, there are still many problems with NNs, as can be evidenced by how difficult RL still is. A lot of these problems we believe are captured by our dimensions.

Further, the fact that CNNs can classify and learn even random noise (e.g., see [1] and [2] below) shows that even random data should be sufficient to identify inductive biases as long as the key dimension is present in the data. Additionally, since the formulation of most RL algorithms we are aware of is agnostic to the structure of *P* and *R*, even random *P*s and *R*s should allow us to identify inductive biases.

Finally, these dimensions can be present in different “amounts” within the same environment across space and time. To give an example for a specific environment, in the Atari game qbert initially when no cubes have been stepped on you get a reward for jumping to any cube with a single action but later on, when enough cubes have been stepped on, you need to follow a sequence of steps to, say, the nearest unstepped on cube to obtain a reward. To calculate all the possible sequences that would lead to a reward at even a single instant of time would itself be an enormously difficult task. If we tried to do it across whole episodes, it would, in fact, become intractable, given the state and action space of qbert, even if we programmed it. And yet humans can tell the rough “amounts” of these dimensions at any given instant. Our goal in the long term is to make machines be able to identify and roughly measure the “amount” of each of these dimensions in smaller, independent sub-spaces of environments by breaking down the environment into parts just as humans can. And we want to enable researchers to play around with new ideas to be able to identify inductive biases needed to do that.

[1] Arpit, Devansh, et al. "A closer look at memorization in deep networks." Proceedings of the 34th International Conference on Machine Learning-Volume 70. 2017.

[2] Chiyuan Zhang, et al. “Understanding deep learning requires rethinking generalization” 5th International Conference on Learning Representations, {ICLR} 2017, Toulon, France, April 24-26, 2017, Conference Track Proceedings

### Testing and Debugging with MDP Playground (3.2)
As we have said in the paper (last paragraph section 5), we consider bsuite (and other more complex benchmarks) to come later in a *curriculum* of benchmarks than ours, that is to say, they are more towards specific instantiations of application domains while we consider our benchmark to be in between theory and practice and to be used first in the pipeline for RL algorithm testing and development. As example use cases, we point to how we have used MDP Playground for debugging existing RL agents in Appendix E.1. In the first example there, the key feature of reward sparsity helped us debug the behaviour of bsuite’s DQN epsilon-greedy implementation. Such an observation couldn’t have been easily made from the performance of DQN on bsuite’s environments because they had other confounding factors present in their environments while we only needed the key feature of varying reward sparsity to get this insight. **This is the problem with debugging on real environments: they are too complex; sometimes being unrealistically simple can be useful.**

---

### Author Response · Authors · 2020-11-17
**Common response to reviewers part 3**

## Common concerns among reviewers (4)
### Structure dimensions into groups (4.1)
R3: *A second drawback is readability: the approach is somehow proposing too many metrics such that being able to understand which model is good and which model is bad is very difficult. I would propose the authors to think about organizing these metrics in a way that they can be easily presented to users (e.g using spider plots ? by using a hierarchy ? )*

R4: *Overall I think this is a good direction of work, but it is a bit too unprincipled, and the paper structure kind of confusing. I would prefer perhaps less degrees of hardness, or perhaps a couple 'families' to make your thought process easier to understand.*

We hope the principle (1) above now makes the principled nature of the work clearer.
By design, the dimensions are orthogonal, which complicates grouping. However, we can group them by the component of MDPs that they impact. Would that work for you? If the reviewers have concrete suggestions or propose another specific hierarchy of these, we would be happy to use those.

### Measure dimensions in real-world environments / Match real-world distributions (4.2)
R3: *Said otherwise, the methodology would gain if these metrics could be connected to real use-cases e.g what are the relevant dimensions underlying atari environments, robotics ones, etc.*

R4: *MDP Playground exacerbates this issue -- since its MDPs are randomly generated and don't have a natural interpretation, it's difficult to get even an intuition for a good behavior in them, and difficult to understand how to generate problems that that have a combination of hardness along different dimensions that is representative of realistic scenarios.*

R2: *Tangentially, it might be interesting to have a tool that could analyze a particular complex environment and automatically generate a corresponding Playground MDP that would somehow capture the same measures of difficulty, such that agents could be debugged/optimized in this low-cost environment before being transferred back to the complex environment.*

Yes, that’s a very nice idea indeed and we have thought of that too but we believe it is intractable to measure the “amounts” of these dimensions in real world environments, even within the same environment. For a concrete example of how difficult it would be to do this, please see the qbert example in 3.1 above. We believe if it could be as simple as being able to measure in real-world environments what the "amount" of each dimension was, RL would be much easier to solve. This is why we do not claim to be able to capture the variance of real world environments in MDP Playground and view it as a bridge between theory and practice that should be used for design, development and testing. We must make it clear that we are not trying to capture real world distributions. There is so much complexity needed to do that that even more complex benchmarks can’t claim that performance on their benchmarks always transfer to real world environments. There is a whole Sim2Real community dedicated to researching this aspect of RL. Our aim is to give researchers a powerful tool to identify inductive biases and test and debug the way we have mentioned in 3.1 and 3.2 above.

---

### Author Response · Authors · 2020-11-17
**Common response to reviewers part 4**

### Unrealistic environments (4.3)
R1: *MDP Playground's problems are completely unintuitive. They are randomly generated and aren't inspired by any real scenarios. This raises the question of how meaningful they are.*

R2: *Additionally, why do we believe that the structure of the MDPs generated by MDP Playground will resemble that of the problems that RL practitioners in the community are interested in solving? Specifically for discrete environments having a completely connected transition function consisting of 8 states and 8 actions seems like it may not resemble more complicated environments like Atari.*

We have discussed in 3, 3.1 and 3.2 how MDP Playground can be used for practice and theory and why being unrealistic can be useful. We have discussed the design decisions regarding the MDPs used in our experiments in Appendix D. We have chosen the regular, "high bias", completely connected structure due to the very reason that imposing a structure which instantiates very specific *P*s moves in the direction of more specific applications which is the domain of complex benchmarks. There will be occasions where we use *P*s and *R*s which can seem unrealistic but many theoretical researchers will still be happy to use them we believe. For practical researchers, we have shown that high-level trends for some of the dimensions transfer to more complex benchmarks despite the toy nature of the MDPs in MDP Playground. For practical researchers aiming to be even closer to their empirical scenarios, we do in fact allow users to use define custom *P*s and *R*s so that they can apply the dimensions to any toy MDP which they think captures essential characteristics of the *P* of the environment they are interested in. It's only for the experiments that we have imposed a certain structure (and this is discussed also in Appendix D). We do not want to bias our environment distribution in the direction of a specific domain. For practical researchers aiming to directly inject our dimensions in their complex environments, we have Atari and Mujoco wrappers for some of the dimensions and will continue to develop wrappers for the ones it is possible to develop them for.

## Short history of paper and contradictory emphasis proposed by reviewers (5)

We would also like to provide a short summary of the history of the paper so that some of the statements we make below are clearer to the reviewers. This is the 3rd iteration of the paper. In the 2nd iteration, we added many more dimensions (including all the continuous ones) in addition to making other improvements as asked by reviewers. In the 3rd iteration, we added the experiments showing transfer to more complex environments, clarified the principle of choosing the dimensions and added tabular agents in addition to other improvements as also asked by reviewers. We are deeply disappointed that after putting in so much hard work, the scores have actually decreased in this iteration. While we value the constructive criticism of the reviewers and reiterate our commitment (2) to always keep improving and making the platform more usable, we believe we have reached a point where a lot of the suggested future directions to improve the paper, while given in the earnest and best interests of our paper, seem to us to contradict either past or current suggestions. In past iterations, reviewers largely held a consensus view and that made it simpler for us to improve our platform. We will now show what is confusing us and would kindly ask the reviewers to also please read each other’s reviews and to please provide us with concrete suggestions to move forward.

---

### Author Response · Authors · 2020-11-17
**Common response to reviewers part 5**

### More description of dimensions vs more description of insights (5.1)
R2: *but little is shown about new insights gained toward understanding shortcomings of existing algorithms*

R4: *More generally, not all dimensions were clearly described, in particular stochasticity, it wasn't clear to me how this was defined. I would have appreciated more time spent on describing the challenges rather than the analysis of the results on all sorts of environments, in the end the core contribution here is the framework and its structure, the analysis is slightly out of scope given the length of the paper.*

R4: *Then, further discussion and grounding for each family of tasks would be great, to understand where these dimensions of hardness would manifest themselves.*

R4: *Finally, spending more time on describing each hardness dimension clearly instead of compacting it all into the end of Section 2 would also make this an easier read.*

Thank you for the suggestions. We want to note that we have cited literature wherever we could in order to support our selection of dimensions. Further, in previous iterations of the paper, we have included detailed descriptions of the dimensions in the main paper; since the past reviewers suggested that we discuss the insights from the experiments more than the dimensions themselves, and since there are many dimensions now, we moved the detailed descriptions of the dimensions to Appendices A, B and C. Since this paper tackles many different aspects of RL, we think many researchers’ takes will be different on what is more important. Most previous reviewers were more interested in seeing transfer experiments on complex environments, and that’s why we focussed on these. If all the reviewers agree, we can move the discussion of dimensions to the main paper and move transfer experiments to the Appendix. Personally, we like all the aspects of MDP Playground and have put in a lot of hard work and wish we could include everything in the main paper, but of course, we want to focus on what’s most interesting to the community.

R1: *A similar remark goes for several other experiments: varying time units, target variance, irrelevant features. Their results are completely expected, and while they might be a useful sanity check, the results description can be condensed to 1 line each and placed in the figure captions.*

What R1 has termed "sanity check" is exactly what previous reviewers asked for. They wanted to see transfer between toy and complex domains. They also wanted us to discuss the experiments and results in more detail.

R1: *A similar comment goes for the entire content of Section 4.3 as well-- it is very verbose and can be greatly condensed and structured into a short list with bullet points summraizing the findings.*

Again, we are deeply disappointed to read this: all previous 7 reviewers of the paper wanted more about transfer to complex domains and that's precisely what we have done now; the transfer to complex environments is what we believe should also allay concerns about the toy MDPs being unrealistic. Could you please think about this more and let us know what you think? We have shown, in keeping with our view that our platform is a bridge between theory and practice, that even if the toy MDPs could seem unrealistic to some readers, the trends of the agents on the dimensions transfer to more complex environments. If the dimensions were something only relevant in toy environments, one would have expected really unreliable results in complex environments. As for showing stronger forms of transfer, we would like to reiterate what we said in 4.2 above: even more complex benchmarks can’t claim that performance on their benchmarks always transfer to real world environments; so, we are at a bit of a loss how we should achieve this.

---

> ### Comment · AnonReviewer1 · 2020-11-19
> **Verbosity**
>
> Regarding my points about the verbosity of experimental result descriptions, your feeling of disappointment may have been caused by a misunderstanding. I wasn't suggesting dropping this content, but rather phrasing it more concisely.

---

> > ### Author Response · Authors · 2020-11-23
> > **Thank you for clarification**
> >
> > Thank you for trying to clear a possible misunderstanding. We did understand what you had asked for and were merely pointing out that some of the other reviewers (past and present) have asked for a more detailed description of these insights and shorter descriptions of the dimensions. It is unclear to us which of these two to have more of in the paper because reviewers seem to differ.

---

### Author Response · Authors · 2020-11-17
**Common response to reviewers part 6**

### Adding more dimensions vs reducing number of dimensions (5.2)
R4: *I would prefer perhaps less degrees of hardness, or perhaps a couple 'families' to make your thought process easier to understand.*

R4: *I find the use of two procedurally generated MDP, one continuous and the other discreet, in a very simple task setup to be a good design choice, and also allows for quick iteration.*

R1: *A case in point is the paper's omission of MDP non-ergodicity (and presence of constraints on the desired policy as a common real factor that causes non-ergodicity) from its list of hardness dimensions.*

R2: *Similarly, moving a pointmass in a 2D plane likely has many differences from learning how to locomote a multi-jointed robot.*

We are disappointed that some of you seem to feel that MDP Playground has too many dimensions. Since the dimensions in MDP Playground are orthogonal, researchers who don't care about a dimension can leave it at its default which basically turns it off. If the concern is only about readability, as we have said above, we’ll try to group them to make the paper easier to read. In an earlier review, one of the main reasons for rejection was that we had too few dimensions. Reducing the number of dimensions would be in violation of our principle (1) above of how we selected the dimensions exhaustively. On the other hand, it is near impossible to capture all the breadth and depth of RL. Our work aims to distill at a high-level the difficulties faced in RL. Most research goes in depth in a research direction and trying to distill the truly relevant dimensions across the breadth of RL is a very hard task. This is why we have said in the paper (Section 6) that we can’t possibly have collected all the relevant dimensions and welcome dimensions from readers that they think are relevant.

## Concrete Improvements (6)
Based on the reviewers’ feedback, we feel the concrete ideas that can objectively improve the paper are:
1. Enabling custom *P* and *R*s again to allow users the power and flexibility to move in the direction of a more specific domain that they prefer.
2. Enabling a community voting mechanism (e.g. through voting for open Github issues) and/or an expert committee which adds new dimensions to MDP Playground.
3. Grouping together dimensions under the component of MDPs that they affect, i.e., *S*, *A*, *P*, *R*, etc.
4. Specific changes regarding concerns of individual reviewers including adding spider plots with optional user-defined weighting of relevances of values of each dimension, improving text to be more accurate, and updating related literature (for context regarding these, please see our individual responses to the reviewers).

We can have all of these easily ready by the end of the rebuttal and will upload PDFs with intermediate changes so that you can see these changes as soon as possible.

We would like to point out that the RL algorithms we have tested are supposed to be general RL algorithms and yet they have not been the silver bullet that we had hoped they would be. Research is always a work in progress and meant to be bettered and while we appreciate the strong opinions our research tends to evoke, we believe we have reached a point where these have become too varied to satisfy at once. We believe placing such a powerful tool in the researcher’s toolbox is a great boon as it lets them focus at a high level without having to fiddle with low-level code and would hope reviewers see our point of view too and increase their scores.

---

### Comment · AnonReviewer1 · 2020-11-19
**Key issue: high-level response to the authors' rebuttal**

First of all, I honestly sympathize with you having been getting contradictory feedback on several iterations of this work.

At the same time, having read other reviews and re-read my own, I see one common theme jump out at me, although different reviewers talk about it slightly differently. I bet that if you were to address this criticism somehow, reviewers' reaction to this work will be very different.

This criticism is that MDP Playground MDP's don't *feel* realistic/relatable.  Sorry to be blunt here, but long explanations about why these MDPs are still useful don't help much in addressing this, and numerous statements about how disappointed you are to read reviewers' reactions don't either. MDPs modeling real problems are points or small clusters in the vast space of hardness dimensions. Most of this hardness space is uninteresting, and finding these interesting clusters by twisting separate knobs for each hardness dimension is almost impossible. Complaints about there being "too many" hardness dimensions are a reflection of this -- the more independent knobs there are, the harder it is to find practically interesting problem instances.

Here is another way to look at this: most practically impactful RL algorithms are those that work well on problems that *feel* realistic/relatable. Of course, each such algorithm will have its failure modes. However, even among failure modes there are "interesting" and "uninteresting" ones. Most failure modes corresponding to random hard MDPs are uninteresting.

I see at least two ways of dealing with this issue:

- Redesigning MDP Playground to keep the same or even larger number of hardness dimensions, but drastically reduce the number of knobs, so that variations across sets of hardness dimensions become correlated.

- "Selling" this work in a different way, so that the lack of realistic *feel* of MDP Playground's MDPs isn't a problem.

---

> ### Author Response · Authors · 2020-11-23
> **Response to AnonReviewer1's Feedback**
>
> Thank you very much for your concern and detailed explanation of why the realistic feel aspect of MDP Playground would be important for readers. After reading the initial reviews, we did feel that we have not clarified the application domain of MDP Playground well enough. And having an explanation of where MDP Playground fits in in the RL development pipeline was, indeed, an attempt at clarifying things and selling things differently. We don’t think toy MDPs can really be used towards the end of the RL development pipeline, i.e., real environments, because they are far too complex to be captured. Do you think there *is* a way **toy MDPs** can capture everything that is needed from a real environment when even complex benchmarks struggle to do that?

---

> > ### Comment · AnonReviewer1 · 2020-11-24
> > **--**
> >
> > I don't think toy MDPs need to capture every aspect of a real environment. If they capture the difficult combinations of aspects that more complex benchmarks capture, this will be good enough. The problem with complex benchmarks is the computation and complexity cost of working with them. If toy MDPs can capture as much as those complex benchmarks can but without the computation and complexity overhead, it's a win.

---

### Author Response · Authors · 2020-11-25
**Final thank you comment to all the reviewers**

Dear Reviewers,

We have submitted a final version of MDP Playground incorporating the concrete improvements we had summarised from your feedback:

1. Enabling custom *P* and *R*s to allow users the power and flexibility to move in the direction of a more specific domain that they prefer
2. Categorising dimensions according to the component of (PO)MDPs that they affect, i.e., *S*, *A*, *P*, *R*, etc.
3. Provided a more detailed description of many of the dimensions in the main paper instead of the Appendix
4. Adding spider plots functionality to the analysis Python notebook
5. Improving text to be more accurate with regard to POMDP terminology
6. Updating related literature to include the *Challenges of real world RL* benchmark and the *How hard is my MDP* paper
7. Adding a separate section on using MDP Playground including the spider plots, a plot of an experiment varying 2 dimensions together and debugging use cases. We also moved most of the complex experiments to the appendix and briefly discussed them in this new section
8. Enabling a community voting mechanism (e.g. through voting for open Github issues) and/or an expert committee which adds new dimensions to MDP Playground


We would like to once again thank all the reviewers very much for their comments and the time and effort they took to review our paper. We wish them all continued health and a joyous festive period.

---

### Decision · Program_Chairs · 2021-01-07
**Final Decision**

**Decision:**

Reject

**Comment:**

I thank the authors for their submission and very active participation in the author response period. The reviewers and I acknowledge the importance of designing toy environments that allow the community to systematically investigate strengths and weaknesses of RL approaches. That said, the reviewers have criticized that it is unclear whether experiments on the proposed toy MDPs would transfer to more complex standard RL benchmarks (such as Atari) [R1 & R2], and that the proposed metrics and axes of variation seem not well motivated or systematic [R1,R2,R3 & R4], thus casting doubts regarding what insights the community will be able to gain from experiments on MDP Playground. In particular, I agree with the reviewers R1's and R4's assessment that proposing many dimensions of variation, even if they are orthogonal, without a well formulated motivation and grounding in actual tasks the community cares about is not particularly helpful for advancing our understanding of current challenges in RL. Post rebuttal, R2 and R4 stand by their strong stance against acceptance; and R1 has increased their score as a result of the improvements of the updated paper, but they still lean towards rejection. Thus, I recommend rejection.